**Resource**

# Longitudinal autophagy profiling of the mammalian brain reveals sustained mitophagy throughout healthy aging

Anna Rappe [ID] [1], Helena A Vihinen [ID] [2], Fumi Suomi [ID] [1], Antti J Hassinen [ID] [3], Homa Ehsan [ID] [1], Eija S Jokitalo [ID] [2] & Thomas G McWilliams [ID] [1,4✉]

## Abstract

**Mitophagy neutralizes mitochondrial damage, thereby preventing cellular dysfunction and apoptosis. Defects in mitophagy have been strongly implicated in age-related neurodegenerative disorders such as Parkinson's and Alzheimer's disease. While mitophagy decreases throughout the lifespan of short-lived model organisms, it remains unknown whether such a decline occurs in the aging mammalian brain—a question of fundamental importance for understanding cell type- and region-specific susceptibility to neurodegeneration. Here, we define the longitudinal dynamics of basal mitophagy and macroautophagy across neuronal and non-neuronal cell types within the intact aging mouse brain in vivo. Quantitative profiling of reporter mouse cohorts from young to geriatric ages reveals cell- and tissue-specific alterations in mitophagy and macroautophagy between distinct subregions and cell populations, including dopaminergic neurons, cerebellar Purkinje cells, astrocytes, microglia and interneurons. We also find that healthy aging is hallmarked by the dynamic accumulation of differentially acidified lysosomes in several neural cell subsets. Our findings argue against any widespread age-related decline in mitophagic activity, instead demonstrating dynamic fluctuations in mitophagy across the aging trajectory, with strong implications for ongoing theragnostic development.**

**Keywords** Aging; Autophagy; Brain; Mitochondria; Mitophagy
**Subject Categories** Autophagy & Cell Death; Neuroscience

## Introduction

Aging precipitates the manifestation of many neurodegenerative diseases, particularly those characterized by loss of memory and locomotor control (Markaki et al, 2021). Mitochondrial integrity underpins healthspan and longevity in numerous model organisms, from budding yeast to nematodes to mice, and a key pathway through which cells alleviate the burden of damaged mitochondria is autophagy (Leidal et al, 2018; Killackey et al, 2020; Collier et al, 2021a; Klionsky et al, 2021; Rappe and McWilliams, 2022). Autophagy pathways degrade cargo in either a nonselective or selective fashion. During macroautophagy (hereafter referred to as autophagy), cytoplasmic contents can be indiscriminately degraded (Leidal et al, 2018). On the other hand, selective autophagy eliminates specific cellular components, such as damaged organelles, which become destined for autophagic destruction *via* specialized "eat me" signals. The best studied selective autophagy pathway is mitochondrial autophagy (mitophagy), which contributes to the controlled decommissioning of damaged mitochondria. Impairments in both basal macroautophagy and mitophagy have strong links to human neuropathology (Collier et al, 2021a, 2021b; Suomi and McWilliams, 2019) and muscle disease (Mito et al, 2022). Despite their essential contributions to neural integrity, the coordination and synchrony of distinct autophagy pathways within the aging mammalian brain remains poorly understood.

Once thought to be an acute stress response, our previous work established that mitophagy is a basal, homeostatic process that operates during normal physiology to clear damaged mitochondria through the autophagic pathway (McWilliams et al, 2016). This steady-state mitochondrial turnover occurs independently of mitochondrial stress-induced PINK1-Parkin signaling (McWilliams et al, 2018a, 2018b, 2019; Lee et al, 2018; Wrighton et al, 2021) in a cell- and tissue-specific fashion with a complex regulatory network (Schweers et al, 2007; Singh et al, 2021; Yamada et al, 2018; Hoshino et al, 2019; Alsina et al, 2020; Long et al, 2022; Gok et al, 2023; Elcocks et al, 2023). In short-lived model organisms such as yeast and nematodes, mitophagy levels decrease with age, leading to the widely held hypothesis that mitophagic capacity may also decline in aged mammalian tissues and could contribute to age-related neurodegeneration. Significant translational efforts are underway to enhance or restore mitophagy levels in these pathological contexts (Leidal et al, 2018; Killackey et al, 2020; Klionsky et al, 2021). However, because short-lived and

[1]Translational Stem Cell Biology and Metabolism Program, Faculty of Medicine, Biomedicum Helsinki, University of Helsinki, Haartmaninkatu 8, Helsinki 00290, Finland. [2]Electron Microscopy Unit (EMBI), Institute of Biotechnology, Helsinki Institute of Life Science, University of Helsinki, Viikinkaari 9, Helsinki 00790, Finland. [3]High Content Imaging and Analysis Unit (FIMM-HCA), Institute for Molecular Medicine, Helsinki Institute of Life Science, University of Helsinki, Tukholmankatu 8, Helsinki 00290, Finland. [4]Department of Anatomy, Faculty of Medicine, University of Helsinki, Haartmaninkatu 8, Helsinki 00290, Finland. ✉E-mail: thomas.mcwilliams@helsinki.fi

long-lived species have distinct evolutionary pressures, it remains unclear whether lessons learned from short-lived organisms and cell lines actually translate to mammalian physiology—a question that has great translational significance. Mammalian postmitotic neurons survive for decades, have high energy demands and are particularly sensitive to homeostatic impairments. Understanding organelle homeostasis in long-lived mammals is crucial, given how aging accelerates cognitive decline and disease.

How natural aging modifies mammalian mitophagy in distinct brain regions and cellular subtypes remains to be examined, because profiling mitophagy within intact tissues and brain circuits is not straightforward using conventional techniques, which include:

(1) Transmission electron microscopy (TEM). While TEM provides unrivaled definition of cellular membranes, it cannot reliably differentiate between selective and nonselective autophagy. In addition, its application in heterogeneous tissues is labor-intensive and not amenable to high-throughput spatiotemporal analysis, especially for anatomically expansive and anisotropic brain circuits.

(2) Biochemical measurements of mitochondrial protein depletion. Immunoblotting and sophisticated capture proteomics afford valuable and unbiased mechanistic insights. However, these approaches lack the spatial and cell-type specificity essential for understanding the complex spatiotemporal dynamics in clinically-important neural subtypes and circuits. Furthermore, the specificity of many mitophagy-related antibodies for mouse tissues is suboptimal compared to those used in human studies.

(3) Transcriptomics. Although tissue transcriptomes change with age, bulk RNA-seq results mask the spatial and cellular complexity of mammalian organ systems. While single-cell and single-nucleus RNA-seq methods are beginning to offer valuable insights, sensitivity is limited and there is no consensus on transcriptional signatures that accurately reflect mammalian mitophagic activity, which may exhibit cell and tissue-specific dynamics.

(4) In vitro systems. Cell culture paradigms afford high-resolution mechanistic insights, yet they lack the metabolic complexity of vascularized tissues. Most cultured cells exhibit negligible levels of basal mitophagy, requiring stress-induction to provoke measurable turnover levels, which differs from physiological mitophagy in vivo.

Thus, while major insights have been possible from tractable short-lived model organisms, the question of how autophagy pathways are modulated in natural mammalian aging has remained an intractable question. Overcoming these limitations, genetically encoded optical reporter mouse models have recently emerged as powerful tools to monitor specific stages of physiological autophagy and mitophagy in intact tissues at high resolution with cell-specific precision (Kuma et al, 2017; McWilliams and Ganley, 2019). Combining fixation-compatible reporter systems with immunohistochemistry assays further enables rich insights into organelle subtypes.

Here, using two genetically encoded reporter mouse strains, we tracked mitophagy and autophagy longitudinally throughout the mouse lifespan in several pathophysiologically important brain regions. We defined aging-related dynamics in mitophagy and autophagy, providing strong evidence that decreased mitophagy and autophagy are not general features of healthy mammalian brain aging. We highlight intriguing findings, including age- and region-specific decoupling of mitophagy and autophagy, that may reflect cargo prioritization and differential acidification of lysosomes in aged neurons. These findings provide a broad atlas of the selective and nonselective mitochondrial quality control capacity of the aging mammalian brain at steady state and suggest new directions for understanding cell type-specific susceptibility to neurological dysfunction.

# Results

## Mitophagy and autophagy are sustained in the mammalian brain throughout life

To investigate the impact of healthy aging on physiological mitophagy in vivo, we systematically analyzed brain tissues from male and female mice carrying one of two reporter constructs across an aging timecourse (based on the JAX guidelines on aging animals: mature (3 m and 6 m), middle aged (15 m), and geriatric (18–26 m); roughly corresponding to human age ranges 20–30, 38–47, and 56–69 years, respectively) (Fig. 1A). Hereafter, ages are referred to as follows; young (3 m), adult (6–7 m), middle aged (15–16 m) and geriatric (18–26 m). *mito*-QC mice harbor a genetically encoded mCherry-GFP construct that localizes to the outer mitochondrial membrane (mCherry-GFP-FIS1$^{mt101-152}$) (McWilliams et al, 2016; Allen et al, 2013), resulting in mitochondria that fluoresce in both red and green. During mitophagy, the fusion of acidic endolysosomes with autophagosomes quenches GFP fluorescence, whereas the mCherry signal remains unaffected by the pH shift (Fig. 1B,C). Thus, mitochondria within lysosomes (mitolysosomes) can be quantified as mCherry-only puncta in a variety of cell and tissue preparations. To uncouple basal mitophagy events from macroautophagy in vivo, we also analyzed brain tissues from *auto*-QC autophagy reporter mice across the same aging trajectory (Fig. 1D,E) (McWilliams et al, 2018a, 2019; Singh et al, 2021). The *auto*-QC model is highly similar to *mito*-QC with the same genetic background, but the reporter construct is replaced with mCherry-GFP-LC3 to label autophagosomes regardless of their cargo and detect lysosomal fusion and acidification through loss of GFP signal. Lysosomal activity is essential for the formation of mCherry-only structures in both *mito*-QC and *auto*-QC reporter systems. Classical flux assays demonstrate that Bafilomycin A1 treatment blocks the formation of mCherry-only mitolysosomes and autolysosomes, using deferiprone for mitophagy induction and the selective mTORC1 inhibitor AZD8055 for macroautophagy induction (Fig. EV1). Accordingly, these reporter systems enable a robust end-point readout of mitophagy and macroautophagy based on this and extensive prior validation in different contexts (McWilliams et al, 2016, 2018a; Lee et al, 2018; Singh et al, 2021). Because mitophagy is a specific type of autophagy, the comparison of *mito*-QC and *auto*-QC models enables us to dissociate these pathways and monitor the specificity of autophagy pathways and their dynamics in vivo.

In accordance with previous studies, we found that both *mito*-QC and *auto*-QC mice exhibit widespread fluorescence throughout the brain, but with differential patterns evident even at the

## Study overview

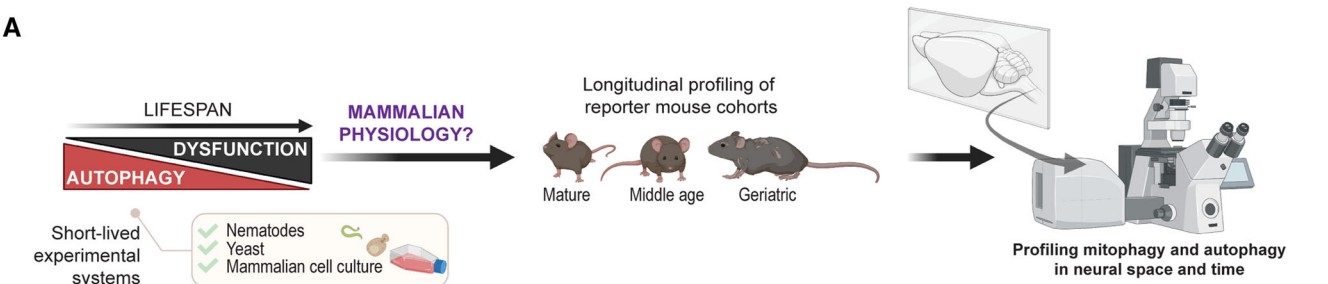

## Reporter systems

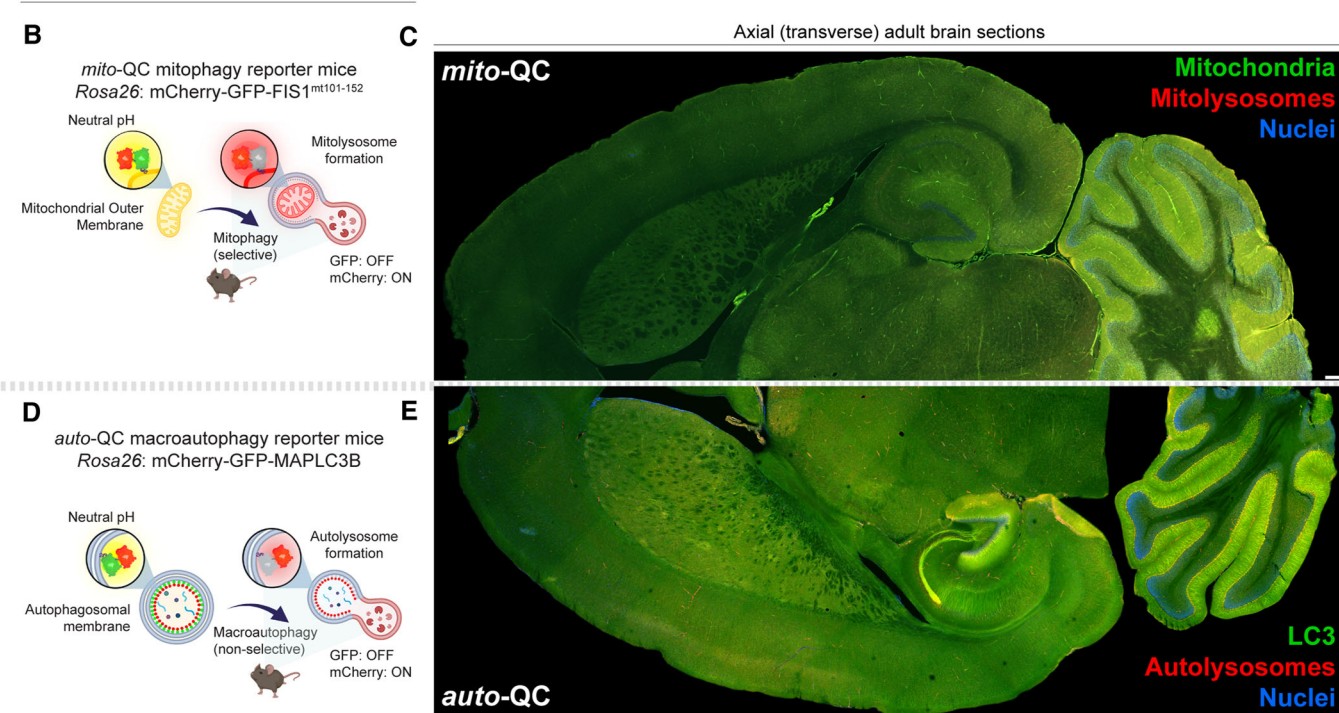

**Figure 1. Mapping longitudinal changes in mitophagy and macroautophagy in the aging mammalian brain.**

(A) Study overview. Mitophagy and macroautophagy decrease throughout the lifespan of short-lived experimental systems such as nematodes and yeast. Cell-specific alterations in mitophagy, and its relationship to macroautophagy in the aging mammalian brain are unknown. We examined the longitudinal progression of mammalian mitophagy and macroautophagy in natural aging using cohorts of reporter mice and quantified the spatiotemporal regulation of selective and nonselective turnover pathways in vivo using high-resolution confocal microscopy. (B) Mitophagy assay principle. The well characterized *mito*-QC mouse model (mCherry-GFP-FIS1[mt101-152]) reports the abundance of mitolysosomes in mammalian cells and tissues. Functional mitochondria fluoresce in yellow (red and green) due to the endogenous mCherry-GFP tandem tag on their outer mitochondrial membrane. A pH-dependent fluorescence shift is observed when mitophagy occurs, as GFP fluorescence becomes quenched within the acidic endolysosomal network. This approach enables mitophagy to be quantified as mCherry-only puncta within cells and tissues. (C) Confocal tile scan of an axial brain section from the *mito*-QC mouse, showing widespread fluorescence. Scale bar 200 μm. (D) Macroautophagy assay principle. The *auto*-QC macroautophagy reporter mouse was made identical to the *mito*-QC mouse and maintained on the same genetic background. The *auto*-QC reporter works through the addition of MAPLC31B to the mCherry-GFP reporter. When nonselective macroautophagy occurs, autolysosomes are visualized through a pH-dependent fluorescence shift, enabling macroautophagy to be quantified as mCherry-only puncta within cells and tissues. (E) Confocal tile scan of an axial brain section from the *auto*-QC mouse, showing widespread fluorescence.

macroscopic level. Confocal imaging revealed that mitolysosomes and autolysosomes are ubiquitous in CNS tissues throughout natural aging, challenging the general notion gleaned from short-lived model organisms that these processes decline in old age. In the following sections, we zoom in and elaborate on our cell type- and brain region-specific findings.

## Spatiotemporal dynamics of mitophagy and macroautophagy in the aging CNS

Since we observed significant heterogeneity between the mitophagy and autophagy reporters across gross anatomical regions of the brain, we next quantified mitophagy and autophagy levels across

## *mito*-QC regional analysis

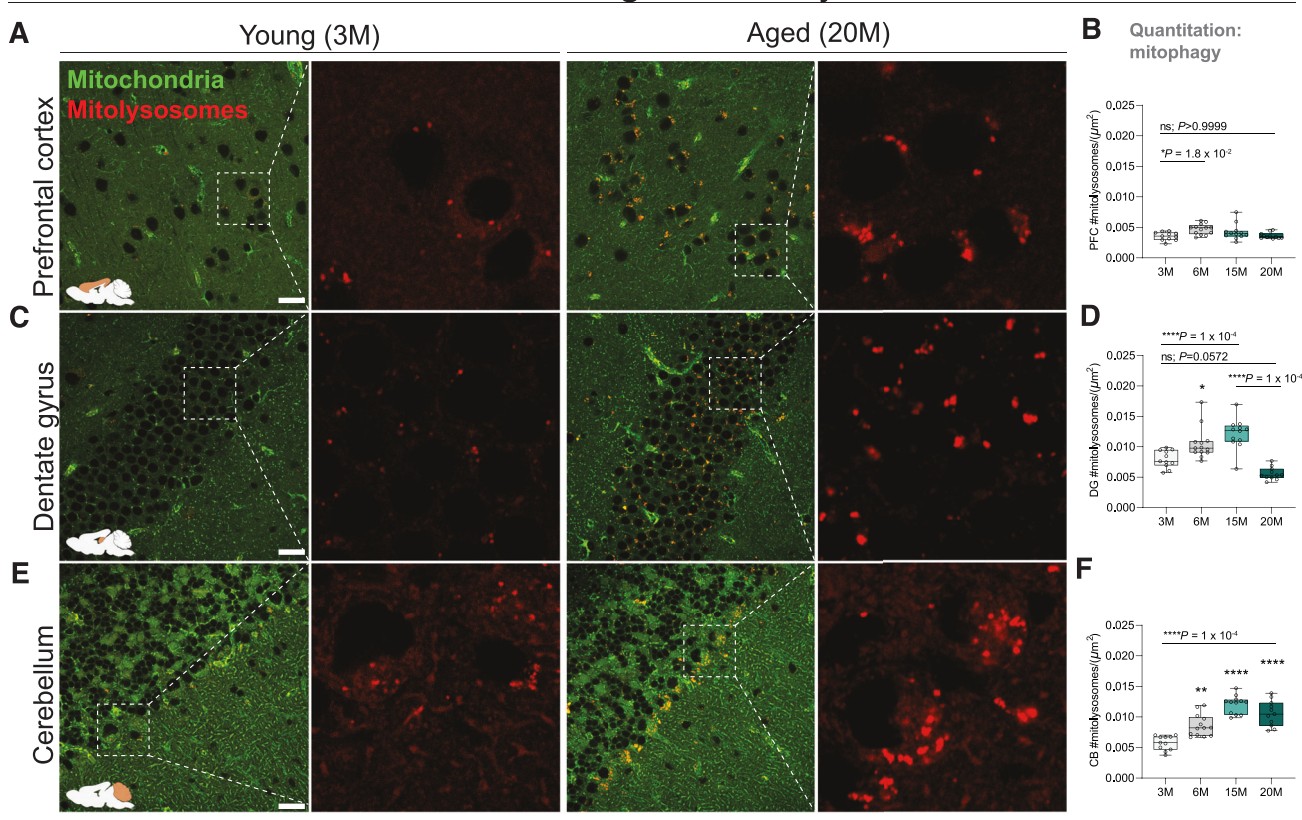

## *auto*-QC - regional analysis

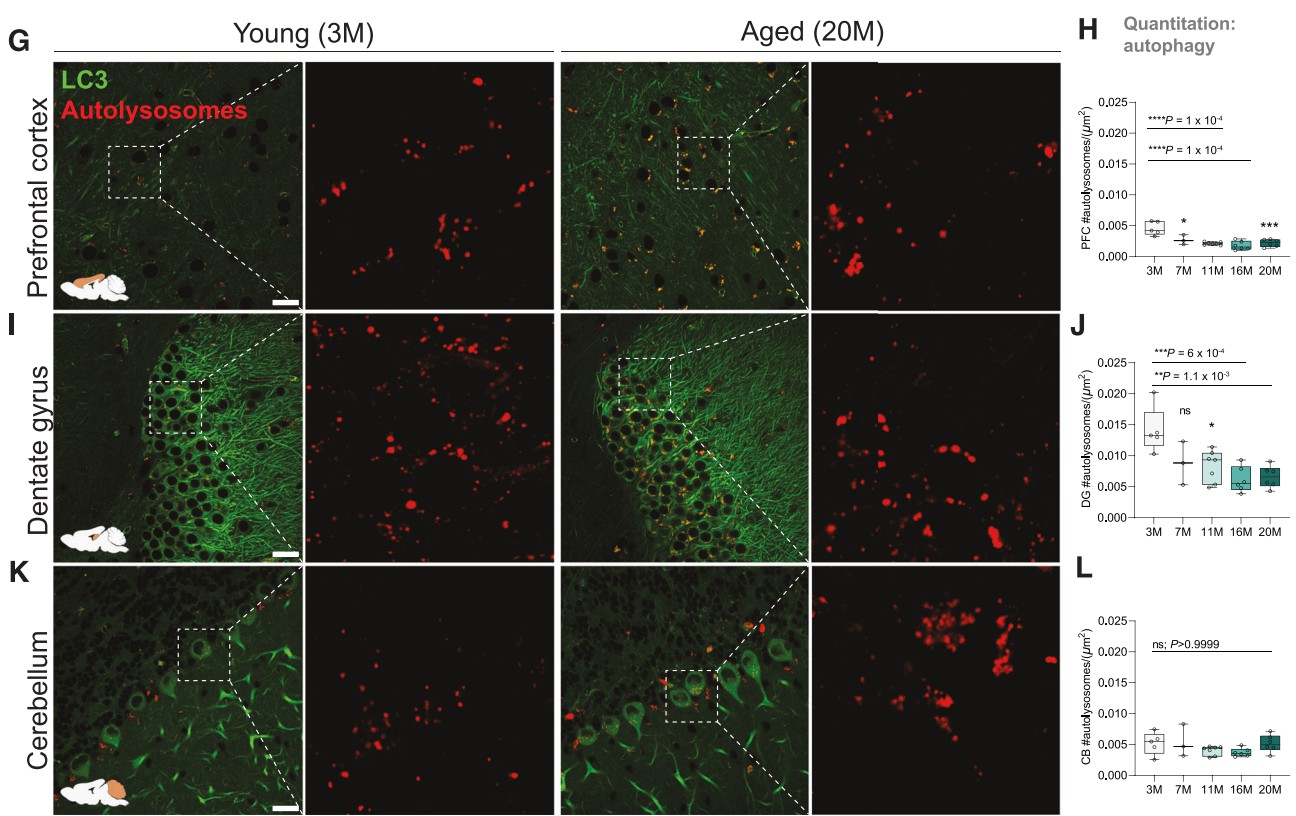

**Figure 2. Temporal dynamics of regional mitophagy and macroautophagy in the CNS.**

(A, B) PFC mitophagy. Representative confocal photomicrographs detailing instances of PFC mitophagy in young and geriatric mice. GFP and mCherry channels are shown for clarity, alongside quantitative analysis of PFC mitophagy levels across all ages. *$P = 0.0188$, ns = not significant; $P > 0.9999$. $n = 47$. (C, D) DG mitophagy. Representative confocal photomicrographs detailing instances of DG mitophagy in young and geriatric mice. GFP and mCherry channels are shown for clarity, alongside quantitative analysis of DG levels across all ages. ****$P < 0.0001$; ns = not significant; $P = 0.0572$. $n = 47$. (E, F) CB mitophagy. Representative confocal photomicrographs detailing instances of CB mitophagy in young and geriatric mice. GFP and mCherry channels are shown for clarity, alongside quantitative analysis of CB mitophagy levels across all ages. ****$P < 0.0001$. $n = 47$. Scale bar 20 µm. (G, H) PFC macroautophagy. Representative confocal photomicrographs detailing instances of PFC macroautophagy in young and geriatric *auto*-QC mice. GFP and mCherry channels are shown for clarity, alongside quantitative analysis of PFC macroautophagy levels across all ages. ****$P < 0.0001$. $n = 27$. (I, J) DG macroautophagy. Representative confocal photomicrographs detailing instances of DG macroautophagy in young and geriatric *auto*-QC mice. GFP and mCherry channels are shown for clarity, alongside quantitative analysis of DG macroautophagy levels across all ages. **$P = 0.0011$, ***$P = 0.0006$. $n = 27$. (K, L) CB macroautophagy. Representative confocal photomicrographs detailing instances of CB macroautophagy in young and geriatric *auto*-QC mice. GFP and mCherry channels are shown for clarity, alongside quantitative analysis of CB macroautophagy levels across all ages. ns $P > 0.9999$. $n = 27$. Scale bar 20 µm. Box plots extend from the 25th to the 75th percentiles, with a median line positioned inside the box. Whiskers denote the minimum and maximum values. Source data are available online for this figure.

the aging timecourse in functionally distinct regions of the CNS, focusing on brain regions of particular interest to neurological and neurodegenerative pathology. We quantified in vivo mitophagy and autophagy as a ratiometric index, where the number of mCherry-only mitolysosomes or autolysosomes is normalized to cell or anatomical area. Standardizing data as percentages relative to the youngest subjects did not affect the trajectories or patterns observed (Fig. EV2), and thus all in vivo measurements are presented as mitolysosomes or autolysosomes per area (µm²), which provides a readout of end-point turnover dynamics, consistent with conventions in the field (McWilliams et al, 2018a; Singh et al, 2021).

We first focused on the prefrontal cortex (PFC), which mediates decision-making, behavioral flexibility, attention, and aspects of working memory. PFC dysfunction is associated with several neuropsychiatric conditions, such as depression and schizophrenia (Chini and Hanganu-Opatz, 2021). We show that mitophagy levels in the PFC remain relatively stable across healthy aging, with a modest increase in mitochondrial turnover between young and adult mice (Fig. 2A,B; $P < 0.05$). Interestingly, mitophagy levels in the geriatric PFC were comparable to those observed in young animals (Fig. 2A,B; $P > 0.05$). In contrast, analysis of the *auto*-QC PFC revealed peak autophagy levels in young animals. In the mature PFC, autophagy decreased significantly by 40% compared to young mice (Fig. 2G,H; $P < 0.0001$) and stabilized thereafter for the remainder of their lifespan (Fig. 2G,H; $P > 0.05$).

We next examined mitophagy in the hippocampal dentate gyrus (DG), which supports spatial information processing and memory and is known to be affected in neurodegenerative dementias (Hainmueller and Bartos, 2020). Strikingly, mitophagy levels in the DG increased with aging, peaking at middle age (Fig. 2C,D; $P < 0.0001$), but then sharply declined in the geriatric DG, comparable to levels in young mice (Fig. 2C,D; $P < 0.05$). Similar to the PFC, autophagy levels in the DG decreased steadily throughout aging (Fig. 2I,J; $P < 0.001$).

Next, we analyzed the cerebellum (CB), which coordinates locomotion, influences nonmotor behaviors and is considered to be particularly resilient in neuropsychiatric and neurodegenerative disorders (Liang and Carlson, 2020; Sathyanesan et al, 2019). Aging-related changes in functional circuitry between the CB and cortical regions underlie decreased cognitive and motor performance in older adults (Bernard and Seidler, 2014). In comparison to the PFC and DG, mitophagy levels in the CB followed a consistent upward trajectory, with the highest levels observed in

middle aged and geriatric tissues (Fig. 2E,F; $P < 0.0001$). Conversely, the CB emerged as one of the most stable regions for macroautophagy in our analyses, with levels unchanged across the mouse lifespan (Fig. 2K,l; $P > 0.05$).

These results demonstrate that physiological mitochondrial turnover occurs robustly throughout natural aging in the mammalian CNS, challenging the assumption that widespread decreases in mitophagy and autophagy are a universal consequence of mammalian aging. Importantly, our results reveal that the mammalian brain exhibits differential region-specific dynamics in autophagy pathways throughout life that we posit may contribute to the varying susceptibility of particular brain regions to degenerative processes.

## Selective and spatial modulation of mitophagy in aging Purkinje neurons

While our regional profiling revealed persistent mitophagy in the CB throughout the aging process, we next sought to unravel the cell type-specific interplay between autophagy and mitophagy. Purkinje cells (PCs) (Fig. 3A) and granule cells (GCs) comprise the fundamental microcircuitry of the CB. PCs modulate core processing and output of the cerebellar cortex and play crucial roles in motor control, learning, cognition, and emotion (Cerminara et al, 2015). Notably, PCs have an expansive morphology with high metabolic demands and heightened vulnerability to mitochondrial dysfunction and defective autophagy (Chen et al, 2007; Hara et al, 2006; Komatsu et al, 2006). Clinical phenotypes associated with congenital deficiencies in autophagy genes *ATG7* (Collier et al, 2021a), *ATG5* (Kim et al, 2016) and *SQSTM 1* (Haack et al, 2016) further support the selective vulnerability of PCs, particularly in gait ataxias. Prior work also established robust levels of basal mitophagy within PCs in vivo in adult mice, but its longitudinal regulation in this neural subset has not been investigated (McWilliams et al, 2016; Singh et al, 2021). Cerebellar granule cells (GCs), on the other hand, are the excitatory neuronal population that integrates sensorimotor information from precerebellar regions, transmitting it to PC dendrites. GCs are the smallest yet most abundant neuronal subtype in the mammalian CNS.

Our analysis revealed a robust increase in mitolysosome abundance with age in calbindin-reactive PCs, with a significant 2.3-fold increase observed in geriatric compared to young animals

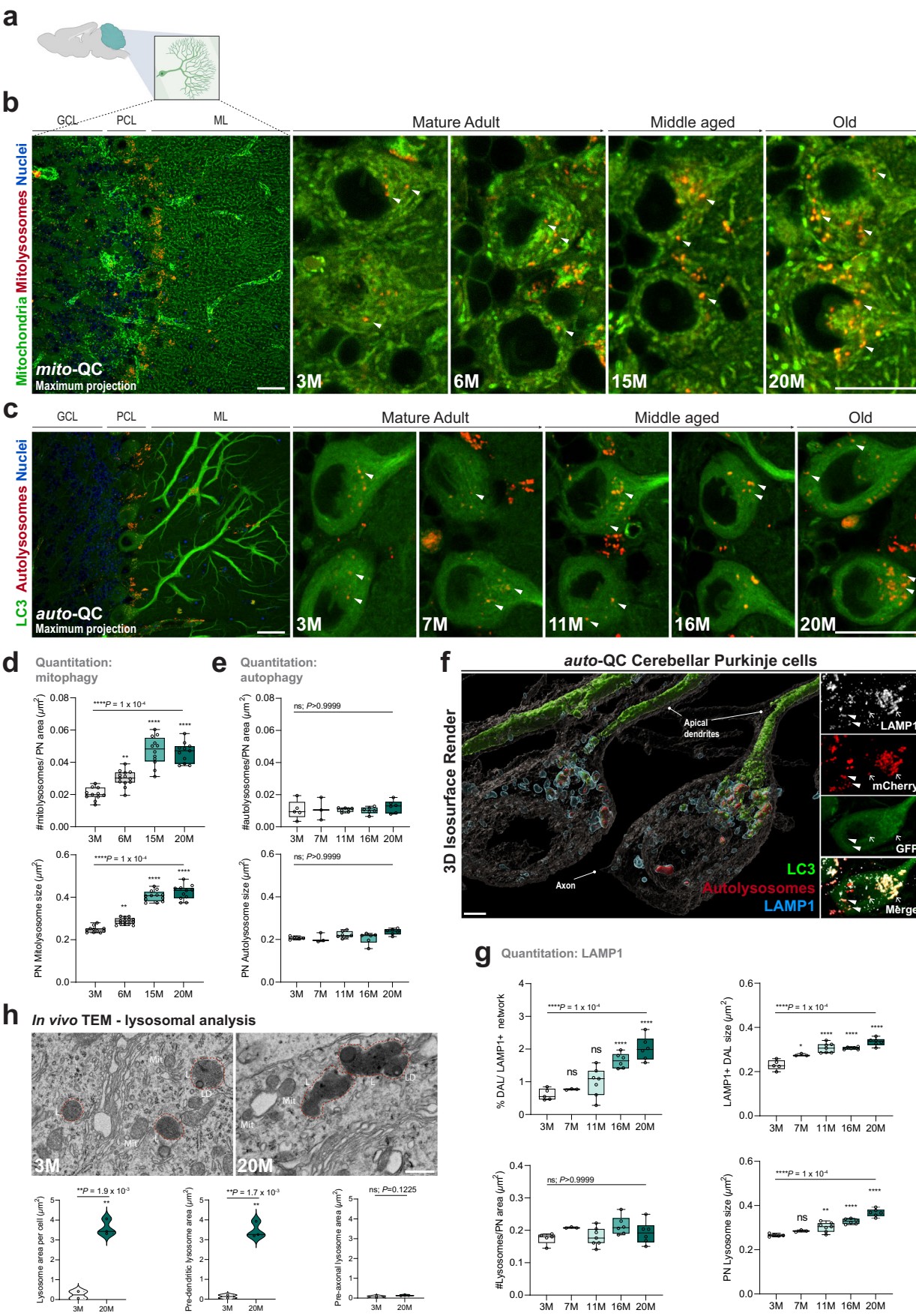

◄ **Figure 3.  Increased levels of neuronal mitophagy and differential lysosomal acidification define cerebellar aging in vivo.**

(A) Schematic of a parasagittal brain section highlighting the cerebellum and Purkinje cells analyzed. (B) Mitophagy profiling in Purkinje neurons. Confocal photomicrographs from *mito*-QC brain sections showing representative instances of mitophagy throughout aging in cerebellar Purkinje neurons. Arrowheads indicate events of mitophagy. Scale bar = 20 μm. GCL granule cell layer, PCL Purkinje cell layer, ML molecular layer. (C) Macroautophagy profiling in Purkinje neurons. Confocal photomicrographs from *auto*-QC brain sections showing representative instances of nonselective macroautophagy throughout aging in cerebellar Purkinje neurons. Arrowheads indicate events of autophagy. Scale bar = 20 μm. (D) Quantitative analysis of Purkinje cells in *mito*-QC mice reveals a robust age-dependent increase in the number of acidified mitolysosomes in vivo, in addition to increases in mean mitolysosome size throughout time. One-way ANOVA with Bonferroni post hoc. ****$P < 0.0001$. $n = 47$. (E) Quantitative analysis of Purkinje cells in *auto*-QC mice demonstrates the numbers of acidified autolysosomes remain constant throughout aging and do not change in size. One-way ANOVA with Bonferroni post hoc. ns = not significant; $P = > 0.9999$. $n = 27$. (F) Identification of differentially acidified autolysosomes formed throughout aging within Purkinje cells in vivo. Immunohistochemical analysis of endolysosomal network using LAMP1 antibody identifies distinct lysosomal subpopulations. Shown is a 3D isosurface render generated using IMARIS, alongside confocal photomicrograph insets. Arrowheads indicate LAMP1-mCherry-positive (GFP-negative) colocalization, arrows indicate LAMP1-mCherry-GFP colocalization. Lysosomes are predominantly enriched at the pre-dendritic zones. Scale bar = 5 μm. (G) Quantitative analysis reveals an age-dependent increase in the number and mean size of differentially acidified lysosomes (LAMP1-mCherry-GFP-positive structures) over time. Although the overall number of LAMP1-positive endolysosomal structures remains constant throughout mammalian life, this compartment expands in size as a function of age. One-way ANOVA with Bonferroni post hoc. ns = not significant; $P > 0.9999$, ***$P = 0.0002$ and ****$P = 0.0001$. $n = 27$. (H) In vivo transmission electron microscopy (TEM) imaging of young and geriatric *mito*-QC mice. Scale bar = 1 μm. Quantitation of TEM images reveal an age-dependent increase in lysosomal (L) area, consistent with LAMP1 data in (G), with further TEM analysis demonstrating selective alterations in lysosomal area occurs at the somatodendritic zones of Purkinje neurons, while no change occurs at the axonal aspect. Mitochondria (Mit), Lipid droplet (LD). Student's unpaired two-tailed $t$ test. **$P = 0.0019$ and 0.0017; ns = not significant; $P = 0.1225$. $n = 5$. Box plots extend from the 25th to the 75th percentiles, with a median line positioned inside the box. Whiskers denote the minimum and maximum values. Source data are available online for this figure.

(Fig. 3B,D; $P < 0.0001$). In addition, the mean size of PC mitolysosomes was significantly increased in geriatric animals compared to their younger counterparts, reflecting either increased autophagic occupancy or an age-dependent adaptation in the PC endolysosomal system (Fig. 3D; $P < 0.0001$). These alterations were also evident from morphometric profiling of PC mitolysosomes (Fig. EV3G; $P < 0.0001$), which further revealed age-dependent changes in morphology (Fig. EV4G; $P = 0.0047$).

To determine whether the increased levels of PC mitophagy are attributable to increased macroautophagy, we quantified general autophagy levels in vivo using the *auto*-QC model. We found that PCs maintained stable, unaltered levels of acidified autolysosomes throughout aging, with no change in autolysosome size (Fig. 3C,E; $P > 0.999$). Similar to geriatric PCs, excitatory cerebellar GCs exhibited increased mitophagy levels and mitolysosome size during aging (Fig. EV5C–E, $P < 0.0001$), while the levels of autophagy and autolysosome size remained unchanged (Fig. EV5F–H; $P > 0.999$). These results indicate that both of these key CB cell types exhibit a selective increase in mitophagic capacity independent of any differential regulation of autophagy in general.

LAMP1 and LAMP2 immunostaining confirmed the lysosomal nature of the acidified signals from both reporters in vivo, consistent with previous validations of the *mito*-QC reporter (Mito et al, 2022; McWilliams et al, 2016, 2018a; Lee et al, 2018; Singh et al, 2021; Allen et al, 2013) (Appendix Fig. S1). Autolysosomes also stained positively for Cathepsin B, Cathepsin D and TMEM55B, as expected (Appendix Fig. S1c–f). Intriguingly, while mitophagy has been heavily studied in neuronal axons and cell bodies, our findings indicate a distinct somatodendritic allocation of PC mitolysosomes within neuronal somata, demonstrated by their proximity relative to the axon initial segment (AIS) and confirmed by transmission electron microscopy (TEM) (Appendix Fig. S2a,b). In the *auto*-QC mice, PCs were among the most distinctive neuronal population in the adult CNS, hallmarked by a striking spatial enrichment of LC3 in the complex dendritic arbors of PC cells (Fig. 3C). We speculate that this conspicuous concentration of LC3 within PC dendrites may suggest a potential role for autophagy in synaptic plasticity or may hint at autophagy-

independent functions of ATG8 proteins in this distinct neuronal subcompartment.

## The neuronal endolysosomal network is remodeled throughout healthy aging

In addition to mCherry-only acidified structures, we consistently observed the presence of puncta that were mCherry-GFP 'double-positive' in geriatric *auto*-QC PCs. This indicates autophagosomal structures that have fused with lysosomes, but are not sufficiently acidic to quench GFP fluorescence (hereafter designated differentially acidified lysosomes or DALs). We verified that these mCherry-GFP 'double-positive' puncta were also LAMP1-positive (Fig. 3F). Although the levels of mCherry-only (acidified) autolysosomes did not change throughout aging (Fig. EV5A,B; $P > 0.05$), quantification of these DALs (LAMP1-mCherry-GFP triple-positive) revealed an accumulation starting in middle age, with the highest levels observed in geriatric PCs (Fig. 3G; $P < 0.0001$). Geriatric LAMP1-mCherry-GFP-positive structures were also larger in size than those in younger animals.

Further quantitative analysis of the PC endolysosomal network by TEM revealed an age-dependent increase in lysosomal area at the somatodendritic aspect (Fig. 3H; $P = 0.0017$), while no change was detectable at the pre-axonal aspect (Fig. 3H; $P = 0.1225$).

Collectively, these observations suggest that in these superaged PCs, mitochondria are selectively captured and efficiently trafficked through the autophagic pathway, but that a subset of lysosomes becomes differentially acidified, resulting in an accumulation of undegraded autolysosome-bound cellular contents with unusually long half-lives in geriatric and superaged animals.

## Subregional analysis of mitophagy and macroautophagy in the hippocampus

We next profiled autophagy pathways in the hippocampal formation, a brain region crucial for learning and memory. Autophagy has been studied extensively in context of hippocampal neuropathology, yet the spatiotemporal dynamics of physiological

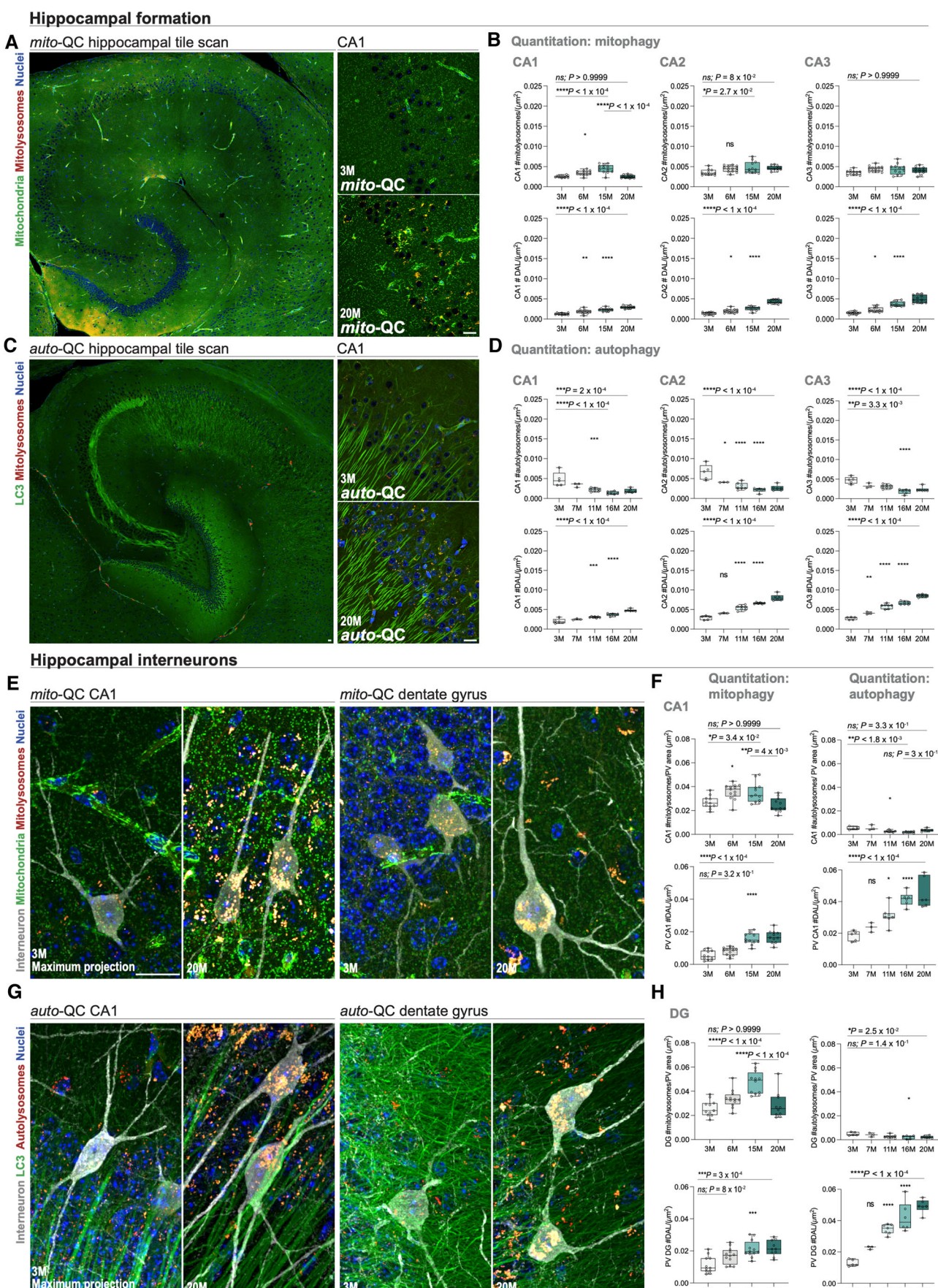

**Figure 4.    Spatiotemporal profiling of turnover pathways in aging mammalian hippocampal formation with emphasis on interneuronal subpopulations.**

(A) Confocal tile scan of the hippocampal formation from the *mito*-QC mouse. Representative images from young and geriatric CA1 region. Scale bars = 20 µm. (B) Mitophagy in aging hippocampal formation. Quantitative analysis reveals a significant increase in CA1 mitophagy up until 15 months (****$P$ value 0.0001), before a robust drop at geriatric age, returning to similar levels as in their young counterparts (ns $P$ value 0.9999). The CA1 undergoes a significant increase in the number of differentially acidified lysosomes as a function of age (****$P$ value 0.0001). $n = 46$. Quantitative analysis reveals no significant age-dependent changes in CA2 mitophagy, although a modest increase can be observed at 15-months (*$P$ value 0.0272, ns = not significant; $P$ value 0.0799). A robust increase in differentially acidified lysosomes can be observed in the CA2 as a function of age (****$P$ value 0.0001). $n = 46$. Quantitative analysis reveals no significant changes in CA3 mitophagy throughout aging (ns $P$ value 0.9999). A robust age-dependent increase in differentially acidified lysosomes can be observed in the CA3 (****$P$ value 0.0001). $n = 46$. (C) Confocal tile scan of the hippocampal formation from the *auto*-QC mouse. Representative images from young and geriatric CA1 region. Scale bars = 20 µm. (D) Macroautophagy in the aging hippocampal formation. Quantitative analysis reveals a significant decline in CA1 macroautophagy as a function of age (****$P$ value 0.0001, ***$P$ value 0.0002). The CA1 also exhibits a significant age-dependent increase in the number of differentially acidified lysosomes (****$P$ value 0.0001). $n = 27$. Quantitative analysis reveals that CA2 undergoes a robust decline in macroautophagy during aging (****$P$ value 0.0001), with a simultaneous increase in differentially acidified lysosomes (****$P$ value 0.0001). $n = 27$. Quantitative analysis reveals a significant age-dependent decline in CA3 macroautophagy (****$P$ value 0.0001, **$P$ value 0.0033). A significant increase in CA3 number of differentially acidified lysosomes can be seen as a function of age (****$P$ value 0.0001). $n = 27$. (E) Representative maximum projections of mitophagy events in young and geriatric CA1 and dentate gyrus parvalbumin interneurons. Scale bar = 20 µm. (F) Mitophagy in CA1 parvalbumin interneurons. Quantitative analysis reveals no significant difference in young and geriatric CA1 interneuron mitophagy, although a small increase in levels can be seen through adulthood and mid-life (*$P$ value 0.0335, ns = not significant; $P$ value 0.9999). The CA1 interneurons undergo a robust increase in the number of differentially acidified lysosomes as a function of age (****$P$ value 0.0001). $n = 47$. Mitophagy in dentate gyrus parvalbumin interneurons. Quantitative analysis reveals a robust increase in DG interneuron mitophagy up until 15-months before a substantial drop back to non-significant difference levels in geriatric mice compared to young (****$P$ value 0.0001, ns = not significant; $P$ value 0.9999). Dentate gyrus interneurons show a significant age-dependent increase in the number of differentially acidified lysosomes (***$P$ value 0.0003). $n = 46$. (G) Representative maximum projections of macroautophagy events in young and geriatric CA1 and dentate gyrus parvalbumin interneurons. Scale bar = 20 µm. (H) Macroautophagy in CA1 parvalbumin interneurons. Quantitative analysis reveals a significant decline in CA1 interneuron macroautophagy thoughout adulthood and mid-life before at geriatric age returning to similar levels as their young counterparts (**$P = 0.0018$; ns = not significant; $P = 0.3266$). CA1 interneurons exhibit a robust increase in the number of differentially acidified lysosomes in geriatric mice compared to young (****$P < 0.0001$). $n = 27$. Macroautophagy in dentate gyrus parvalbumin interneurons. Quantitative analysis reveals a modest decline in dentate gyrus interneuron macroautophagy in geriatric mice compared to young (*$P = 0.0252$). Dentate gyrus interneurons exhibit a robust increase in the number of differentially acidified lysosomes in geriatric mice compared to young. (****$P < 0.0001$). $n = 27$. Box plots extend from the 25th to the 75th percentiles, with a median line positioned inside the box. Whiskers denote the minimum and maximum values. Source data are available online for this figure.

autophagy during healthy aging remain largely unexplored. We quantified the abundance of acidified mitolysosomes and autolysosomes in *mito*-QC and *auto*-QC brain sections, respectively (Fig. 4A,C), and quantified DAL abundance to assess lysosomal homeostasis throughout life.

In mitophagy reporter mice, we observed a complex age-related trend in the CA1 region. Here, mitophagy increases throughout life, before a significant and sharp decline in old age (Fig. 4B; $P < 0.0001$). While acidified mitolysosomes increase in abundance with aging, this increase is not statistically significant between the youngest (3 m) and geriatric (20–26 m) animals. In contrast, the CA2 and CA3 subregions displayed relatively stable mitophagy levels across all age groups (Fig. 4B; ns; $P > 0.05$). Concurrently, the abundance of DALs increased significantly with age in all hippocampal regions assessed (Fig. 4B; $P < 0.0001$), suggesting a progressive loss of lysosomal homeostasis in the hippocampus with aging.

In macroautophagy reporter mice, acidified autolysosomes decreased consistently with age in CA1, CA2, and CA3 regions. The youngest animals had consistently higher levels of autolysosomes compared to their elder counterparts (Fig. 4D; $P < 0.0001$). No significant differences were found between middle aged and geriatric groups. This decline in acidified autolysosomes suggests a deterioration of clearance mechanisms in the aging hippocampus. Conversely, DALs increased significantly with age in all regions. The middle aged and geriatric age groups exhibited higher levels of DALs compared to young animals (Fig. 4D; $P < 0.0001$), reinforcing the notion of age-related impairments in lysosomal function.

These integrated findings from both mitophagy and macroautophagy reporter models reveal a dynamic and age-related landscape of autophagic turnover in the hippocampus. The initial increase in mitophagic activity suggests a compensatory upregulation of selective autophagy in early aging, possibly to neutralize

accumulating cellular damage, or as an adaptive response to sustain high homeostatic demands. Clearance mechanisms appear to become exhausted over time, leading to a marked decline in both mitophagy and macroautophagy at advanced ages. The reciprocal increase in the abundance of DALs indicates a reduced capacity for effective degradation and potential autophagic failure in the aging hippocampus. Our longitudinal data suggests midlife is a critical period for autophagy dynamics and may be an important inflection point for hippocampal integrity during healthy aging.

## Characterization of mitophagy and macroautophagy within inhibitory interneurons

Autophagy studies in the hippocampus have traditionally focused on pyramidal neurons. However, recent findings underscore the crucial contributions of gamma-aminobutyric acid-expressing (GABAergic) inhibitory interneurons in the microcircuits and networks that modulate memory encoding and consolidation (Tzilivaki et al, 2023). These inhibitory neurons have substantial energy requirements (Hu and Vervaeke, 2018; Gulyás et al, 2006) and their dysfunction has been implicated in epilepsy as well as neurodevelopmental and neurodegenerative disorders. The dynamics of physiological autophagy within interneurons remains poorly defined.

We profiled the well-characterized parvalbumin-positive (PV⁺) inhibitory interneurons (PV-INs) of the hippocampal CA1 region (Fig. 4E,G). PV-INs exhibit remarkable morphological diversity (Tzilivaki et al, 2023), and broadly comprise basket cell and axo-axonic cell subtypes. Strikingly, we observed robust levels of mitophagy and autophagy in CA1 PV-INs, as evidenced by stable levels of acidification of both mitolysosomes and autolysosomes throughout life (Fig. 4F; ns; $P > 0.05$). We contrasted these observations against hippocampal DG PV-INs (Fig. 4E,G), which

**A**  Midbrain dopaminergic circuits

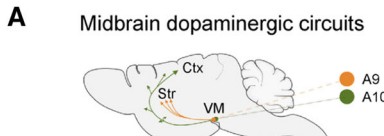

**B**  *mito*-QC Substantia nigra (A9)

Mature Adult → Middle aged → Old →

Isosurface render A9

3M | 6M | 15M | 20M

**C**  Quantitation: mitophagy

**P* = 1.4 × 10⁻³

#mitolysosomes/A9 area (μm²)

3M  6M  15M  20M

**D**  *auto*-QC Substantia nigra (A9)

Mature Adult → Middle aged → Old →

Maximum projection

3M | 7M | 11M | 16M | 20M

**E**  Quantitation: autophagy

ns; *P* > 0.9999

#autolysosomes/A9 area (μm²)

3M  7M  11M  16M  20M

**F**  *mito*-QC VTA (A10)

Mature Adult → Middle aged → Old →

Maximum projection

3M | 6M | 15M | 20M

**G**  Quantitation: mitophagy

ns; *P* > 0.9999
*P* = 2.4 × 10⁻²

#mitolysosomes/A10 area (μm²)

3M  6M  15M  20M

**H**  *auto*-QC VTA(A10)

Mature Adult → Middle aged → Old →

Isosurface render A10

3M | 7M | 11M | 16M | 20M

**I**  Quantitation: autophagy

**P* = 7.6 × 10⁻³

#autolysosomes/A10 area (μm²)

3M  7M  11M  16M  20M

**Figure 5.    Temporal profiling of turnover pathways in aging mammalian dopaminergic circuits in vivo.**

(A) Simplified schematic of A9 and A10 DA circuits. Both A9 and A10 neurons reside in the ventral mesencephalon (VM) and have distinct targets within the brain. A9 DA neurons project to the dorsolateral striatum, forming the nigrostriatal pathway that modulates voluntary locomotion, whereas A10 DA neurons project to the ventromedial striatum and forebrain to control reward, motivation, and aversion. (B, C) Mitophagy in aging A9 DA neurons. 3D Isosurface renders of young and geriatric *mito*-QC A9 dopaminergic neurons (scale bar = 5 μm) and representative confocal images of mitophagy events throughout aging, scale bar = 20 μm. Quantitative analysis reveals increased mitophagy levels as a function of age. One-way ANOVA with Bonferroni post hoc. \*\*$P = 0.0014$. $n = 43$. (D, E) Macroautophagy in aging A9 DA neurons. Representative maximum projections of autophagy events throughout aging in A9 dopaminergic neurons (scale bar = 20 μm). Quantitative analysis of autophagy in A9 dopaminergic neurons shows no age-dependent alterations in autophagy levels. One-way ANOVA with Bonferroni post hoc. ns = not significant; $P > 0.9999$. $n = 19$. (F, G) Mitophagy in aging A10 DA neurons. Representative maximum projections of mitophagy events throughout aging in A10 dopaminergic neurons (scale bars = 20 μm). Quantitative analysis of mitophagy in A10 dopaminergic neurons reveals no change between young and geriatric mice. Some fluctuations in mitophagy levels were detected with a modest increase at 15 months. One-way ANOVA with Bonferroni post hoc. ns = not significant; $P > 0.9999$, \*$P = 0.024$. $n = 44$. (H, I) Macroautophagy in aging A10 DA neurons. Isosurface renders of young and geriatric *auto*-QC A10 dopaminergic neurons (scale bar = 5 μm). Representative images of autophagy events throughout aging in A10 dopaminergic neurons (scale bar = 20 μm). Quantitative analysis of autophagy in A10 dopaminergic neurons reveals decreased levels of autophagy as a function of age. One-way ANOVA with Bonferroni post hoc. \*\*$P = 0.0076$. $n = 26$. Box plots extend from the 25th to the 75th percentiles, with a median line positioned inside the box. Whiskers denote the minimum and maximum values. Source data are available online for this figure.

exhibited identical trends. However, autophagy modulation in DG interneurons appears to be a dynamic process, with diminished mitophagy levels observed in super-aged mice compared to mid-life (Fig. 4H; $P < 0.0001$) and a modest decline in macroautophagy (Fig. 4H; $P = 0.0252$). Like cerebellar Purkinje cells, we observed a progressive accumulation of differentially acidified lysosomes in both CA1 and DG PV$^+$ interneurons (Fig. 4F,H). Our in vivo data indicates that basal autophagy is a characteristic feature of mammalian interneurons, potentially contributing to their homeostatic integrity throughout life.

## Differential regulation of autophagy pathways in aging dopaminergic circuits

There is mounting evidence that either basal or mitochondrial stress-induced mitophagy may be impaired in neurodegenerative disease such as Parkinson's disease (PD) (Ganley, 2022). PD is an age-dependent neurodegenerative disorder characterized by the selective degeneration of the highly arboreous dopamine (DA)-producing A9 nigrostriatal neurons (Matsuda et al, 2009) (Fig. 5A) that leads to progressive loss of voluntary motor control. Previous studies demonstrated robust levels of basal mitophagy—which occurs independently of the PINK1-Parkin pathway (McWilliams et al, 2018a, 2018b; Lee et al, 2018; Wrighton et al, 2021) and is influenced by LRRK2 kinase activity (Singh et al, 2021)—in the A9 DA neurons of young animals. Here, we asked whether an age-dependent reduction in basal mitophagy in A9 DA neurons could provide a compelling explanation for the age-related risk of PD pathology. Surprisingly, analysis of tyrosine hydroxylase (TH)-positive A9 neurons in the aging *mito*-QC brain revealed a gradual increase in mitophagy throughout aging in vivo, with mitochondrial turnover modestly higher in healthy geriatric animals than in their young counterparts (Fig. 5B,C; $P = 0.0014$). Conversely, autophagy levels were stable across aging in A9 DA neurons (Fig. 5D,E; $P > 0.9999$), with similar alterations to the size and morphology of autolysosomes as observed in PCs (Appendix Fig. S3v; $P > 0.9999$ and Appendix Fig. S3h; $P = 0.1265$). Importantly, these findings clearly show that basal levels of selective mitophagy and generalized autophagy appear to be sustained in a PD-susceptible neuronal population during healthy aging.

Unlike A9 neurons, preclinical and human postmortem studies suggest that neighboring A10 midbrain DA neurons in the ventral tegmental area (VTA) (Fig. 5A) exhibit greater resilience to PD

pathology (Björklund and Dunnett, 2007). A10 neurons comprise a functionally distinct DA subclass that makes an enormous contribution to behavior through modulation of reward, motivation, and aversion (Garritsen et al, 2023), and VTA dysfunction is associated with addiction and schizophrenia. Here, we examined A10 DA neurons to determine whether we could detect differences in basal mitophagy that could account for the difference in susceptibility to PD pathology. Intriguingly, our findings revealed a distinct profile of mitophagy in A10 DA neurons, with a modest increase at middle age before returning to levels comparable to early life, which persisted throughout geriatric age (Fig. 5F,G; $P = 0.024$, and $P > 0.9999$). Basal mitophagy levels appear relatively similar between the A9 DA neurons (mean=0.005 mitolysosomes/μm$^2$ for young animals; mean = 0.018 in geriatric) and A10 DA neurons (mean = 0.007 mitolysosomes/μm$^2$ for young mice; mean = 0.008 in geriatric). In contrast, macroautophagy levels declined with age in A10 DA neurons (Fig. 5H,I; $P = 0.0076$). Comparable to mitophagy, basal macroautophagy levels appear relatively similar in A9 (mean=0.003 autolysosomes/μm$^2$ for young animals; mean = 0.002 in geriatric) and A10 DA neurons (mean = 0.005 autolysosomes/μm$^2$ for young mice; mean=0.002 in geriatric). Moreover, although we observed a significant decline in A10 DA neurons (Fig. 5I; $P = 0.0076$), macroautophagy persisted, albeit at a reduced level. These findings provide compelling evidence that mitochondrial networks are selectively altered by autophagy throughout the natural aging process in a cell type- and region-specific manner. Furthermore, our data may explain how altered trajectories of mitochondrial turnover could promote pathology in specific contexts of PD. For instance, the pathogenic kinase activity of the LRRK2$^{G2019S}$ mutant protein significantly suppresses basal mitophagy levels in DA neurons (Singh et al, 2021). Given that our findings reveal increased mitophagy in A9 DA neurons throughout healthy aging, it is plausible that dysregulated mitophagy dynamics contribute to the vulnerability of these neurons in PD. The contrasting temporal patterns of autophagy and differences in cargo captured within A9 and A10 cellular subtypes may explain aspects of both resilience and vulnerability of specific DA populations during disease.

## Regulation of autophagy pathways in distinct astroglial niches

To extend our understanding of cell-specific autophagy dynamics in the aging brain, we next moved beyond neuronal populations to examine astroglia. Astroglia play crucial roles in adaptive plasticity,

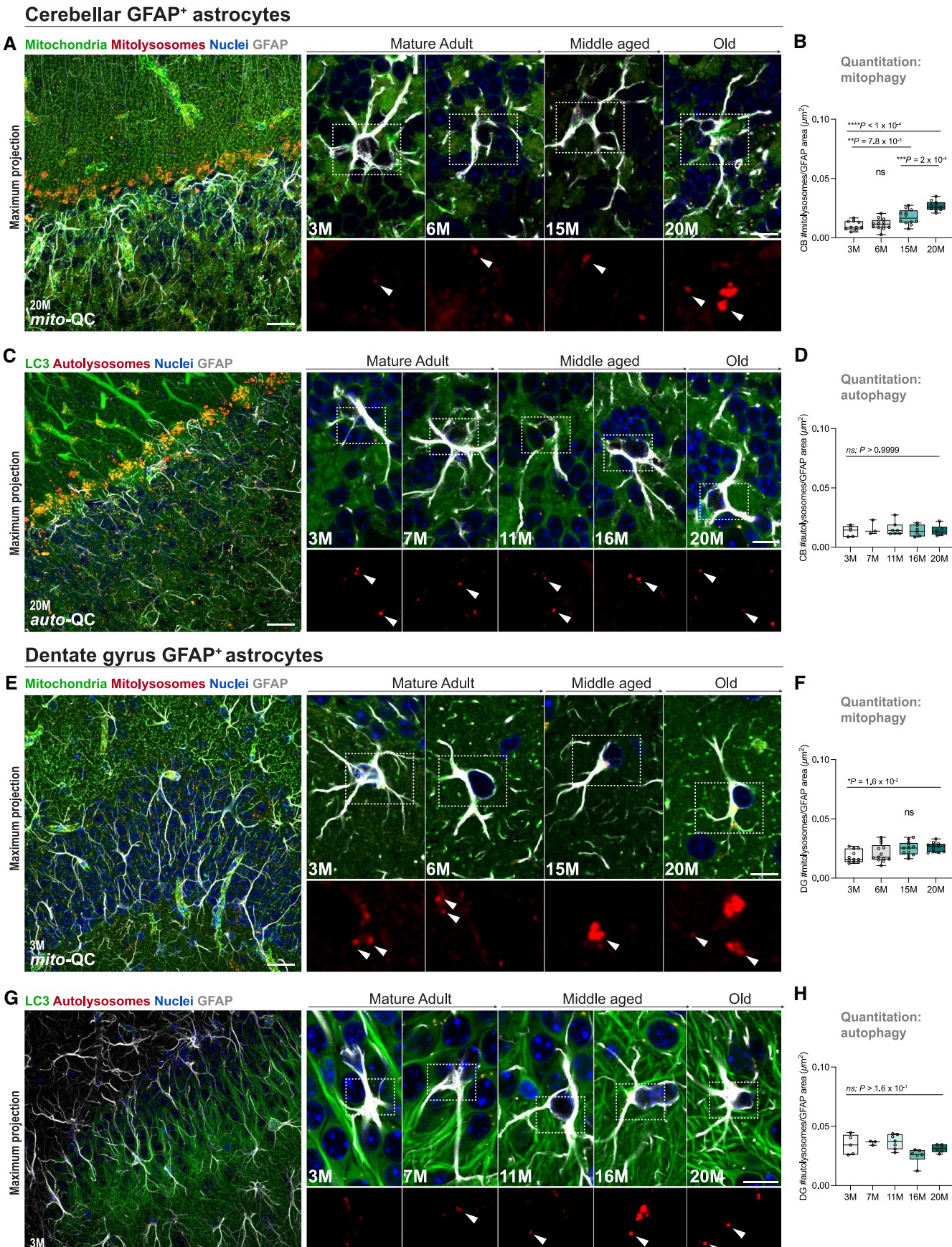

**Figure 6.   Temporal analysis of mitophagy and macroautophagy pathways in two anatomically and functionally distinct astroglial niches.**

(A, B) Cerebellar astrocyte mitophagy. Representative images of mitophagy events throughout aging in cerebellar astrocytes in *mito*-QC mice (Overview, scale bar = 20 μm; close-up scale bar = 10 μm). Insets display mitolysosomes. Quantitative analysis reveals a robust increase in cerebellar astrocyte mitophagy levels as a function of age. Arrowheads indicate events of mitophagy. One-way ANOVA with Bonferroni post hoc. ****P < 0.0001, ***P < 0.0002, **P = 0.0078. n = 46. (C, D) Cerebellar astrocyte macroautophagy. Representative images of autophagy events throughout aging in cerebellar astrocytes in *auto*-QC mice (Overview, scale bar = 20 μm; close-up scale bar = 10 μm). Quantitative analysis shows no significant changes in cerebellar astrocyte autophagy levels during aging. Arrowheads indicate events of autophagy. One-way ANOVA with Bonferroni post hoc. ns = not significant; P > 0.9999. n = 27. (E, F) Dentate gyrus astrocyte mitophagy. Representative images of mitophagy events throughout aging in DG astrocytes in *mito*-QC mice (Overview, scale bar = 20 μm; close-up scale bar = 10 μm). Insets display mitolysosomes. Quantitative analysis reveals a modest increase in DG astrocyte mitophagy levels in geriatric mice compared to young. Arrowheads indicate events of mitophagy. One-way ANOVA with Bonferroni post hoc. *P = 0.0162. n = 47. (G, H) Dentate gyrus astrocyte macroautophagy. Representative images of autophagy events throughout aging in DG astrocytes in *auto*-QC mice (Overview, scale bar = 20 μm; close-up scale bar = 10 μm). Quantitative analysis shows no significant age-dependent changes in DG astrocyte autophagy levels during aging. Arrowheads indicate events of autophagy. One-way ANOVA with Bonferroni post hoc. ns = not significant; P = 0.1614. n = 27. Box plots extend from the 25th to the 75th percentiles, with a median line positioned inside the box. Whiskers denote the minimum and maximum values. Source data are available online for this figure.

synaptic function and responding to injury. We focused on two regionally distinct niches throughout life, the cerebellum and hippocampal DG to understand how autophagy is regulated in diverse neural environments as the brain ages.

In the cerebellum (Fig. 6A–D), astroglial mitophagy remained stable until middle age, where we observed a significant increase that persisted in older animals (Fig. 6A,B; P < 0.0001). In addition, we observed a corresponding increase in the size of mitolysosomes (Appendix Fig. S3y; P < 0.0001), suggesting age-related adaptive homeostasis in astrocyte endolysosomes. Notably, cerebellar GFAP+ astrocytes were the only cell subtype across all examined brain regions to show a sex-related difference in mitophagy, with higher levels in female astrocytes during mid-life compared to males (Appendix Fig. S4y; P = 0.0006). In contrast, analysis of macro-autophagy reporter mice revealed no significant differences in the abundance (Fig. 6B,C; ns; P > 0.05) or size (Appendix Fig. S3z; ns; P > 0.05) of acidified autolysosomes throughout life, nor between sexes (Appendix Fig. S4z; ns; P > 0.05). Dentate gyrus resident-astrocytes exhibited similar trends (Fig. 6E), however, mitophagy was only modestly increased (Fig. 6F; P = 0.0162) and macroautophagy remained stable throughout life (Fig. 6G,H; ns; P > 0.05).

We also detected a subset of differentially acidified lysosomes within astrocytes of both regional niches. However, unlike the DALs we observed earlier in neurons, these did not exhibit an age dependent increase throughout life (Appendix Fig. S5y–aa; ns; P > 0.05) except in the *auto*-QC dentate gyrus where astrocytes exhibit a robust age-dependent increase in DALs (Appendix Fig. S5ab; P < 0.0001). These results highlight distinct autophagy dynamics in astrocytes, underscoring key differences in endolyso-somal regulation compared to aging neurons.

## Autophagy dynamics in neuroimmune cells and tissues of the aging brain

Although neurodegenerative disease research has rightly focused on neurons and astrocytes, the important contributions of additional cell types are increasingly being recognized (Cathomas et al, 2022). Having profiled basal mitophagy and autophagy within different neuronal and astroglial populations in vivo during healthy aging, we next asked whether similar principles of sustained mitophagy apply to microglia, which play important roles in neurodegenerative disease and whose function appears to be specialized according to their different anatomical locations (Tan et al, 2020). Previous work established the presence of basal mitophagy in microglia

in vivo (McWilliams et al, 2018a; Singh et al, 2021), but the impact of aging on mitochondrial turnover or autophagy in this clinically important cell subtype remains unclear. Here, we identified microglia in the cerebellum of *mito*-QC and *auto*-QC mice using the marker Iba1 (ionized calcium-binding adapter molecule 1) (Fig. 7A,C). Cerebellar microglia exhibit a distinct aging trajectory from other microglial populations, becoming highly immunoreactive and susceptible to dysfunction (Stowell et al, 2018; Stoessel and Majewska, 2021). We found that cerebellar microglia exhibit a progressive age-dependent 5.4-fold enhancement of mitophagy throughout life (Fig. 7A,B; P < 0.0001), with an increased mitolysosomal profile in geriatric animals compared to their young counterparts (Appendix Fig. S3ac; P < 0.0001). Conversely, cerebellar microglia had a relatively constant rate of autophagy over the aging timecourse (Fig. 7C,D; P = 0.6325), although a modest decline was observed at middle age compared to their young counterparts (Fig. 7C,D; P = 0.0131). To investigate whether these patterns were specific to cerebellar microglia, we also profiled functionally distinct microglia in the DG. We found that DG microglia also showed increased mitophagy levels in geriatric animals (Fig. 7E,F; P < 0.0001) and stable levels of generalized autophagy (Fig. 7G,H; P = 0.999) across the mouse lifespan.

Another brain structure of particular interest to us is the choroid plexus (ChP), a fenestrated, cuboidal epithelial monolayer suspended in the brain ventricles that produces cerebrospinal fluid (CSF) (Praetorius and Damkier, 2017). The ChP is a mitochondria-rich tissue with high metabolic demand due to a lifetime of constitutive exocytosis of CSF (Cornford et al, 1997), which is vital for brain metabolic signaling and immune surveillance. ChP integrity is diminished by aging and disrupted in hydrocephalus and neurodegeneration (Preston, 2001; Mesquita et al, 2015). We detected a highly mitochondria-rich structure in the ventricular system of *mito*-QC brain sections, confirmed as the ChP based on its distinctive anatomical position and morphological features observed under light microscopy (Fig. 8A). We observed characteristic networks of spherical mitochondria in the caudal ChP of young *mito*-QC mice, accompanied by robust levels of mitophagy (Fig. 8B). Mitophagy levels increased steadily with age in the ChP (Fig. 8C; P = 0.0287), while mitolysosome size remained constant (Appendix Fig. S3ag; P > 0.05). In comparison to mitophagy levels, the number of acidified autolysosomes we observed in the *auto*-QC mice remained unaltered throughout aging (Fig. 8D,E; P > 0.9999). The baseline level of macroautophagy in the ChP was the highest across all CNS regions examined in this study (mean = 0.031

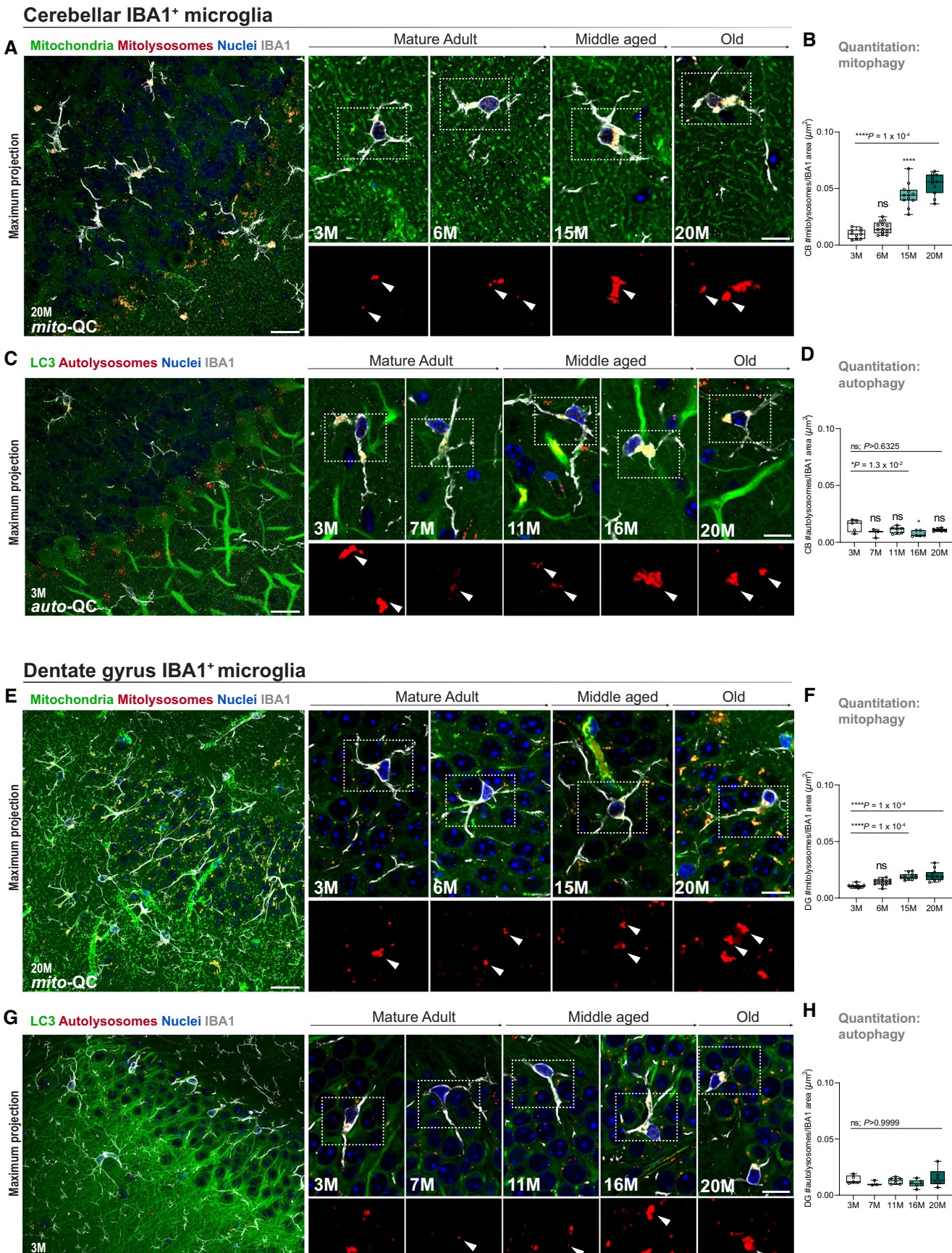

**Figure 7. Region-specific analysis of microglial mitophagy and macroautophagy pathways throughout healthy brain aging.**

(A, B) Cerebellar microglial mitophagy. Representative images of mitophagy events throughout aging in cerebellar microglia in *mito*-QC mice (Overview, scale bar = 20 μm; close-up scale bar = 10 μm). Insets display mitolysosomes. Quantitative analysis reveals a robust increase in cerebellar microglia mitophagy levels as a function of age. Arrowheads indicate events of mitophagy. One-way ANOVA with Bonferroni post hoc. ****$P < 0.0001$. $n = 46$. (C, D) Cerebellar microglial macroautophagy. Representative images of autophagy events throughout aging in cerebellar microglia in *auto*-QC mice (Overview, scale bar = 20 μm; close-up scale bar = 10 μm). Quantitative analysis shows no significant changes in cerebellar microglia autophagy levels during aging, although a modest fluctuation in autophagy was detected at 16 months. Arrowheads indicate events of autophagy. One-way ANOVA with Bonferroni post hoc. ns = not significant; $P = 0.6325$, *$P = 0.0131$. $n = 27$. (E, F) Microglial mitophagy in the dentate gyrus. Representative images of mitophagy events throughout aging in dentate gyrus microglia in *mito*-QC mice (Overview, scale bar = 20 μm; close-up scale bar = 10 μm). Quantitative analysis reveals an increase in dentate gyrus microglia mitophagy levels throughout aging. Arrowheads indicate events of mitophagy. One-way ANOVA with Bonferroni post hoc. ****$P < 0.0001$. $n = 47$. (G, H) Microglial macroautophagy in the dentate gyrus. Representative images of autophagy events throughout aging in dentate gyrus microglia in *auto*-QC mice (Overview, scale bar = 20 μm; close-up scale bar = 10 μm). Quantitative analysis shows no significant age-dependent changes in dentate gyrus microglia autophagy levels during aging. Arrowheads indicate events of autophagy. One-way ANOVA with Bonferroni post hoc. ns = not significant; $P > 0.9999$. $n = 27$. Box plots extend from the 25th to the 75th percentiles, with a median line positioned inside the box. Whiskers denote the minimum and maximum values. Source data are available online for this figure.

autolysosomes/μm² for young animals; mean = 0.027 autolysosomes/μm² in geriatric). Our collective findings demonstrate that non-neuronal cells and neuroimmune tissues of the aging brain undergo sustained selective and general autophagic turnover throughout life, with a notable increase in mitophagic capacity during healthy aging.

## Discussion

Autophagy is a disease-relevant homeostatic quality control mechanism. While we understand how different forms of cellular stress induce specific autophagy pathways in cultured cells, our knowledge of how physiological autophagy pathways are regulated in healthy brain aging is extremely limited. Studies in short-lived model systems suggest that diminished mitophagy and autophagic capacity may sensitize certain brain cell types to degenerative processes as we age. However, the spatiotemporal modulation of mitophagy during healthy brain aging remains unclear.

Here, we establish the first dynamic landscape of mitochondrial turnover in the intact, aging mammalian brain at the single-cell level using high-resolution confocal imaging and cutting-edge reporter mice. Our findings reveal that decreased mitophagy is not a general hallmark of healthy aging in vivo but that different brain regions and neural subsets exhibit distinct mitophagy dynamics over time, usually remaining stable or even increasing throughout the mouse lifespan. By comparing different regions of the brain, including disease-associated neuronal and non-neuronal cell types, we revealed uncoupled and cell type-specific regulation of mitophagy and generalized autophagy throughout natural aging (Fig. 9). While our article was under revision, Boya and colleagues published a study comparing mitophagy in young and old *mito*-QC mice, which supports our findings (Jiménez-Loygorri et al, 2024).

We found that mitophagy levels gradually increased throughout the aging process in several cell types, including cerebellar GCs and microglia, seemingly independent of basal autophagy levels. In some cases, we observed more complex trajectories: we detected an age-dependent increase in mitophagy in the hippocampus CA1 and DG subregions up until middle age, followed by a significant decline during old age, although not falling below those of young subjects. It will be crucial to determine whether these altered autophagy dynamics are causally linked to age-related cognitive changes observed in healthy aging (Salthouse, 2019). Clarifying the mechanisms driving age-dependent mitophagy dynamics in these

hippocampal subregions may hold interventional relevance for memory-related pathologies such as dementia and Alzheimer's disease (Jiang et al, 2022; Fang et al, 2019). In another example, comparing A9 nigrostriatal dopaminergic neurons that degenerate during PD and their neighboring unaffected A10 DA neurons, we showed an age-related increase in mitophagy in the A9 population —a finding that supports the hypothesis that age-related mitophagy dysregulation could sensitize certain neuronal cell types to degeneration, at least in healthy animals.

In contrast to rapidly aging model organisms with short generation times, long-lived mammalian neural circuits have evolved under entirely different selection pressures that may contribute to species-specific differences in mitophagy levels. Mitochondria are pleiotropic organelles required for diverse processes, supplying ATP for energetics and metabolites critical for epigenetic modifications (Martínez-Reyes and Chandel, 2020). The increased mitochondrial turnover we detected in distinct neuronal subsets may be a response to counteract the gradual burden of stress and the onset of neurological dysfunction that arises as we age. While we did not detect any widespread decline in mitophagy throughout the healthy aging process, impaired autophagy is known to compromise neural integrity and promote the selective vulnerability that hallmarks human neuropathology (Collier et al, 2021a; Hara et al, 2006; Komatsu et al, 2006; Kim et al, 2016), and it is plausible that disrupting the temporal dynamics of cell-specific mitochondrial quality control could do the same. In this regard, the transition from midlife to old age marks a critical inflection point for the regulation of autophagy pathways in the brain. These findings align with recent research suggesting that 'middle-age' represents a pivotal period in life, where the balance between degenerative and developmental processes shifts, significantly influencing neural integrity and cognitive health (Dohm-Hansen et al, 2024).

Strong links have emerged between impaired physiological mitophagy and neurodegeneration. Pathogenic hyperactivating mutations in LRRK2 (e.g., G2019S) promote human nigrostriatal degeneration, and the suppression of basal mitophagy independently of general autophagy appears to be a critical mechanism in this process (Singh et al, 2021). Further research will be crucial to identify cell- and tissue-specific effectors of mitophagy in vivo, considering the substantial differences observed between the regulation of physiological mitophagy and its pharmacological manipulation in cultured cells. Beyond quality control, the observed age dependent mitophagy alterations may reflect an

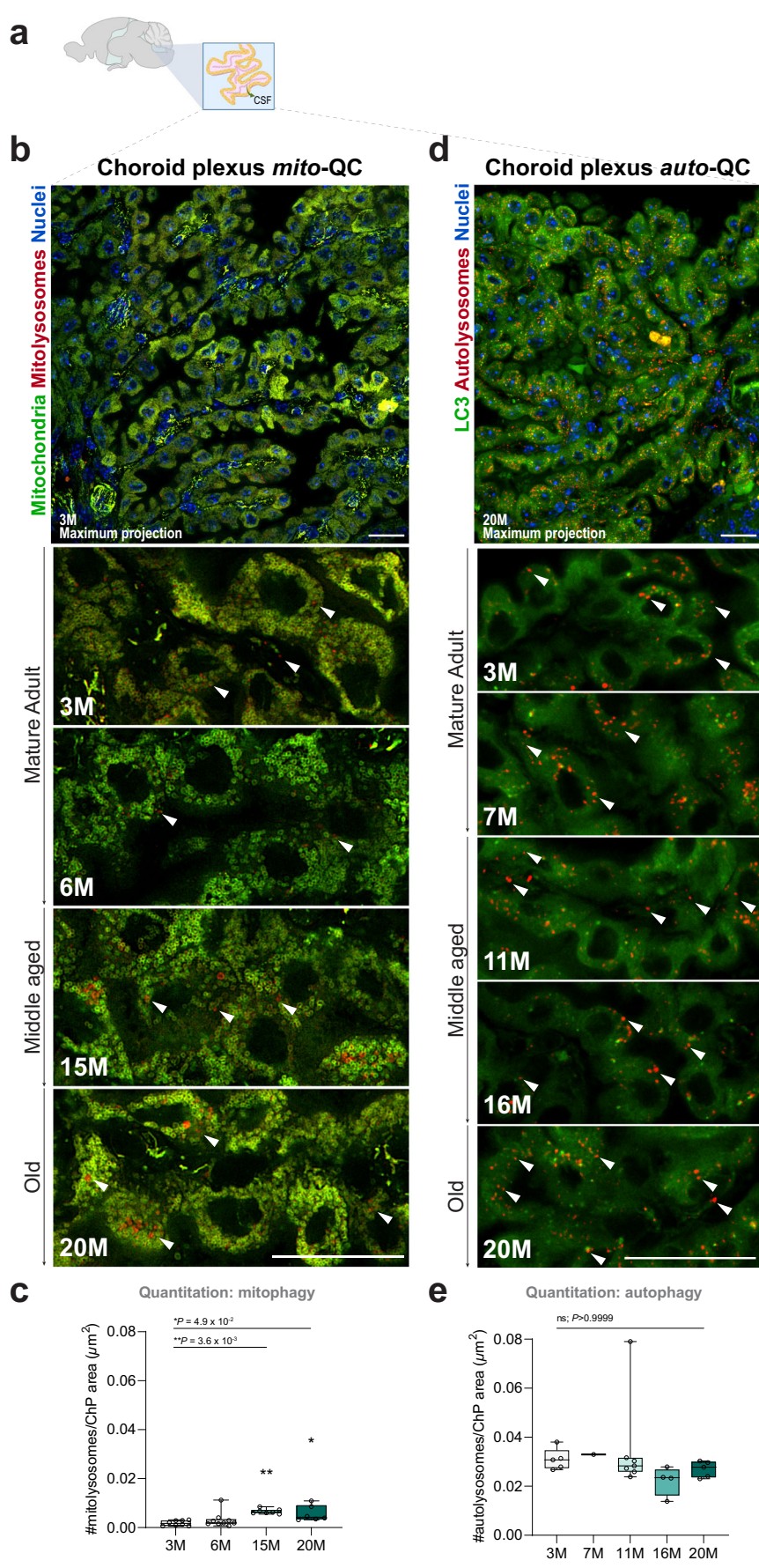

**Figure 8.  Aging modifies CSF-producing choroid plexus turnover.**

(A) Schematic of a parasagittal brain section highlighting the ChP within ventricle IV. (B, C) Mitophagy in the aging ChP. Representative images of mitophagy events throughout aging in the *mito*-QC choroid plexus. Quantitative analysis reveals increased levels of mitophagy in geriatric mice compared to young. Arrowheads indicate events of mitophagy. One-way ANOVA with Bonferroni post hoc. Scale bars = 20 µm. **$P$ = 0.0287, *$P$ = 0.045. $n$ = 30. (D, E) Macroautophagy in the aging ChP. Representative images of autophagy events throughout aging in the *auto*-QC choroid plexus. Quantitative analysis reveals no age-dependent changes in autophagy through time. Arrowheads indicate events of autophagy. Scale bars = 20 µm. ns = not significant; $P$ > 0.9999. $n$ = 22. Box plots extend from the 25th to the 75th percentiles, with a median line positioned inside the box. Whiskers denote the minimum and maximum values. Source data are available online for this figure.

adaptative response, adjusting neural circuit homeostasis to align with evolving metabolic demands.

Interestingly, our analysis also revealed age-dependent adaptations in neuronal lysosomes. In aged Purkinje cells, hippocampal neurons and inhibitory interneurons, we detected a mixture of acidified and differentially acidified LAMP1-positive autolysosomes. From this, we infer that a subset of lysosomes in aged neurons exhibit a defect in dynamics, fusion, or activity in vivo, despite apparently normal levels of autophagic sensing and cargo capture. Numerous factors, such as lipid metabolism, proteostasis, and cellular stress, affect lysosomal homeostasis, which in turn impacts the autophagy pathway (Udayar et al, 2022). Importantly, we cannot preclude that progressive lysosomal dysfunction may in some cases impair autophagic flux, which may somehow contribute to the accumulation of red-only puncta. Future work will determine the molecular identity of accumulating lysosomal cargo in aged Purkinje cells and the precise nature of the lysosomal phenotype. Defining the molecular profile and mechanistic origin of the differentially acidified LAMP1-positive puncta will be particularly important, as LAMP1 can demarcate a variety of endolysosomal structures (Cheng et al, 2018). However, similar reporter strategies to ours have been used by the Nixon group to demonstrate profound accumulation of poorly-acidified autolysosomes (pa-ALs) in brains of Alzheimer's disease preclinical models (Lee et al, 2022). In agreement with this, our findings support the idea that such pathomechanisms may be the animating force that compounds natural aging-associated lysosomal dysfunction, leading to selective neuropathology and disease. Our findings highlight the importance of integrating autophagy-enhancing therapeutic strategies with assessments of endolysosomal homeostasis to prevent cellular dysfunction and achieve effective neuroprotection. The abundance and dynamic allocation of lysosomal subsets constitute both a metabolic adaptation and a natural vulnerability that could be exploited by 'triggers, facilitators, and aggravators' to promote neurodegeneration (Johnson et al, 2019). It will therefore be important in future work to determine the mechanisms leading to differential lysosome acidification during healthy aging. Ultimately, developing strategies to preserve endolysosomal homeostasis in the aging brain may be crucial to safeguard autophagy dynamics and neural integrity.

Several reporter strategies have been developed to monitor mitophagy and mitochondrial homeostasis in vivo. A previous study using the mt-Keima reporter mouse concluded that brain mitophagy generally decreases with age, based on a pairwise comparison in the dentate gyrus (Sun et al, 2015). However, there are important considerations to the mt-Keima model that may confound the interpretation of these data. First, mt-Keima tissues require acute imaging after isolation since fixation abolishes its signal, hindering the spatially resolved, systematic, and longitudinal analyses required to understand cell type-specific and age-

dependent effects. Second, the mt-Keima mouse line was constructed on the FVB genetic background, which presents numerous pathological phenotypes, including natural neurodegeneration, that may influence mitophagy levels (Mahler et al, 1996; Schauwecker and Steward, 1997; Goelz et al, 1998; Chang et al, 2002; Pugh et al, 2004; Farley et al, 2011; Eltokhi et al, 2020). Importantly, mitochondrial fluorescence was almost undetectable in images of the aged mt-Keima dentate gyrus and hippocampus (Sun et al, 2015), despite this region being energetically demanding and mitochondria-rich in mice and humans (Kageyama and Wong-Riley, 1982; Gulyás et al, 2006; Sisková et al, 2010; Cottrell et al, 2001a, 2001b; Tyynismaa et al, 2005), which was also confirmed by our analysis of *mito*-QC brain sections. Given the inherent fluctuations observed in mitochondrial matrix pH (Chalmers and Nicholls, 2003; Ghafourifar and Richter, 1999; Pravdic et al, 2012; Wei-LaPierre et al, 2013; Hawrysh and Buck, 2019), ascertaining whether fluorescence-based alterations in mt-Keima are exclusive to mitophagy poses significant challenges. Considering these caveats with the mt-Keima model and our conflicting conclusions, we posit that the age-related decline in mitophagy observed in the previous study may be attributed to the severity of FVB mouse pathology rather than a physiologically relevant process in a typical aging trajectory. While our findings provide contrasting evidence, further investigation is necessary to fully understand the relationship between mitophagy dynamics and neuropathology and to differentiate whether mitophagy decline is a cause or consequence of neurodegeneration. It will also be intriguing to examine how longitudinal autophagy dynamics and lysosomal homeostasis are modified by emergent geroprotective and neuroprotective interventions. The integration of spatiotemporal data from optical reporter systems and intact tissue analysis, combined with advanced proteomics strategies (Kallergi et al, 2023; Goldsmith et al, 2022), holds substantial promise to significantly enhance our understanding of physiological autophagy in the mammalian brain.

In summary, our findings establish a resource that reveals dynamic changes in mammalian neural mitophagy between diverse neuronal and non-neuronal cell subsets. These findings demonstrate that selective autophagy in the mammalian brain is far more complex than previously appreciated and suggest that targeted, cell type-specific approaches to modulate mitophagy may have greater clinical benefit than attempts to achieve widespread elevations in mitochondrial turnover. The balance and synchrony between selective and nonselective autophagy pathways and lysosomal homeostasis may prove crucial to sustain mammalian neural integrity.

## Limitations of the study

In this study, we profiled the spatiotemporal dynamics of mitophagy and macroautophagy in different regions of the

                                                               

## Synopsis: Dynamic regulation of autophagy pathways in the aging mammalian brain

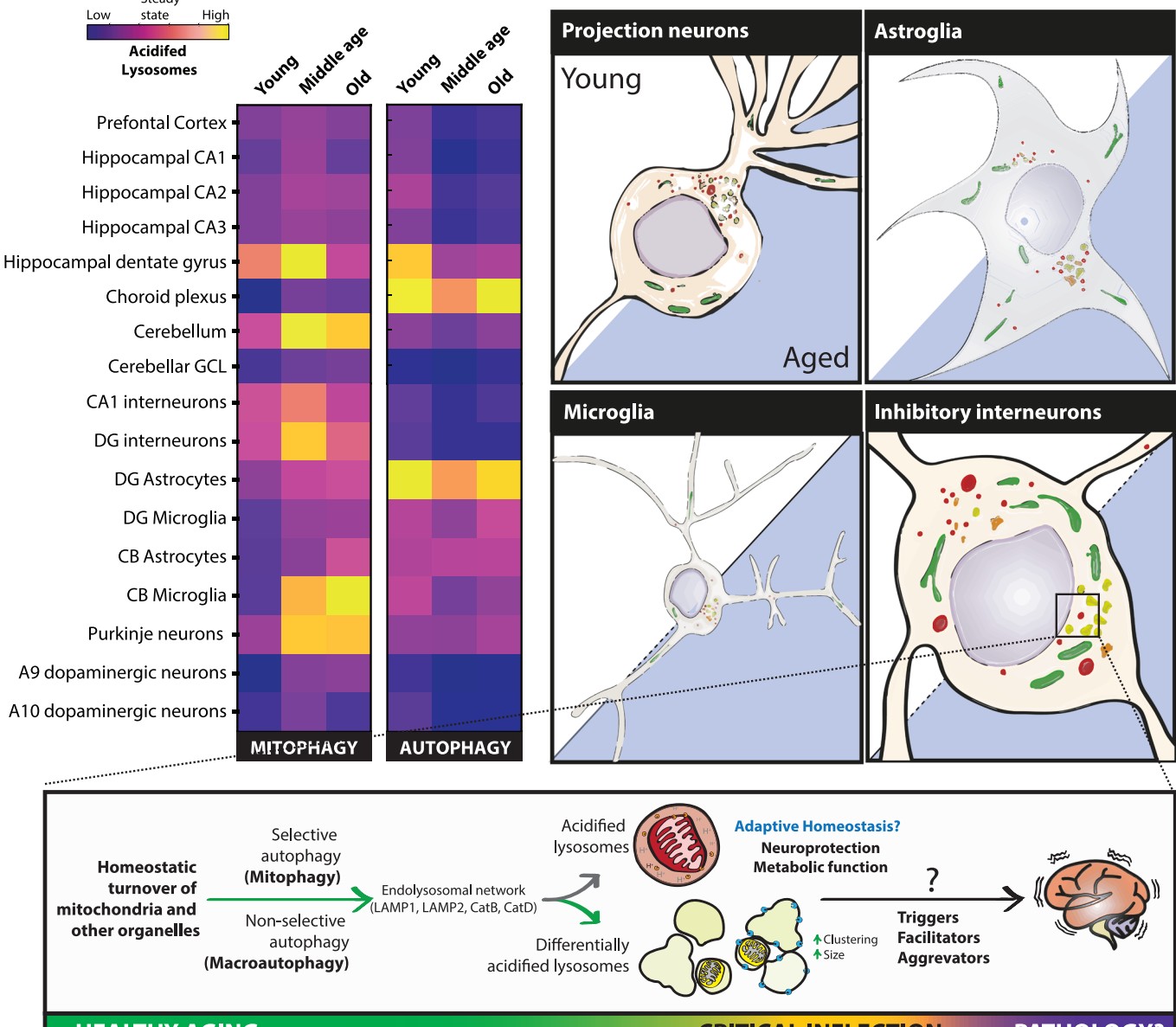

**Figure 9.   Spatiotemporal dynamics of mitophagy and macroautophagy in the brain throughout healthy aging.**

Mitophagy and macroautophagy are dynamic processes that exhibit cell and region-specific changes throughout mammalian life. While mitophagy increases throughout lifespan in some cell types e.g., Purkinje neurons, decreases are observed in others, for instance in the dentate gyrus. Macroautophagy levels appear more stable across the brain than mitophagy levels. These spatiotemporal changes in autophagy pathways provide a valuable preclinical map for future phenotyping efforts and for developing targeted interventions that address impairments in cellular turnover, which can lead to neurological dysfunction and disease. Graphs are visual summaries and conceptual representations of relative trends per region and subset analyzed, with phenotypic summaries presented in the manuscript.

mammalian CNS longitudinally during aging. While our work provides an important step forward as the first comparative and longitudinal in vivo mammalian mitophagy map, we acknowledge that our work raises a number of intriguing questions that are beyond the scope of this work. We show dynamic cell type- and brain region-specific regulation of mitophagy and macroautophagy, but our broad survey is incapable of deciphering regulatory

mechanisms and/or disease associations in each of these cell types. We were unable to analyze some cell types where antibody labeling requires heat-mediated antigen retrieval, as this is currently incompatible with tandem tag reporter systems. Comparing *mito*-QC and *auto*-QC mice provided us with a unique opportunity to decipher selective autophagy from general autophagic turnover, while the shared genetic background of these two mouse models

makes the comparison well controlled, it remains possible that other genetic backgrounds may behave differently than what we observe here. Furthermore, the measurement of mitophagy in vivo is an evolving area, and our reporter system readouts reflect the terminal phases of mitophagy and macroautophagy. Indeed, as mentioned in our discussion, the mt-Keima mouse model showed contrasting results, although we deduce that this is likely attributable to neuropathology, as the mt-Keima model does not exhibit healthy brain aging. While our reporter systems provide a powerful means to resolve mitophagy and autophagy within tissues, we are constrained by the lifespan of laboratory mice, which differs substantially from human subjects. Longitudinal analyses of selective autophagy in longer-lived model organisms may prove informative in the future. Regardless, such studies are currently very rare, and our findings establish a valuable resource that will aid in preclinical phenotyping and the development of precision autophagy therapeutics for incipient neuropathology.

# Methods

### Reagents and tools table

| Reagent/Resource | Reference or Source | Identifier or Catalog Number |
|---|---|---|
| **Experimental models** | | |
| Mouse model: *mito*-QC mice | McWilliams et al (2016). (Mice generated by TaconicArtemis GmbH) | PMID: 27458135 |
| Mouse model: *auto*-QC (mCherry-GFP-Map1lc3b) | McWilliams et al (2018a). (Mice generated by TaconicArtemis GmbH) | PMID: 29337137 |
| Human cells: ARPE *mito*-QC and *auto*-QC lines | McWilliams et al (2018a); Singh et al (2021). Generated by the Ganley lab, MRC PPU, UK. | PMID: 27458135; PMID: 34340748 |
| **Antibodies** | | |
| Rabbit polyclonal anti-tripartite motif containing protein (TRIM 46) | Synaptic Systems | Cat#377003; RRID: AB_2631232 |
| Rabbit polyclonal anti-tyrosine hydroxylase (TH) | Millipore | Cat#: AB152; RRID: AB_390204 |
| Rat monoclonal anti-lysosomal associated membrane protein 1 (LAMP1) (1D4B) | Santa Cruz Biotechnology | Cat#: sc-19992 RRID: AB_2134495 |
| Rat monoclonal anti-lysosomal associated membrane protein 2 (LAMP2) (ABL-93) | Developmental Studies Hybridoma Bank | Cat#ABL-93; RRID: AB_2134767 |
| Rabbit polyclonal anti-lysosomal associated membrane protein 2 (LAMP2) (ab18528) | Abcam | Cat# ab18528; RRID: AB_775981 |
| Goat polyclonal anti-mouse cathepsin D antibody | R&D Systems | Cat# AF1029; RRID: AB_2087094 |
| Goat polyclonal anti-mouse cathepsin B antibody | R&D Systems | Cat# AF965; RRID: AB_2086949 |
| Rabbit polyclonal anti-transmembrane protein 55B (TMEM55B) | Proteintech | Cat# 23992-1-AP; RRID: AB_2879391 |
| Rabbit polyclonal anti-Ionized calcium-binding adapter molecule 1 (IBA1) | FUJIFILM Wako Pure Chemical Corporation | Cat# 019-19741; RRID: AB_839504 |
| Rabbit polyclonal anti-glial fibrillary acidic protein (GFAP) | Dako | Cat# Z0334; RRID: AB_10013382 |
| Guinea pig polyclonal anti-parvalbumin | Synaptic Systems | Cat# 195004; RRID:AB_2156476 |
| Hoechst 33342, trihydrochloride trihydrate | Life Technologies (Thermo Fisher Scientific) | Cat# H1399; RRID: |
| Goat anti-Rabbit IgG (H + L) Cross-Adsorbed Secondary Antibody, Alexa Fluor™ 633 | Life Technologies (Thermo Fisher Scientific) | Cat# A-21070; RRID: AB_2535731 |
| Goat anti-Rabbit IgG (H + L) Secondary Antibody, Pacific Blue | Life Technologies (Thermo Fisher Scientific) | Cat#: P-10994; RRID: AB_2539814 |
| Goat anti-Rat IgG (H + L) Cross-Adsorbed Secondary Antibody, Alexa Fluor 633 | Life Technologies (Thermo Fisher Scientific) | Cat#: A-21094; RRID: AB_141553 |
| Goat anti-Guinea Pig IgG (H + L) Highly Cross-Adsorbed Secondary Antibody, Alexa Fluor™ 633 | Life Technologies (Thermo Fisher Scientific) | Cat# A-21105; RRID: AB_2535757 |
| Donkey Anti-Rat IgG H&L (Alexa Fluor® 405) preadsorbed | Abcam | Cat# ab175670; RRID: AB_3099480 |
| Rabbit polyclonal anti-Calbindin D-28k | Swant | Cat# CB38; RRID: AB_10000340 |
| Biotin-SP (long spacer) AffiniPure™ Donkey Anti-Goat IgG (H + L) | Jackson ImmunoResearch | Cat# 705-065-003; RRID: AB_2340396 |
| Streptavidin, Alexa Fluor™ 633 conjugate | Life Technologies (Thermo Fisher Scientific) | Cat# S-21375; RRID: AB_2313500 |
| **Chemicals, Enzymes, and other reagents** | | |
| Glutaraldehyde solution – grade I, 50% in H₂O | Sigma-Aldrich | Cat#: G7651 |
| VECTASHIELD Antifade Mounting Medium | Vector Laboratories | Cat#: H-1000; RRID: AB_2336789 |
| **Software** | | |
| Ilastik interactive learning and segmentation toolkit | Ilastik developers | https://www.ilastik.org/ |
| ImageJ/FIIJI | NIH | https://imagej.net/ |
| IMARIS Bitplane | Oxford Instruments | https://imaris.oxinst.com/ |

| Reagent/Resource | Reference or Source | Identifier or Catalog Number |
|---|---|---|
| CellProfiler cell image analysis software | Broad Institute | https://cellprofiler.org/ |
| Harmony software v5.22 | PerkinElmer | https://www.perkinelmer.com |
| Microscopy Image Browser (MIB) | Belevich et al (2016) | PMID: 26727152, https://mib.helsinki.fi/ |
| **Other** | | |
| ANDOR Dragonfly 505 high speed spinning disc confocal microscope with SRRF-stream | Oxford Instruments, ANDOR | N/A |
| Hitachi transmission electron microscope HIT7800 equipped with a bottom-mounted Rio9 CMOS camera | Hitachi High-Technologies | N/A |
| Opera Phenix Spinning Disk Confocal Microscope (High content screening platform) | PerkinElmer | N/A |

## Animals

All mice were maintained and housed according to ethically approved guidelines and welfare endpoints approved by the Office for Animal Experimentation of Finland. The mitophagy (*mito*-QC) reporter mouse model used in this study was generated and used as previously described (McWilliams et al, 2016, 2018a, 2018b; Lee et al, 2018; McWilliams et al, 2019; Alsina et al, 2020; Singh et al, 2021; Mito et al, 2022; Long et al, 2022) and maintained on the C57BL6 RccHsd background. Autophagy reporter mice (*mCherry-GFP-Map1lc3b*, referred to as *auto*-QC) were generated identically to *mito*-QC using targeted transgenesis by TaconicArtemis GmbH. Homozygous reporter animals were used in all analyses, and genotypes were confirmed by standard end-point PCR. Both female and male animals were analyzed in the study; the exact *n* numbers, refer to individual biological subjects which are specified in all figure legends. Four age categories are specified for *mito*-QC mice (3, 6, 15, and 19–26 months) and five age categories for *auto*-QC mice (3, 7, 11, 16, and 18–25 months). Aging categories were selected according to (Flurkey et al, 2007) the Jackson Laboratories (JAX) criteria on aging mice.

## Tissue processing and immunohistochemistry

Tissues were rapidly collected following euthanasia and processed by overnight immersion in freshly prepared and filtered fixative: 3.7% paraformaldehyde (Sigma, P6148), 200 mM HEPES, pH = 7.00. The following day, fixed tissues were washed extensively (at least three times) in DPBS and subjected to density-dependent cryoprotection by immersion in 30% (w/v) sucrose solution containing 0.04% sodium azide. Samples were stored at 4 °C in sucrose solution until further processing. Free-floating brain sections in both axial and para-sagittal orientations were acquired at a thickness of 40 μm using a Leica SM2010R sledge microtome equipped with a temperature-controlled

freezing stage and constant water supply (Physitemp, CA, USA). Free-floating *mito*-QC or *auto*-QC sections were either counterstained with the nuclear dye Hoechst 33342 (Life Technologies) or immunostained with antibodies where appropriate. For labeling of subcellular structures in vivo, *mito*-QC and *auto*-QC tissue sections were processed as previously described (McWilliams et al, 2016, 2018a, 2019) using the following antibodies: rabbit anti-calbindin D28K (1:5000; Swant; Catalog #CB38); rat anti-LAMP1 (1:1000; Santa Cruz Biotechnology; Catalog #19992); rabbit anti-tyrosine hydroxylase (1:2000; Millipore; Catalog #AB152), rabbit anti-TRIM46 (1:600; Synaptic Systems; Catalog #377005)); and rabbit anti-IBA1 (1:1000; Wako; Catalog #019-19741); rat anti-LAMP2 (1:200; DSHB; Catalog #ABL-93); rabbit anti-LAMP2 (1:200; Abcam; Catalog # AB18528); goat IgG Cathepsin D (1:200; R&D Systems; Catalog #AF1029); goat IgG Cathepsin B (1:200; R&D Systems; Catalog #AF965); rabbit anti-TMEM55B (1:750; Proteintech; Catalog #23992-1-AP); rabbit anti-GFAP (1:1000; DAKO; Catalog #Z0334) guinea pig anti-parvalbumin (1:1000; Synaptic systems; Catalog #195004). An additional step of streptavidin-biotin immunostaining was added for Cathepsin B and D. Secondary antibodies (1:500; Pacific Blue or Far Red) were purchased from Life Technologies; Donkey anti-rat 405 (1:500) was purchased from Abcam; Donkey anti-goat Biotin (1:200) was purchased from Jackson Immunoresearch and Streptavidin conjugate 633 (1:200) from Thermo Fisher Scientific. Tissue sections were mounted using Vectashield H-1000 (Vector Laboratories) on Leica SurgiPath X-tra Adhesive slides, Deckgläser Precision Cover Glasses (Marienfeld GmbH, DE) and sealed with transparent and rapid-drying nail polish before storage at 4 °C.

## Tissue confocal microscopy

Confocal photomicrographs (tile scans, z-stacks, and conventional images) were obtained by uniform random sampling using an ANDOR Dragonfly 505 High-Speed Spinning Disk Confocal Microscope with SRRF Stream SuperResolution, equipped with a Zyla sCMOS camera (Oxford Instruments, UK), and Plan Apochromat VC X60 objective (NA 1.2, water) with PerfectFocus System. Low-resolution tile scans were imaged using a 10× objective (NA 0.3, air) or 40× objective (NA 1.15, water). Representative images for lysosomal validation staining were imaged using a 100× objective (NA 1.49, oil). High-resolution representative images were deconvolved using the associated IMARIS Fusion software. For representative images, photomicrographs were digitally altered within linear parameters, with minimal adjustments to levels and linear contrast applied to all images.

## Image analysis and quantitation of mitophagy in vivo

Quantification of mitophagy and macroautophagy in vivo was conducted on at least 5 images per sample for cell type-specific analysis and at least 10 images per sample for unstained tissues. Single plane images were used for quantitation. Quantification was conducted using open source CellProfiler software (version 4.2.1) to threshold and segment mCherry-only positive mitolysosomes and autolysosomes. Briefly, the GFP signal was segmented out after thresholding and subtracted from the mCherry channel to ensure that only mCherry-positive puncta were detected for quantitation. For cell type-specific quantitation, an additional cell-segmentation

step was added. For this purpose, the machine learning software Ilastik was used to segment the cells, after which images were fed into CellProfiler for further segmentation, surface hole filling and quantitation. Due to the nature of the *auto*-QC reporter GFP signal, PCs were segmented using Ilastik, while *mito*-QC PC somata were segmented manually using FIJI ImageJ 1.53 (64-bit). IMARIS BitPlane was used to generate 3D isosurface renders of PCs, dopaminergic neurons and the AIS in vivo.

## Transmission electron microscopy and quantitation

Adult brain tissues were rapidly harvested, and <1 mm parasagittal cerebellar tissue sections were collected and immersion fixed in 4% PFAn in 0.1 M phosphate buffer (PB), pH 7.4, for 45 min. Following fixation, 300 μm parasagittal cerebellar tissue sections were acquired using a Leica VT1200/S 1v5 RevH vibratome immersed in 0.1 M PB. Vibratome sections were immediately fixed with 2% PFA and 2% glutaraldehyde (EM-grade) in 0.1 M sodium cacodylate buffer (NaCac), pH 7.4, supplemented with 2 mM $CaCl_2$ for 2 h at 4 °C. After 2 h, the fixative was exchanged for 0.1 M NaCac buffer supplemented with 2 mM $CaCl_2$, and the sections were stored at 4 °C until further processing. The vibratome sections were washed with 0.1 M NaCac buffer and postfixed with 1% osmium tetroxide in the same buffer for 1 h on ice, dehydrated through a series of ethanol and acetone solutions, and gradually embedded into epoxy resin (hard, TAAB 812, Aldermaston, UK). After polymerization of the epoxy resin at 60 °C for 18 h, a pyramid was trimmed, and a 500-nm-thick histological section was stained with toluidine blue to identify and verify cerebellar cortex regions containing PC profiles. Correlating with the light microscopy image from the histological section, the pyramid was downsized for electron microscopy, and 60 nm ultrathin sections were cut using an ultramicrotome (Leica EM Ultracut UC7, Leica Mikrosysteme GmbH, Austria), collected on Pioloform-coated single slot grids and post-stained with uranyl acetate and lead citrate. A Hitachi transmission electron microscope (HT7800, Hitachi High-Technologies, Tokyo, Japan) operated at 100 kV and equipped with a bottom-mounted Rio9 CMOS camera (Gatan Inc., Pleasanton, CA) was used for image acquisition. For quantitation, two images from the dendritic aspect and two images from the axonal aspect of the PC somata were acquired at a nominal magnification of 6000× from a total of 12 PCs per sample using systematic random sampling. Lysosomes were detected and area quantified based on ultrastructural features of contrast, morphology and presence of lipid droplets. For unbiased analysis, the detection of lysosomes and measurement of areas were performed on anonymized micrographs using MIB software (Belevich et al, 2016).

## Cell culture and phenotypic profiling of mitophagy and macroautophagy

Primary *mito*-QC and *auto*-QC mouse embryonic fibroblasts (MEFS) were isolated and prepared from reporter embryos as previously described (McWilliams et al, 2016, 2018a). Embryos were staged according to the criteria of Theiler (Theiler, 1972). Primary adult lung fibroblasts were isolated and prepared from 3-month *mito*-QC reporter mice (as described in Seluanov et al, 2010). Human ARPE19 cells expressing *mito*-QC or *auto*-QC were kindly provided by Professor Ian Ganley (MRC Protein Phosphorylation and Ubiquitylation Unit, University of Dundee, Scotland).

All cells were cultured in standard conditions, in DMEM-F12/10% FBS/Glutamax and penicillin-streptomycin at 37 °C/5% $CO_2$. All mammalian cells used in the study were seeded on Ibidi 24-well m-plates prior to imaging. Mitophagy and macroautophagy were induced as previously described (Long et al, 2022; Singh et al, 2021; Allen et al, 2013). For mitophagy induction, cultures were treated with 1 mM deferiprone (DFP) or vehicle alone for 24 h and 50 nM Bafilomycin A1 (BafA1) was added 6 h prior to fixation. For macroautophagy induction, cultures were treated with 1 μM of the highly selective mTORC1 inhibitor AZD8055 and 50 nM Baf A1 was added 3 h prior to fixation. Cells were fixed for 15 min at room temperature using 3.7% formaldehyde, 200 mM HEPES, pH 7.0. Nuclei were counterstained with Hoechst 33342 (1 μg/mL, Thermo Scientific, 62249) for 5 min. Cells were washed in PBS and stored in PBS prior to light microscopy. Visualizing *mito*-QC and *auto*-QC reporters was accomplished by fluorescence microscopy (PerkinElmer Opera Phenix Spinning Disk Confocal with a water immersion 40× objective (NA 1.1) using the relevant excitation lasers and emission filter sets.

## Image analysis and quantitation of mitophagy and autophagy in vitro

Autophagosomes and autolysosomes were quantified using Harmony software (v5.22, PerkinElmer). Briefly, we developed a custom analysis pipeline involving nuclear segmentation (Hoechst 33342 staining) and segmentation of the surrounding cytoplasmic region based on 488 nm signal background with Gaussian filtering. Puncta within the cytoplasmic region were segmented in both green and red channels using a spot detection module. After thresholding, acidic mitolysosomes or autolysosomes ($mCherry^{+ve}GFP^{-ve}$ i.e., 'red not green' puncta) were identified by masking to avoid quantification of false positives. Morphology and intensity features were then extracted from this selected subpopulation. The number of mitolysosomes or autolysosomes per cell was compared to the mean of the corresponding control as a ratio. Data from four biological replicates were compiled and analyzed. Single-cell results were plotted using GraphPad Prism 10.2.3, and statistical significance was determined using one-way ANOVA.

## Statistical analysis and graphics

Statistical analyses were performed in GraphPad Prism v10.2.3. Student's unpaired, two-tailed $t$ test was used for pairwise comparisons, and multiple comparisons were analyzed by one-way analysis of variance (ANOVA) with Bonferroni post hoc correction, as indicated in the figure legends. Statistical significance is displayed as $*P < 0.05$, $**P < 0.01$, $***P < 0.001$, and $****P < 0.0001$. Figures were assembled in Adobe Illustrator, and graphical illustrations or elements in Figs. 1, 3, 5 and 8, as well as the synopsis image were created with BioRender.com (https://biorender.com/).

# Data availability

All data generated or analyzed during this study are included in the manuscript, and also uploaded to Zenodo with the following identifier: https://doi.org/10.5281/zenodo.13304393.

## Peer review information

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

## Acknowledgements

Research in the T.G.M. laboratory is supported by the Academy of Finland (TGM: 310814), the Novo Nordisk Foundation/Novo Nordisk Fonden, the Sigrid Jusélius Foundation (TGM: 8045), the Päivikki and Sakari Sohlberg Foundation, Sydäntutkimussäätiö and the University of Helsinki. TGM is a Scholar of the FENS-Kavli Network of Excellence. AR is supported by a four-year scholarship from the Doctoral Program for Brain and Mind (University of Helsinki) and the Instrumentarium Science Foundation. We thank our colleagues at the FIMM-HCA for high content imaging and bioimage analysis, and for excellent technical support from colleagues at the HiLIFE Electron Microscopy Unit (both platforms are supported by HiLIFE, University of Helsinki, Biocenter Finland and Euro-Bioimaging ERIC - Finnish Advanced Microscopy Node). We are also grateful to our colleagues in the HiLIFE Laboratory Animal Centre Core Facility, University of Helsinki, for outstanding animal care and welfare monitoring. We further thank K.M. Mattinen and J. Salomaa for technical support at the initial phases of the project. We are grateful to Drs. M. Liljeström, K. Vonderstein, and E. Fazeli at the Biomedicum Imaging Unit (Biocenter Finland and HiLIFE supported infrastructure) for their outstanding expertise and guidance in microscopy and image analysis, and to Dr. A. Prescott (University of Dundee) and Dr. F. Singh (University of Iceland, IS) for invaluable discussions. We thank Dr. A. Pawluk (Life Science Editors) for her editorial feedback and input. We are grateful to Professor Ian Ganley (MRC Protein Phosphorylation and Ubiquitylation Unit, University of Dundee, Scotland) for informative discussions and the generous gift of ARPE19 reporter cells.

## Author contributions

**Anna Rappe**: Conceptualization; Data curation; Formal analysis; Investigation; Visualization; Methodology; Writing—original draft; Writing—review and editing. **Helena A Vihinen**: Data curation; Investigation; Visualization; Methodology; Writing—review and editing. **Fumi Suomi**: Data curation; Investigation; Visualization; Methodology; Project administration; Writing—review and editing. **Antti J Hassinen**: Data curation; Software; Formal analysis; Validation; Investigation; Visualization; Methodology; Writing—review and editing. **Homa Ehsan**: Investigation; Methodology; Project administration; Writing—review and editing. **Eija S Jokitalo**: Methodology; Writing—review and editing. **Thomas G McWilliams**: Conceptualization; Resources; Supervision; Funding acquisition; Methodology; Writing—original draft; Project administration; Writing—review and editing.

## Disclosure and competing interests statement

The authors declare no competing interests.

# Expanded View Figures

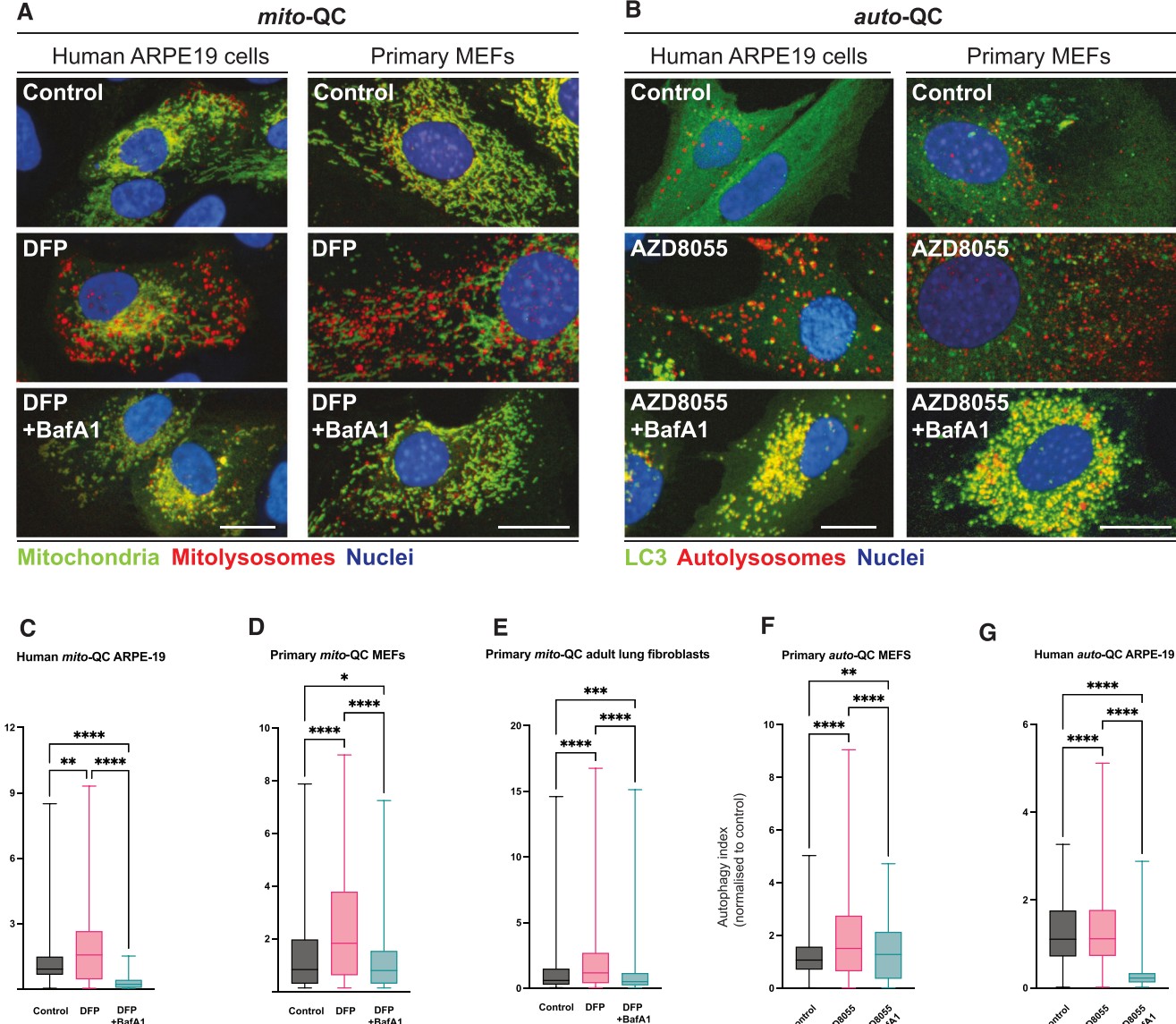

**Figure EV1. Additional validation of optical reporter systems.**

(**A**) Representative images of *mito*-QC human ARPE19 cells and Primary MEFs. (**B**) Representative images of *auto*-QC human ARPE19 cells and Primary MEFs. (**C**) Mitophagy in human *mito*-QC ARPE19 cells. One-way ANOVA with Tukey's post hoc. ****$P < 0.0001$, ***$P = 0.0001$. $n = 4$. (**D**) Mitophagy in *mito*-QC primary MEFS. One-way ANOVA with Tukey's post hoc. ****$P < 0.0001$. $n = 4$. (**E**) Mitophagy in adult *mito*-QC primary fibroblasts. One-way ANOVA with Tukey's post hoc. ****$P < 0.0001$, ***$P = 0.0005$. $n = 4$. (**F**) Autophagy in *auto*-QC primary MEFS. One-way ANOVA with Tukey's post hoc. ****$P < 0.0001$. $n = 4$. (**G**) Autophagy in human *auto*-QC ARPE19 cells. One-way ANOVA with Tukey's post hoc. ****$P < 0.0001$, **$P = 0.0022$. $n = 4$. Source data are available online for this figure.

# *mito*-QC - regional analysis - all channels

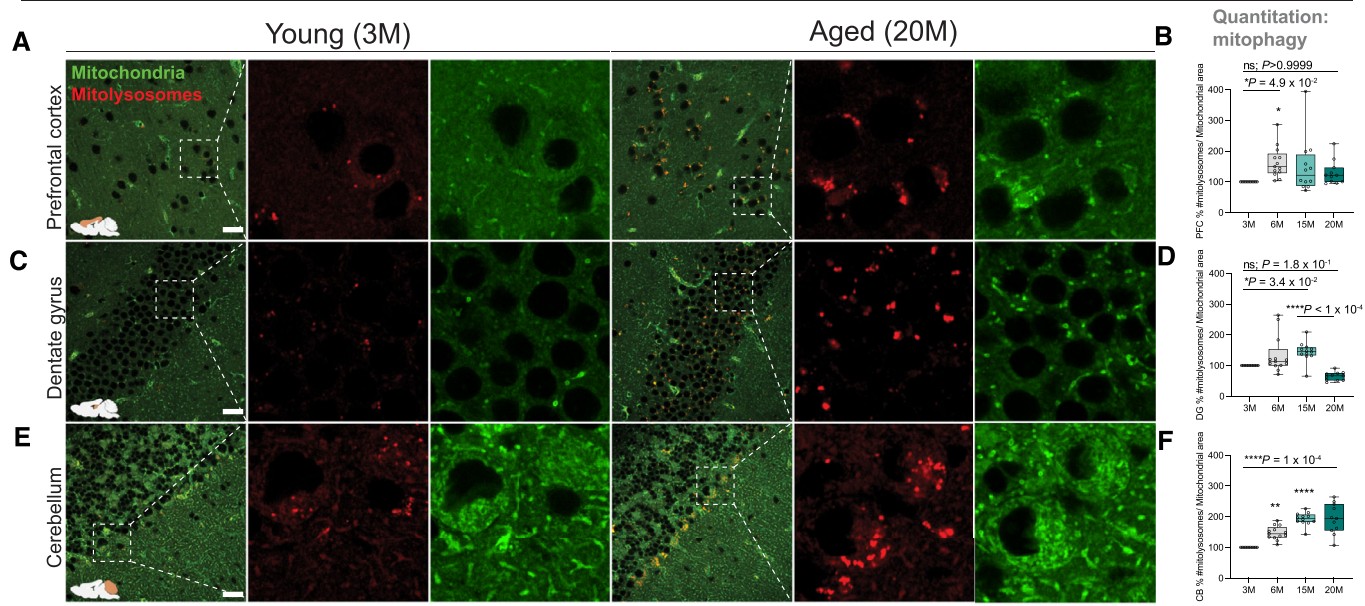

# *auto*-QC - regional analysis - all channels

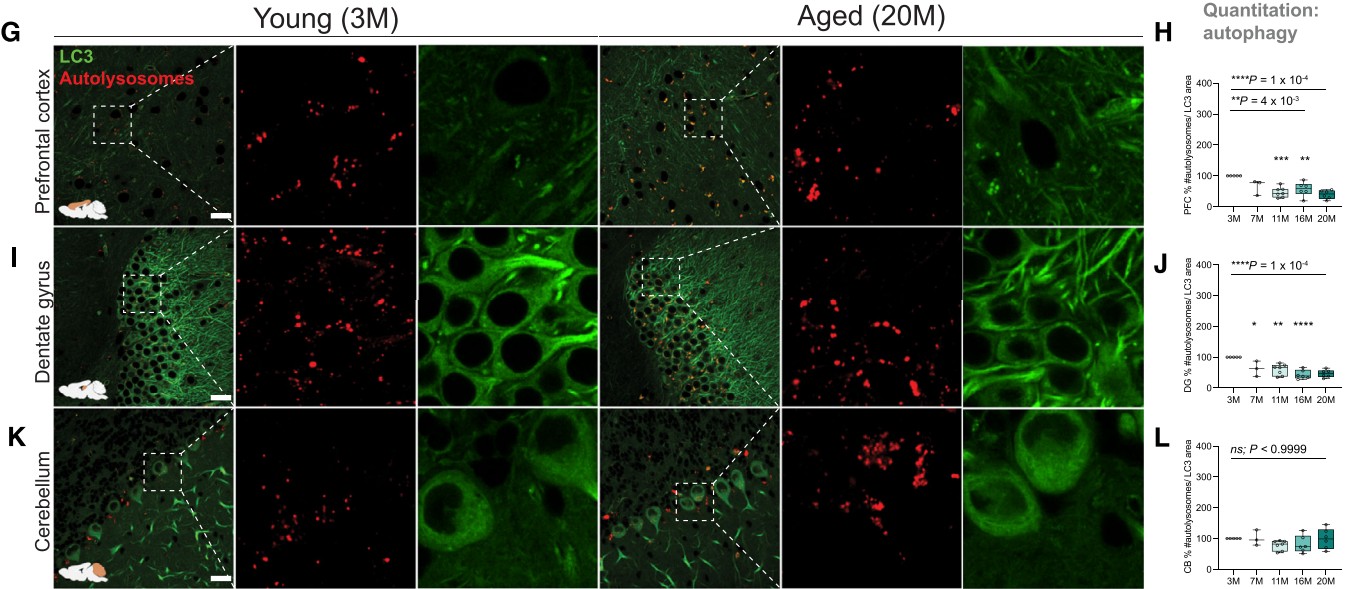

◀

**Figure EV2.  Temporal dynamics of regional mitophagy and macroautophagy in the CNS—all channels (contains redisplay from Fig. 2).**

(A, B) PFC mitophagy (redisplay of Fig. 2A with GFP channel). Representative confocal photomicrographs detailing instances of PFC mitophagy in young and geriatric mice. GFP and mCherry channels are shown for clarity, alongside quantitative analysis of % PFC mitophagy levels relative to mitochondrial area across all ages. Mean of 3-month group standardized to 100%. *$P = 0.0489$; ns = not significant; $P > 0.9999$. $n = 47$. (C, D) DG mitophagy (redisplay of Fig. 2C with GFP channel). Representative confocal photomicrographs detailing instances of DG mitophagy in young and geriatric mice. GFP and mCherry channels are shown for clarity, alongside quantitative analysis of % DG levels relative to mitochondrial area across all ages. Mean of 3-month group standardized to 100%. ****$P < 0.0001$, *$P = 0.0338$; ns = not significant; $P = 0.1759$. $n = 47$. (E, F) CB mitophagy (redisplay of Fig. 2E with GFP channel). Representative confocal photomicrographs detailing instances of CB mitophagy in young and geriatric mice. GFP and mCherry channels are shown for clarity, alongside quantitative analysis of % CB mitophagy levels relative to mitochondrial area across all ages. Mean of 3-month group standardized to 100%. ****$P < 0.0001$, **$P < 0.0014$. $n = 47$. Scale bar 20 µm. (G, H) PFC macroautophagy (redisplay of Fig. 2G with GFP channel). Representative confocal photomicrographs detailing instances of PFC macroautophagy in young and geriatric auto-QC mice. GFP and mCherry channels are shown for clarity, alongside quantitative analysis of % PFC macroautophagy levels relative to LC3 area across all ages. Mean of 3-month group standardized to 100%. ****$P < 0.0001$, ***$P = 0.0003$ **$P = 0.0040$. $n = 27$. (I, J) DG macroautophagy (redisplay of Fig. 2I with GFP channel). Representative confocal photomicrographs detailing instances of DG macroautophagy in young and geriatric auto-QC mice. GFP and mCherry channels are shown for clarity, alongside quantitative analysis of % DG macroautophagy levels relative to LC3 area across all ages. Mean of 3-month group standardized to 100%. ****$P < 0.0001$, **$P = 0.0012$, *$P = 0.0261$. $n = 27$. (K, L) CB macroautophagy (redisplay of Fig. 2K with GFP channel). Representative confocal photomicrographs detailing instances of PFC macroautophagy in young and geriatric auto-QC mice. GFP and mCherry channels are shown for clarity, alongside quantitative analysis of % CB macroautophagy levels relative to LC3 area across all ages. Mean of 3-month group standardized to 100%. ns = not significant; $P > 0.9999$. $n = 27$. Scale bar 20 µm. Box plots extend from the 25th to the 75th percentiles, with a median line positioned inside the box. Whiskers denote the minimum and maximum values. Source data are available online for this figure.

## Additional morphometric data - all regions and cell-types

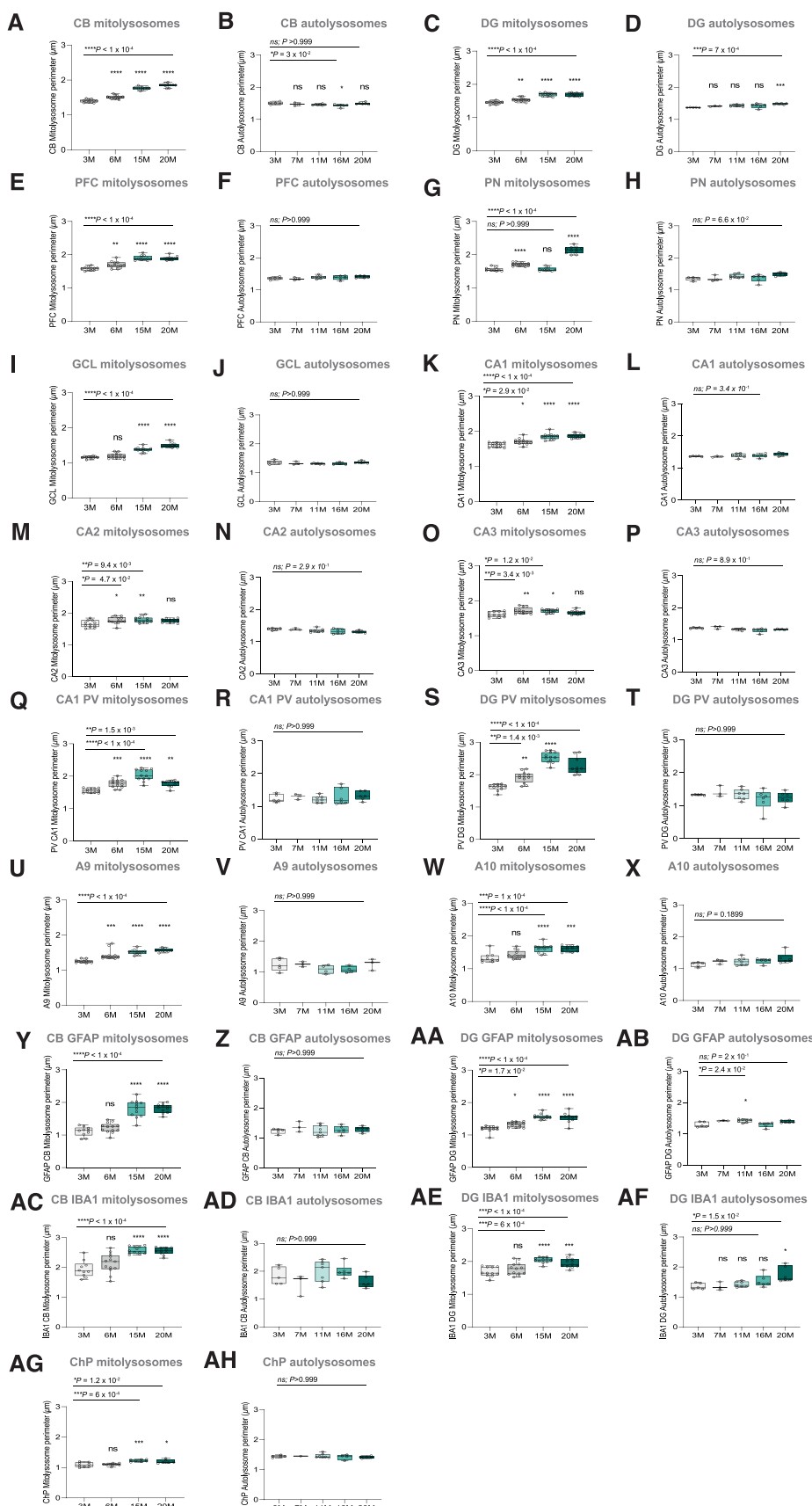

◀ **Figure EV3.  Profiling of mitolysosome and autolysosome perimeter across the aging brain. Contains analysis from all cell-types and regions for completeness.**

(A) Cerebellar mean mitolysosome perimeter. Quantitative analysis reveals increased mitolysosome perimeter in geriatric mice compared to young. One-way ANOVA with Bonferroni post hoc. ****$P < 0.0001$. $n = 47$. (B) Cerebellar mean autolysosome perimeter. Quantitative analysis reveals no significant changes in autolysosome perimeter between geriatric mice compared to young. A modest decline in autolysosome perimeter is observed at 16 months. One-way ANOVA with Bonferroni post hoc. ns = not significant; $P$ value > 0.9999, *$P = 0.0297$. $n = 27$. (C) Dentate gyrus mean mitolysosome perimeter. Quantitative analysis reveals increased mitolysosome perimeter in geriatric mice compared to young. One-way ANOVA with Bonferroni post hoc. ****$P < 0.0001$. $n = 47$. (D) Dentate gyrus mean autolysosome perimeter. Quantitative analysis reveals a significant increase in autolysosome perimeter in geriatric mice compared to young. One-way ANOVA with Bonferroni post hoc. ***$P = 0.0007$. $n = 27$. (E) PFC mean mitolysosome perimeter. Quantitative analysis reveals increased mitolysosome perimeter in geriatric mice compared to young. One-way ANOVA with Bonferroni post hoc. ****$P < 0.0001$. $n = 47$. (F) PFC mean autolysosome perimeter. Quantitative analysis reveals no significant changes in autolysosome perimeter between geriatric mice compared to young. One-way ANOVA with Bonferroni post hoc. ns = not significant; $P > 0.9999$. $n = 27$. (G) Purkinje cell mean mitolysosome perimeter. Quantitative analysis reveals increased mitolysosome perimeter in geriatric mice compared to young. One-way ANOVA with Bonferroni post hoc. ****$P < 0.0001$. $n = 47$. (H) Purkinje cell mean autolysosome perimeter. Quantitative analysis reveals no significant changes in autolysosome perimeter between geriatric mice compared to young. One-way ANOVA with Bonferroni post hoc. ns = not significant; $P = 0.0657$. $n = 27$. (I) Granular cell layer mean mitolysosome perimeter. Quantitative analysis reveals increased mitolysosome perimeter in geriatric mice compared to young. One-way ANOVA with Bonferroni post hoc. ****$P < 0.0001$. $n = 47$. (J) Granular cell layer mean autolysosome perimeter. Quantitative analysis reveals no alterations in autolysosome perimeter between geriatric mice and young. One-way ANOVA with Bonferroni post hoc. ns = not significant; $P > 0.9999$. $n = 27$. (K) CA1 mean mitolysosome perimeter. Quantitative analysis reveals increased mitolysosome perimeter in geriatric mice compared to young. One-way ANOVA with Bonferroni post hoc. ****$P$ value < 0.0001; *$P = 0.0290$. $n = 47$. (L) CA1 mean autolysosome perimeter. Quantitative analysis reveals no alterations in autolysosome perimeter between geriatric mice and young. One-way ANOVA with Bonferroni post hoc. ns = not significant; $P = 0.3350$. $n = 27$. (M) CA2 mean mitolysosome perimeter. Quantitative analysis reveals fluctuations in mitolysosome perimeter in throughout lifespan. One-way ANOVA with Bonferroni post hoc. **$P = 0.0094$, *$P = 0.0467$; ns = not significant; $P = 0.0875$. $n = 47$. (N) CA2 mean autolysosome perimeter. Quantitative analysis reveals no alterations in autolysosome perimeter between geriatric mice and young. One-way ANOVA with Bonferroni post hoc. ns = not significant; $P$ value = 0.2924. $n = 27$. (O) CA3 mean mitolysosome perimeter. Quantitative analysis reveals fluctuations in mitolysosome perimeter in throughout lifespan. One-way ANOVA with Bonferroni post hoc. **$P = 0.0034$, *$P = 0.0122$; ns = not significant; $P = 0.2975$. $n = 47$. (P) CA3 mean autolysosome perimeter. Quantitative analysis reveals no alterations in autolysosome perimeter between geriatric mice and young. One-way ANOVA with Bonferroni post hoc. ns = not significant; $P = 0.8866$. $n = 27$. (Q) CA1 Parvalbumin interneuron mean mitolysosome perimeter. Quantitative analysis reveals increased mitolysosome perimeter in geriatric mice compared to young. One-way ANOVA with Bonferroni post hoc. ****$P < 0.0001$, **$P = 0.0015$. $n = 47$. (R) CA1 Parvalbumin interneuron mean autolysosome perimeter. Quantitative analysis reveals no alterations in autolysosome perimeter between geriatric mice and young. One-way ANOVA with Bonferroni post hoc. ns = not significant; $P$ value > 0.9999. $n = 27$. (S) Dentate gyrus Parvalbumin interneuron mean mitolysosome perimeter. Quantitative analysis reveals increased mitolysosome perimeter in geriatric mice compared to young. One-way ANOVA with Bonferroni post hoc. ****$P < 0.0001$, **$P = 0.0014$. $n = 46$. (T) Dentate gyrus Parvalbumin interneuron mean autolysosome perimeter. Quantitative analysis reveals no alterations in autolysosome perimeter between geriatric mice and young. One-way ANOVA with Bonferroni post hoc. ns = not significant; $P > 0.9999$. $n = 27$. (U) A9 DA neuron mean mitolysosome perimeter. Quantitative analysis reveals increased mitolysosome perimeter in geriatric mice compared to young. One-way ANOVA with Bonferroni post hoc. ****$P < 0.0001$. $n = 43$. (V) A9 DA neuron mean autolysosome perimeter. Quantitative analysis reveals no significant changes in autolysosome perimeter between geriatric mice compared to young. One-way ANOVA with Bonferroni post hoc. ns = not significant; $P > 0.9999$. $n = 19$. (W) A10 DA neuron mean mitolysosome perimeter. Quantitative analysis reveals increased mitolysosome perimeter in geriatric mice compared to young. One-way ANOVA with Bonferroni post hoc. ****$P < 0.0001$; ***$P = 0.0001$. $n = 44$. (X) A10 DA neuron mean autolysosome perimeter. Quantitative analysis reveals no significant changes in autolysosome perimeter between geriatric mice compared to young. One-way ANOVA with Bonferroni post hoc. ns = not significant; $P = 0.1899$. $n = 26$. (Y) Cerebellar astrocyte mean mitolysosome perimeter. Quantitative analysis reveals increased mitolysosome perimeter in geriatric mice compared to young. One-way ANOVA with Bonferroni post hoc. ****$P < 0.0001$. ns = not significant; $P = 0.9203$. $n = 46$. (Z) Cerebellar astrocyte mean autolysosome perimeter. Quantitative analysis reveals no significant changes in autolysosome perimeter between geriatric mice compared to young. One-way ANOVA with Bonferroni post hoc. ns = not significant; $P > 0.9999$. $n = 27$. (AA) Dentate gyrus astrocyte mean mitolysosome perimeter. Quantitative analysis reveals increased mitolysosome perimeter in geriatric mice compared to young. One-way ANOVA with Bonferroni post hoc. ****$P < 0.0001$. *$P = 0.0165$. $n = 47$. (AB) Dentate gyrus astrocyte mean autolysosome perimeter. Quantitative analysis reveals a modest increase in perimeter at midlife compared to young and returning to no significant change in geriatric mice. One-way ANOVA with Bonferroni post hoc. *$P = 0.0240$. ns = not significant; $P = 0.1955$. $n = 27$. (AC) Cerebellar microglia mean mitolysosome perimeter. Quantitative analysis reveals increased mitolysosome perimeter in geriatric mice compared to young. One-way ANOVA with Bonferroni post hoc. ****$P < 0.0001$. $n = 46$. (AD) Cerebellar microglia mean autolysosome perimeter. Quantitative analysis reveals no significant changes in autolysosome perimeter between geriatric mice compared to young. One-way ANOVA with Bonferroni post hoc. ns = not significant; $P > 0.9999$. $n = 27$. (AE) Dentate gyrus microglia mean mitolysosome perimeter. Quantitative analysis reveals increased mitolysosome perimeter in geriatric mice compared to young. One-way ANOVA with Bonferroni post hoc. ****$P < 0.0001$ ***$P = 0.0006$. $n = 47$. (AF) Dentate gyrus microglia mean autolysosome perimeter. Quantitative analysis reveals a modest increase in autolysosome perimeter in geriatric mice compared to young. One-way ANOVA with Bonferroni post hoc. *$P = 0.00153$. $n = 27$. (AG) ChP mean mitolysosome perimeter. Quantitative analysis reveals a modest increase in mitolysosome perimeter in geriatric mice compared to young. Some fluctuation with increased mitolysosome perimeter is observed at 15 months. One-way ANOVA with Bonferroni post hoc. ***$P = 0.0006$, *$P = 0.0118$. $n = 30$. (AH) ChP mean autolysosome perimeter. Quantitative analysis reveals no significant changes in autolysosome perimeter between geriatric mice compared to young. One-way ANOVA with Bonferroni post hoc. ns = not significant; $P > 0.9999$. $n = 22$. Box plots extend from the 25th to the 75th percentiles, with a median line positioned inside the box. Whiskers denote the minimum and maximum values. Source data are available online for this figure.

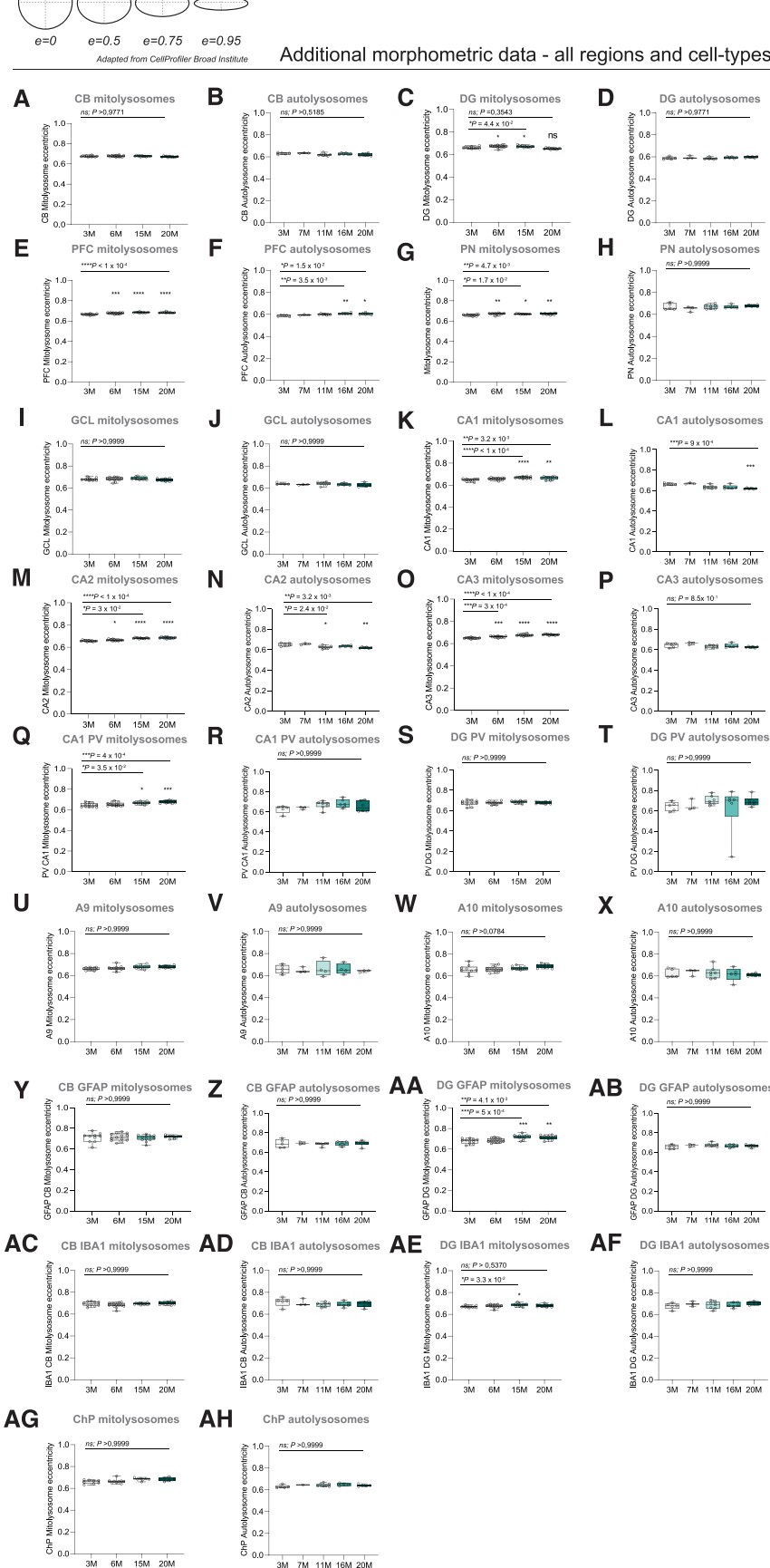

Additional morphometric data - all regions and cell-types

**Figure EV4. Profiling of mitolysosome and autolysosome eccentricity across the aging brain. Contains analysis from all cell-types and regions for completeness.**

(A) Cerebellar mean mitolysosome eccentricity. Quantitative analysis reveals no significant changes in mitolysosome eccentricity in geriatric mice compared to young. One-way ANOVA with Bonferroni post hoc. ns = not significant; $P > 0.9771$. $n = 47$. (B) Cerebellar mean autolysosome eccentricity. Quantitative analysis reveals no significant changes in autolysosome eccentricity between geriatric mice compared to young. One-way ANOVA with Bonferroni post hoc. ns = not significant; $P = 0.5185$. $n = 27$. (C) Dentate gyrus mean mitolysosome eccentricity. Quantitative analysis reveals a modest increased mitolysosome eccentricity in during mature adulthood and midlife stages, before returning to corresponding eccentricity values in geriatric animals as in young. One-way ANOVA with Bonferroni post hoc. *$P < 0.0436$. ns = not significant; $P = 0.3543$. $n = 47$. (D) Dentate gyrus mean autolysosome eccentricity. Quantitative analysis reveals no significant changes in autolysosome eccentricity between geriatric mice compared to young. One-way ANOVA with Bonferroni post hoc. ns = not significant; $P = 0.9231$. $n = 27$. (E) PFC mean mitolysosome eccentricity. Quantitative analysis reveals increased mitolysosome eccentricity in geriatric mice compared to young, indicating an elongated morphology as a function of age. One-way ANOVA with Bonferroni post hoc. ****$P < 0.0001$. $n = 47$. (F) PFC mean autolysosome eccentricity. Quantitative analysis reveals a modest increase in autolysosome eccentricity between geriatric mice compared to young animals, indicating mitolysosome elongation. Peak eccentricity values are observed at 16 months. One-way ANOVA with Bonferroni post hoc. *$P = 0.0146$, **$P = 0.0035$. $n = 27$. (G) Purkinje cell mean mitolysosome eccentricity. Quantitative analysis reveals increased mitolysosome eccentricity in geriatric mice compared to young. PC mitolysosome undergoes modest fluctuations in eccentricity throughout aging One-way ANOVA with Bonferroni post hoc. *$P = 0.0170$, **$P = 0.0047$. $n = 47$. (H) Purkinje cell mean autolysosome eccentricity. Quantitative analysis reveals no significant changes in autolysosome eccentricity between geriatric mice compared to young. One-way ANOVA with Bonferroni post hoc. ns = not significant; $P > 0.9999$. $n = 27$. (I) Granular cell layer mean mitolysosome eccentricity. Quantitative analysis reveals no significant changes in mitolysosome eccentricity in geriatric mice compared to young. One-way ANOVA with Bonferroni post hoc. ns = not significant; $P > 0.9999$. $n = 47$. (J) Granular cell layer mean autolysosome eccentricity. Quantitative analysis reveals no significant changes in autolysosome eccentricity in geriatric mice compared to young. One-way ANOVA with Bonferroni post hoc. ns = not significant; $P > 0.9999$. $n = 27$. (K) CA1 mean mitolysosome eccentricity. Quantitative analysis reveals increased mitolysosome eccentricity in geriatric mice compared to young. One-way ANOVA with Bonferroni post hoc. ****$P < 0.0001$. **$P = 0.0032$. $n = 47$. (L) CA1 mean autolysosome eccentricity. Quantitative analysis reveals a significant decline in geriatric mice compared to young, indicating a more elongated shape with aging. One-way ANOVA with Bonferroni post hoc. ***$P = 0.0009$. $n = 27$. (M) CA2 mean mitolysosome eccentricity. Quantitative analysis reveals increased mitolysosome eccentricity in geriatric mice compared to young. One-way ANOVA with Bonferroni post hoc. ****$P$ value $< 0.0001$. *$P = 0.0302$. $n = 47$. (N) CA2 mean autolysosome eccentricity. Quantitative analysis reveals a significant decline in geriatric mice compared to young, indicating a more elongated shape with aging. One-way ANOVA with Bonferroni post hoc. **$P = 0.0032$. *$P = 0.0237$. $n = 27$. (O) CA3 mean mitolysosome eccentricity. Quantitative analysis reveals increased mitolysosome eccentricity in geriatric mice compared to young. One-way ANOVA with Bonferroni post hoc. ****$P < 0.0001$. ***$P = 0.0003$. $n = 47$. (P) CA3 mean autolysosome eccentricity. Quantitative analysis reveals no significant change in geriatric mice compared to young. One-way ANOVA with Bonferroni post hoc. ns = not significant; $P = 0.8531$. $n = 27$. (Q) CA1 Parvalbumin interneuron mean mitolysosome eccentricity. Quantitative analysis reveals increased mitolysosome eccentricity in geriatric mice compared to young. One-way ANOVA with Bonferroni post hoc. ***$P = 0.0004$. *$P = 0.0350$. $n = 47$. (R) CA1 Parvalbumin interneuron mean autolysosome eccentricity. Quantitative analysis reveals no significant changes in autolysosome eccentricity in geriatric mice compared to young. One-way ANOVA with Bonferroni post hoc. ns = not significant; $P > 0.9999$. $n = 27$. (S) Dentate gyrus Parvalbumin interneuron mean mitolysosome eccentricity. Quantitative analysis reveals no significant change in geriatric mice compared to young. One-way ANOVA with Bonferroni post hoc. ns = not significant; $P > 0.9999$. $n = 46$. (T) Dentate gyrus Parvalbumin interneuron mean autolysosome eccentricity. Quantitative analysis reveals no significant change in geriatric mice compared to young. One-way ANOVA with Bonferroni post hoc. ns = not significant; $P > 0.9999$. $n = 27$. (U) A9 DA neuron mean mitolysosome eccentricity. Quantitative analysis reveals increased mitolysosome eccentricity in geriatric mice compared to young. One-way ANOVA with Bonferroni post hoc. ns = not significant; $P > 0.0689$. $n = 43$. (V) A9 DA neuron mean autolysosome eccentricity. Quantitative analysis reveals no significant changes in autolysosome eccentricity between geriatric mice compared to young. One-way ANOVA with Bonferroni post hoc. ns = not significant; $P > 0.9999$. $n = 19$. (W) A10 DA neuron mean mitolysosome eccentricity. Quantitative analysis reveals increased mitolysosome eccentricity in geriatric mice compared to young. One-way ANOVA with Bonferroni post hoc. ns = not significant; $P > 0.0784$. $n = 44$. (X) A10 DA neuron mean autolysosome eccentricity. Quantitative analysis reveals no significant changes in autolysosome eccentricity between geriatric mice compared to young. One-way ANOVA with Bonferroni post hoc. ns = not significant; $P > 0.9999$. $n = 26$. (Y) Cerebellar astrocyte mean mitolysosome eccentricity. Quantitative analysis reveals no significant change in mitolysosome eccentricity in geriatric mice compared to young. One-way ANOVA with Bonferroni post hoc. ns = not significant; $P > 0.9999$. $n = 46$. (Z) Cerebellar astrocyte mean autolysosome eccentricity. Quantitative analysis reveals no significant changes in autolysosome eccentricity between geriatric mice compared to young. One-way ANOVA with Bonferroni post hoc. ns = not significant; $P > 0.9999$. $n = 27$. (AA) Dentate gyrus astrocyte mean mitolysosome eccentricity. Quantitative analysis reveals increased mitolysosome eccentricity in geriatric mice compared to young animals. One-way ANOVA with Bonferroni post hoc. ***$P = 0.0005$. **$P = 0.0041$. $n = 47$. (AB) Dentate gyrus astrocyte mean autolysosome eccentricity. Quantitative analysis reveals no significant changes between geriatric mice compared to young. One-way ANOVA with Bonferroni post hoc. ns = not significant; $P$ value $> 0.9999$. $n = 27$. (AC) Cerebellar microglia mean mitolysosome eccentricity. Quantitative analysis reveals increased mitolysosome eccentricity in geriatric mice compared to young. One-way ANOVA with Bonferroni post hoc. ns = not significant; $P > 0.7837$. $n = 46$. (AD) Cerebellar microglia mean autolysosome eccentricity. Quantitative analysis reveals no significant changes in autolysosome eccentricity between geriatric mice compared to young. One-way ANOVA with Bonferroni post hoc. ns = not significant; $P > 0.9999$. $n = 27$. (AE) Dentate gyrus microglia mean mitolysosome eccentricity. Quantitative analysis reveals no significant change in mitolysosome eccentricity in geriatric mice compared to young animals, although a modest increase can be observed at 15 months One-way ANOVA with Bonferroni post hoc. *$P = 0.0328$, ns = not significant; $P > 0.5370$. $n = 47$. (AF) Dentate gyrus microglia mean autolysosome eccentricity. Quantitative analysis reveals no significant changes in autolysosome eccentricity between geriatric mice compared to young. One-way ANOVA with Bonferroni post hoc. ns = not significant; $P > 0.9999$. $n = 27$. (AG) ChP mean mitolysosome eccentricity. Quantitative analysis reveals increased mitolysosome eccentricity in geriatric mice compared to young. One-way ANOVA with Bonferroni post hoc. ns = not significant; $P = 0.0922$. $n = 30$. (AH) ChP mean autolysosome eccentricity. Quantitative analysis reveals no significant changes in autolysosome eccentricity between geriatric mice compared to young. One-way ANOVA with Bonferroni post hoc. ns = not significant $P > 0.9999$. $n = 22$. Box plots extend from the 25th to the 75th percentiles, with a median line positioned inside the box. Whiskers denote the minimum and maximum values. Source data are available online for this figure.

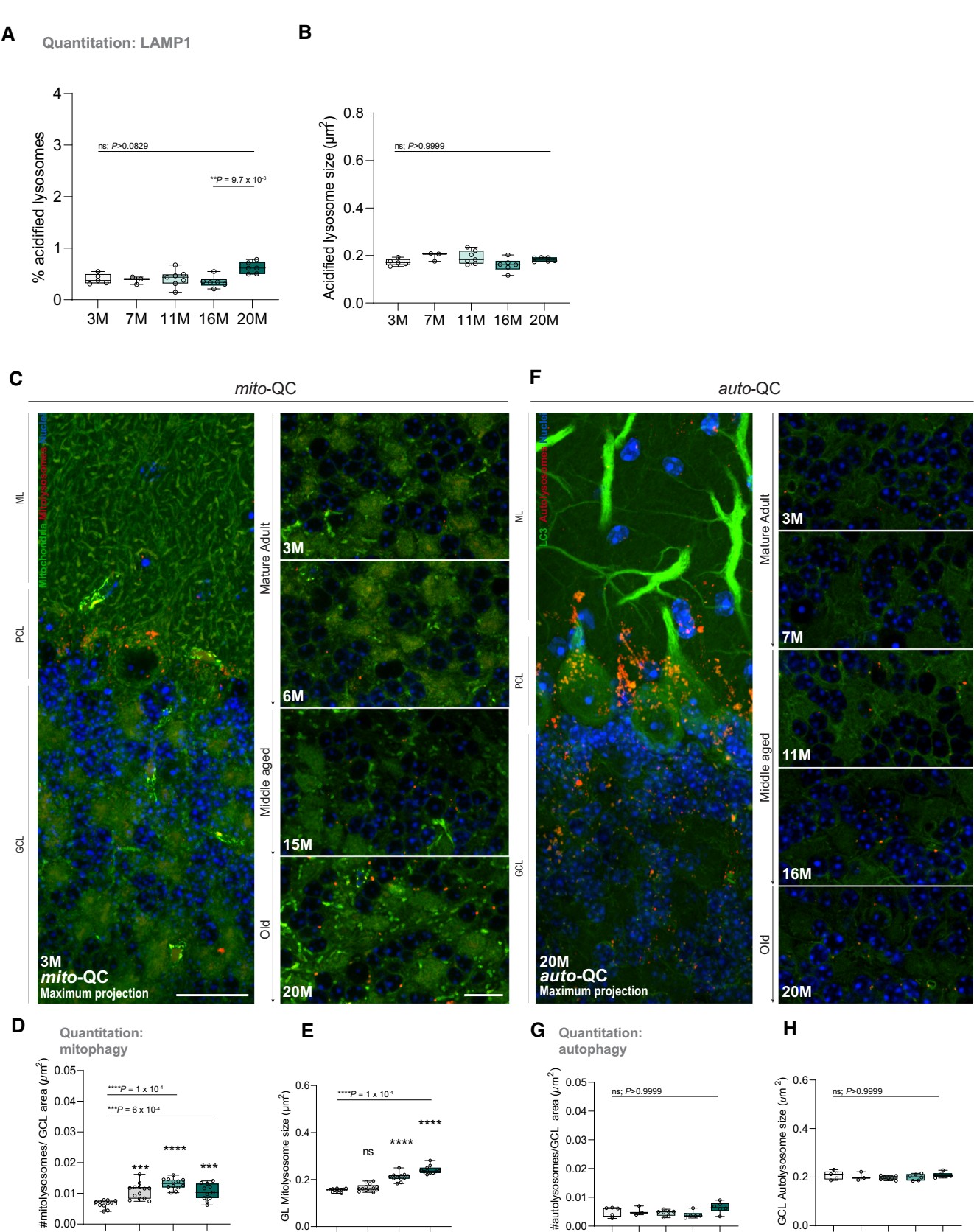

◄ **Figure EV5. Mitophagy and macroautophagy in cerebellar granule neurons in vivo.**

(A) Purkinje cell acidified autolysosomes. The number of LAMP1-positive structures that are double-positive for mCherry-only remains unchanged between young and geriatric mice. Some fluctuations of increased acidified autolysosomes can be observed at 16 months compared to the geriatric. One-way ANOVA with Bonferroni post hoc. **$P < 0.0097$, ns = not significant; $P > 0.0829$. $n = 27$. (B) Purkinje cell mean acidified autolysosome size. mCherry-LAMP1 double-positive structure size remain constant throughout aging. One-way ANOVA with Bonferroni post hoc. ns = not significant; $P > 0.9999$. $n = 27$. (C–E) Mitophagy in the aging GCL. Representative images of mitophagy events throughout aging in the *mito*-QC granular cell layer. Quantitative analysis reveals increased levels of mitophagy in geriatric mice compared to young, as do mean mitolysosome size. One-way ANOVA with Bonferroni post hoc. Scale bars = 20 μm. ****$P < 0.0001$. $n = 47$. (F–H) Autophagy in the aging GCL. Representative images of autophagy events throughout aging in the *auto*-QC granular cell layer. Quantitative analysis reveals unaltered levels of autophagy in geriatric mice compared to young, nor do autolysosome size change. One-way ANOVA with Bonferroni post hoc. Scale bars = 20 μm. ns = not significant; $P > 0.9999$. $n = 27$. Box plots extend from the 25th to the 75th percentiles, with a median line positioned inside the box. Whiskers denote the minimum and maximum values. Source data are available online for this figure.

