## [Peer Review File · The EMBO Journal]

Longitudinal autophagy profiling of the mammalian brain reveals sustained mitophagy throughout healthy aging

Anna Rappe, Helena Vihinen, Fumi Suomi, Antti Hassinen, Homa Ehsan, Eija Jokitalo, and Thomas McWilliams

Corresponding author: Thomas McWilliams (thomas.mcwilliams@helsinki.fi)

Review Timeline:

Submission Date:	17th Nov 23
Editorial Decision:	29th Jan 24
Revision Received:	14th Jun 24
Editorial Decision:	15th Jul 24
Revision Received:	24th Jul 24
Accepted:	16th Aug 24

Editor: Daniel Klimmeck

Transaction Report:

Dear Dr Thomas McWilliams,

Thank you for the submission of your manuscript (EMBOJ-2023-115700) to The EMBO Journal. Please accept my sincere apologies for the unusual delay with the peer-review of your manuscript at this time of the year. Your manuscript has been initially sent to three reviewers, however one reviewer got much delayed and in the end did not send us his-her report even after repeated chasers. We have received reports from the other two referees, which I enclose below, and now decided to proceed with our decision based on these comments.

As you will see, the referees acknowledge the potential interest and novelty of your results, although they also express a number of major issues that will have to be conclusively addressed before they can be supportive of publication of your manuscript in The EMBO Journal. In more detail, reviewer #1 states as important caveat a lack of disambiguation between phagolysosome formation versus clearance thus increased flux versus decreased degradation (ref#1, pts.1,5). Further, this expert asks you to rule out a deficient lysosome state of the alternative potential causal event (ref#1, pt.2). Reviewer #3 requests extension of the analyses to additional brain areas and complementary stainings (ref#3, pts.1-4). In addition, the reviewers raise several points related to additional controls required to corroborate the findings, overall data quantification and representation as well as a revised discussion of the results, that would need to be conclusively addressed to achieve the level of robustness and clarity needed for The EMBO Journal.

Given the overall interest stated and broader angle of your findings, we are able to invite you to revise your manuscript experimentally to address the referees' comments, pending there are no overriding technical concerns by referee #2.

In light of the extensive experimentation requested, I would appreciate if you could contact me during the next weeks for exchange e.g. a video call to discuss your perspective on the comments and potential plan for revisions.

I will let you know once we still receive the report by referee #2.

Please feel free to contact me if you have any questions or need further input on the referee comments.

When submitting your revised manuscript, please carefully review the instructions below.

Please feel free to approach me any time should you have additional questions related to this.

Thank you for the opportunity to consider your work for publication.

I look forward to your revision.

Best regards,

Daniel Klimmeck

Daniel Klimmeck, PhD
Senior Editor
The EMBO Journal

Instruction for the preparation of your revised manuscript:

2) individual production quality figure files as .eps, .tif, .jpg (one file per figure).

3) a .docx formatted letter INCLUDING the reviewers' reports and your detailed point-by-point response to their comments. As part of the EMBO Press transparent editorial process, the point-by-point response is part of the Review Process File (RPF), which will be published alongside your paper.

4) a complete author checklist, which you can download from our author guidelines ([https://wol-prod-cdn.literatumonline.com/pb-assets/embo-site/Author Checklist%20-%20EMBO%20J-1561436015657.xlsx](https://wol-prod-cdn.literatumonline.com/pb-assets/embo-site/Author%20Checklist%20-%20EMBO%20J-1561436015657.xlsx)). Please insert information in the checklist that is also reflected in the manuscript. The completed author checklist will also be part of the RPF.

6) It is mandatory to include a 'Data Availability' section after the Materials and Methods. Before submitting your revision, primary datasets produced in this study need to be deposited in an appropriate public database, and the accession numbers and database listed under 'Data Availability'. Please remember to provide a reviewer password if the datasets are not yet public (see <https://www.embopress.org/page/journal/14602075/authorguide#datadeposition>).

7) Our journal encourages inclusion of *data citations in the reference list* to directly cite datasets that were re-used and obtained from public databases. Data citations in the article text are distinct from normal bibliographical citations and should directly link to the database records from which the data can be accessed. In the main text, data citations are formatted as follows: "Data ref: Smith et al, 2001" or "Data ref: NCBI Sequence Read Archive PRJNA342805, 2017". In the Reference list, data citations must be labeled with "[DATASET]". A data reference must provide the database name, accession number/identifiers and a resolvable link to the landing page from which the data can be accessed at the end of the reference. Further instructions are available at .

8) At EMBO Press we ask authors to provide source data for the main and EV figures. Our source data coordinator will contact you to discuss which figure panels we would need source data for and will also provide you with helpful tips on how to upload and organize the files.

Numerical data can be provided as individual .xls or .csv files (including a tab describing the data). For 'blots' or microscopy, uncropped images should be submitted (using a zip archive or a single pdf per main figure if multiple images need to be supplied for one panel). Additional information on source data and instruction on how to label the files are available at .

9) We replaced Supplementary Information with Expanded View (EV) Figures and Tables that are collapsible/expandable online (see examples in <https://www.embopress.org/doi/10.15252/emboj.201695874>). A maximum of 5 EV Figures can be typeset. EV Figures should be cited as 'Figure EV1, Figure EV2' etc. in the text and their respective legends should be included in the main text after the legends of regular figures.

11) For data quantification: please specify the name of the statistical test used to generate error bars and P values, the number (n) of independent experiments (specify technical or biological replicates) underlying each data point and the test used to calculate p-values in each figure legend. The figure legends should contain a basic description of n, P and the test applied. Graphs must include a description of the bars and the error bars (s.d., s.e.m.).

We realize that it is difficult to revise to a specific deadline. In the interest of protecting the conceptual advance provided by the work, we recommend a revision within 3 months (28th Apr 2024). Please discuss the revision progress ahead of this time with the editor if you require more time to complete the revisions.

Referee #1:

This study is a tour de force characterizing how the number of mitolysosomes and autophagolysosomes change in mouse brain during ageing. The authors imaged brain sections of mice expressing the mitoQC and the autoQC reporters and monitoring how subcellular structures positive for acidic-pH quenched mitoQC and autoQC change with age and in different areas of the brain. From this approach, authors conclude that mitophagy and autophagy fluxes are not decreased with age in certain areas of the brain, which is different from what was observed in other organisms with shorter lifespans.

The major caveat of the study is that one cannot conclude changes in autophagic and mitophagic fluxes only based on the number of quenched mitoQC and autoQC structures. With this limitation, the major concerns are the following:

- 1) Without blocking the autophagic and mitophagic flux, one cannot conclude whether a higher number of red structures represent more formation of auto- and mito-phagolysosomes or decreased clearance of these structures. Authors could procure aged mice, perform intracranial injection of virus encoding these reporters and treat with compounds that block digestion and/or formation of mitoQC and autoQC structures. This is essential to determine whether an accumulation of these structures truly represent an increase in flux, as a higher number of these structures could also suggest impaired degradation and thus decreased flux, as observed in other organisms.
- 2) Dysfunctional lysosomes with poor digestion activity can still retain some acidity that would block GFP fluorescence, but such dysfunctional lysosomes would still decrease mito- and auto-lysosomal flux. Thus, it is a possibility that the accumulation in red structures is reflecting a lysosomal defect.
- 3) In the graphical scheme describing the reporters, the auto-QC probe seems to be facing the cytosolic side of the autophagolysosomal membrane, which it should not be acidic. Are the authors proposing that GFP will be quenched when the mCherry-GFP is exposed to the cytosol or is it a graphical typo? Is it a possibility that the auto-QC probe is reporting on leaky lysosomes and thus decreasing the pH on the auto- mito-phagosome membrane side facing the cytosol? The latter could suggest that the probe is also reporting on dysfunctional lysosomes.
- 4) It is not clear what the n represent in the figure legends, as well as the individual points in the graphs. Do they represent different mice? How many brain fields were imaged per mouse? Are there differences in the number of the different types of neurons induced by ageing that could explain differences in the number of structures per image?
- 5) An alternative to address point number 1 would be to promote a healthier ageing of these mice and see what healthier ageing does to the number of structures. Caloric restriction could be an example, but it is not an experiment feasible for a revision given that 18 months are needed. Maybe treatment with a compound that is demonstrated to improve brain function during ageing.

Referee #3:

EMBOJ-2023-115700, corr. author Prof. McWilliams

"Longitudinal autophagy profiling of mammalian brain circuits reveals sustained mitophagy throughout healthy aging"

This resource manuscript by Rappe and colleagues employs two well-characterized reporter mice, the mito-QC and the auto-QC

to investigate *in vivo* the effects of postnatal maturation and aging on brain mitophagy and general autophagy, respectively. To this end, they focus on three main brain areas, the prefrontal cortex, dentate gyrus of the hippocampus and cerebellum. Subsequently, they also investigate three distinct neuronal types, namely Purkinje neurons of the cerebellum, and TH-positive neurons of the substantia nigra and of the ventral tegmental area (VTA). Finally, they also investigate microglia of the cerebellum and of the dentate gyrus, as well as cells of the choroid plexus. The main conclusion of these analyses is that a) the regulation of mitophagy across lifespan in the brain does not follow a general pattern but is instead cell-type specific; b) Contrary to previous reports, mitophagy does not decline with aging across all brain cells and may even increase with aging in some cell types.

Major concerns:

Overall, I find this to be a very interesting approach. In addition to providing a spatiotemporal map of mitophagy *in vivo*, it also showcases the potential of these reporter mice for studying the regulation of mitophagy in brain cells under other physiological or pathological conditions. However, there are some weaknesses mainly related to the breadth of the approach and to how the analyses have been performed, that need to be addressed for this work to be suitable for publication in EMBOJ.

1. A major concern relates to the quantification method. The authors measure mitophagy and autophagy by reporting the density of red-only puncta, meaning of mitochondria or autophagosomes that have fused with acidic lysosomes. I find this analysis to provide incomplete information, as it doesn't tell us whether an increase or decrease in this number reflects corresponding changes in the total number of mitochondria or autophagosomes that appear yellow. Therefore, one cannot deduce how homeostatic the differences in mitophagic or autophagic flux and activity are. One would expect that a cell with significantly more mitochondria will correspondingly have more mitolysosomes, but this may not always be as trivial an equation, especially in the aged condition. I believe that it's essential to present the mito- and auto-lysosomes as percentages of all mitochondria and autophagosomes, respectively, for all the different areas and cell types investigated. This will allow for additional or alternative interpretations of the data.
2. If I understood it well, when investigating brain areas, 2-D images were analyzed and the results are normalized for area (μm^2). The total tile area imaged for the PFC, DG and CB is not clearly indicated in the materials and methods, making it difficult to judge how representative the resulting data are of the entirety of each structure. However, when analyzing immunolabelled cell types (such as TH-positive neurons, or IBA-1-positive microglia), a 3-D z-stack is imaged and analyzed. Therefore, the results in Figure 4 and 5 should be normalized to volume and not to area.
3. While I appreciate that it is challenging for this type of approach to be exhaustive and address all brain cells, the breadth of the study seems fragmented and should be expanded. There can be a justification for focusing on most brain areas or cell subtypes. There are several very interesting brain areas, such as the CA1 and CA3 regions of the hippocampus, the amygdala etc., that have not been considered, making this manuscript not as encompassing as expected for a resource. I urge the authors to at least provide data for all the hippocampal areas to complement their analyses in the dentate gyrus. Similarly, the fact that they have not investigated mitophagy and autophagy in astrocytes and oligodendrocytes, despite the fact that there are good antibodies for immunohistochemistry to mark these cells, is not satisfactory given the mounting interest in autophagy in these pivotal glial cell types. These analyses should be performed and included.
4. The findings on microglia are very interesting. They suggest that both in the DG and the cerebellum, mitophagy follows a similar pattern, increasing progressively with aging, while general autophagy is more or less stable across ages. I wonder whether this reflects the state of microglia as opposed to their homogeneity. IBA-1, the marker used to label these cells, is known to be enriched in active microglia, which may be more abundant or activated to a greater extent in the aged brain. To exclude this possibility, the authors should use an additional microglial markers, such as P2Y12R for example.
5. With regards to Figure 3, there's a lot of evidence that LAMP1 is a promiscuous marker that does not exclusively mark lysosomes (see for example PMID: 29940787). In neuronal dendrites, about 70% of LAMP1-positive structure contain lysosomal enzymes (see for example PMID: 37590146). Antibodies against active lysosomal enzymes and other lysosomal markers, such as LAMP2, should be used to support any claim to the fusion of the yellow mitochondria with non-acidified lysosomes.

Minor concerns:

1. In their discussion, the authors should compare their approach not only to the mito-keima reporter mouse, but also to what has been recently learned about mitophagy from other types of approaches, such as proteomic analyses of purified brain autophagosomes and of autophagosome enriched brain fractions.
2. It seems that some graphs are repeated in the main figures. Unless I am mistaken, the top graphs of Figure 3d and e are the same as panels Figure 2f and l, respectively. A better arrangement should be applied to avoid such repetitions.
3. High magnification representative images should also be provided for the green channel in Figure 2 and where else applicable.

Author response

We are grateful to the editor and reviewers for their time and effort in evaluating our manuscript and helping to improve it. We appreciate their constructive feedback and recognition of the quality, originality and significance of the study. As detailed below, we have endeavoured to address the reviewers' constructive criticisms as comprehensively as possible. We have revised several figures to include new primary data, and the manuscript now includes several new important pieces of data, including the characterization of hippocampal subregions and distinct niches of both astroglia and inhibitory interneurons, in which mitophagy and macroautophagy have not previously been studied *in vivo*, to our knowledge. The revisions have strongly supported our conclusion that the dynamic modulation of autophagy pathways is a clear feature of mammalian brain aging. Our manuscript is greatly strengthened as a result and paves the way for many exciting future investigations.

We wish to address that during the preparation of our revisions, a related manuscript was published (PMID: 38280852), presenting a snapshot aging analysis with a similar conceptual finding. An important distinction between the studies is that our resource manuscript reveals the longitudinal progression of both mitophagy and macroautophagy dynamics across multiple neural subsets at single cell resolution *in vivo*, whereas that paper had a distinct inflammaging and retinal emphasis. Despite the publication of this complementary study, our manuscript contains many distinct findings and a broad-scope resource that will advance the field. We believe that the existence of this related work strengthens the findings of our study and further emphasizes its timeliness. Ultimately, we expect our findings will be of broad utility to researchers in discovery and translational neuroscience who seek to study and target autophagy pathways for therapeutic benefit.

Tom McWilliams

Table summary of changes to revised manuscript

Figure number	Former Figure number	Modifications	
Figure 1	Unchanged	Figure 1	Updated annotation and corrected graphical typo
Figure 2	Unchanged	Figure 2 a-k	Updated annotation and minor graphical elements added
Figure 3	Unchanged	Figure 3 g	Graph header changed for clarity, title moved and modified
Figure 4	New data	Figure 4 a	mito -QC hippocampal tile scan and CA1 representative maximum projections
		Figure 4 b	mito -QC CA1-CA3 quantitation of mitophagy and DALs
		Figure 4 c	auto -QC hippocampal tile scan and CA1 representative maximum projections
		Figure 4 d	auto -QC CA1-CA3 quantitation of autophagy and DALs
		Figure 4 e	mito -QC CA1 and DG interneuron representative images
		Figure 4 f	CA1 mitophagy and autophagy quantitation
		Figure 4 g	auto -QC CA1 and DG interneuron representative images
		Figure 4 h	DG mitophagy and autophagy quantitation
Figure 5	Figure 4	Figure 5d,f	Titles added for maximum projections
Figure 6	New data	Figure 6 a	mito -QC cerebellar astrocyte representative images
		Figure 6 b	mito -QC CB astrocyte quantitation
		Figure 6 c	auto -QC cerebellar astrocyte representative images
		Figure 6 d	auto -QC CB astrocyte quantitation
		Figure 6 e	mito -QC dentate gyrus astrocyte representative images
		Figure 6 f	mito -QC DG astrocyte quantitation
		Figure 6 g	auto -QC dentate gyrus astrocyte representative images
		Figure 6 h	auto -QC DG astrocyte quantitation
Figure 7	Former Figure 5		Titles updated for clarity
Figure 8	Former Figure 6		Titles added for maximum projections
Figure 9	New figure	Summary figure	
EV1	New data	Figure EV1 a	mito -QC representative images of human AEP19 cells and Primary MEFs DFP and BafA1 treatment
		Figure EV1 b	auto -QC representative images of human AEP19 cells and Primary MEFs AZD8055 and BafA1 treatment
		Figure EV1 c	mito -QC AEP19 quantitation
		Figure EV1 d	mito -QC Primary MEFs quantitation
		Figure EV1 e	mito -QC young (3M) primary fibroblast quantitation
		Figure EV1 f	auto -QC Primary MEFs quantitation
		Figure EV1 g	auto -QC AEP19 quantitation
EV2	Figure 2 + new data	Figure EV2 a, c, e	mito -QC added micrograph of GFP signal
		Figure EV2 b, d, f	mito -QC alternative quantitation method
		Figure EV2 g, i, k	auto -QC added micrograph of GFP signal
		Figure EV2 h, j, l	auto -QC alternative quantitation method
EV3	Former figure S1 + including new data	Figure EV3 a-ah	mito - and autolysosome perimeter quantitation
EV4	Former figure S2 + including new data	Figure EV4 a-ah	mito - and autolysosome eccentricity quantitation
EV5	Former figure S3		Titles updated for clarity

Figure number	Former Figure number	Modifications	
EV6	New data	Figure EV6 a	Former S4a - mito-QC LAMP1 validation staining
		Figure EV6 b	LAMP2 validation staining
		Figure EV6 c	LAMP2 and LAMP1 validation staining
		Figure EV6 d	LAMP1 and CATB validation staining
		Figure EV6 e	LAMP1 and CATD validation staining
		Figure EV6 f	LAMP1 and TMEM55B validation staining
EV7	Former figure S4		Figure S4a moved to Figure EV2a
EV8	Former figure S5 + including new data	Figure EV8 a-ah	mito- and autolysosome size quantitation
EV9	Former Figure S6 + including new data	Figure EV9 a-ah	mito- and auto -QC sex-dependent quantitation
EV10	New data	Figure EV10 a-ah	mito- and auto -QC DAL quantitation

Referee #1:

“This study is a tour de force characterizing how the number of mitolysosomes and autophagolysosomes change in mouse brain during ageing. The authors imaged brain sections of mice expressing the mitoQC and the autoQC reporters and monitoring how subcellular structures positive for acidic-pH quenched mitoQC and autoQC change with age and in different areas of the brain. From this approach, authors conclude that mitophagy and autophagy fluxes are not decreased with age in certain areas of the brain, which is different from what was observed in other organisms with shorter lifespans.”

- **We thank the reviewer for their positive feedback on our manuscript and their appreciation for its novelty and impact.**

“The major caveat of the study is that one cannot conclude changes in autophagic and mitophagic fluxes only based on the number of quenched mitoQC and autoQC structures. With this limitation, the major concerns are the following:

1) Without blocking the autophagic and mitophagic flux, one cannot conclude whether a higher number of red structures represent more formation of auto- and mito-phagolysosomes or decreased clearance of these structures. Authors could procure aged mice, perform intracranial injection of virus encoding these reporters and treat with compounds that block digestion and/or formation of mitoQC and autoQC structures. This is essential to determine whether an accumulation of these structures truly represent an increase in flux, as a higher number of these structures could also suggest impaired degradation and thus decreased flux, as observed in other organisms.”

- **We appreciate the reviewer's perspective and have integrated supporting experiments and literature to address this concern in the revised manuscript. Our optical autophagy reporters and similar tandem-tag systems have been extensively characterised as providing reliable readouts of autophagic flux (e.g. genetic depletion of *Atg5* *in vivo* preturbs the *mito-QC* readout; PMID: 29500189). While it was not feasible for us to perform the exact suggested experiment *in vivo* as we no longer have aged mice, we demonstrated the following to address the reviewer's point:**
 - **We induced mitophagy in human ARPE-19 cells, as well as in primary mouse embryonic and adult fibroblasts expressing *mito-QC* using deferiprone (DFP), a therapeutic iron chelator that promotes high levels of mitochondrial turnover (PMID: 24176932). Consistent with many previously published studies, DFP-treated cells exhibit abundant red-only mitolysosomes compared to untreated controls. Conversely, administration of bafilomycin A1 (Baf A1) profoundly disrupts lysosomal acidification, resulting in significantly less mCherry-only mitolysosomes.**
 - **In human and primary mouse *auto-QC* reporter cells, we performed analogous experiments, inducing macroautophagy *via* mTOR inhibition using the highly selective ATP-competitive inhibitor, AZD8055. This induced the widespread formation of mCherry-only autolysosomes, whose formation was also impaired upon Baf A1 treatment (please see below).**

- These data collectively demonstrate that the formation of mCherry-only structures in our tandem reporter systems is a function of lysosomal activity, consistent with the many previous validation and data from prior studies (PMID: 24176932, 27458135, 29500189, 31843465, 32525278, 30538104, 32291905, 31661466, 29500189, 31066324, 29895712, 31678243, 30404819, 31361392, 29337137, 30786807, 31900402, 31101940, 30941770, 32164182, 32636217, 32420530, 31339428). We have included these new data in the extended view Figure EV1 of the revised manuscript. We have also amended the text to provide a more coherent narrative context and rationale in the results section, which sets the stage for the subsequent *in vivo* analyses.

“2) Dysfunctional lysosomes with poor digestion activity can still retain some acidity that would block GFP fluorescence, but such dysfunctional lysosomes would still decrease mito- and auto-lysosomal flux. Thus, it is a possibility that the accumulation in red structures is reflecting a lysosomal defect.”

- We respect the reviewers assessment and have endeavoured to be cautious and nuanced in the interpretation of our data. Indeed, we have more formally investigated

the incidence of differential acidification in the revised manuscript. Ultimately, our study demonstrates that subsets of aging neural cells harbour heterogeneous pools of both acidified (mCherry-only) and differentially-acidified lysosomes (DALs). Because DALs accumulate throughout aging (Figure 3), we assert that age-associated lysosomal dysfunction is a key finding of our study and we hope this is now better emphasised in the manuscript (inclusion of 34 new graphs quantifying the trajectory and emergence of DALs across all cell types and regions profiled in the manuscript, new Figure EV6).

- It is a ‘chicken and egg’ problem to determine whether the increased levels of acidified mCherry-only mitolysosomes might eventually become ‘differentially acidified’. If this does occur, it is possible this could explain the increase in DALs within neural subsets. Lysosomes exhibit heterogeneity and it is plausible that a subpopulation of lysosomes becomes dysfunctional through time, or equally, that lysosomal repair mechanisms may be affected by aging. We think our data in the healthy aging context are consistent with findings of the Ralph Nixon laboratory and others, where ‘poorly acidified lysosomes’ (PALs) have been identified as key contributors to neuropathology (PMID: 35654956, 35947489, 35486730, 37287072). Our longitudinal analyses provide a nuanced understanding of the spatiotemporal dynamics of mitophagy, macroautophagy, and lysosomal homeostasis in healthy aging, surpassing conventional snapshot comparisons between young and old mice.

“3) In the graphical scheme describing the reporters, the auto-QC probe seems to be facing the cytosolic side of the autophagolysosomal membrane, which it should not be acidic. Are the authors proposing that GFP will be quenched when the mCherry-GFP is exposed to the cytosol or is it a graphical typo? Is it a possibility that the auto-QC probe is reporting on leaky lysosomes and thus decreasing the pH on the auto- mito-phagosome membrane side facing the cytosol? The latter could suggest that the probe is also reporting on dysfunctional lysosomes.”

- **We thank the reviewer for spotting this graphical typo. We have corrected this error in the revised manuscript.**

*“4) It is not clear what the *n* represent in the figure legends, as well as the individual points in the graphs. Do they represent different mice? How many brain fields were imaged per mouse? Are there differences in the number of the different types of neurons induced by ageing that could explain differences in the number of structures per image?”*

- **We apologise for the lack of clarity. Yes, *n* represents the number of mice and the points in the graphs represent individual animals. We have amended these details accordingly in the figure legends and methods of the revised manuscript.**

“5) An alternative to address point number 1 would be to promote a healthier ageing of these mice and see what healthier ageing does to the number of structures. Caloric restriction could be an example, but it is not an experiment feasible for a revision given that 18 months are needed. Maybe treatment with a compound that is demonstrated to improve brain function during ageing.”

- **We thank the reviewer for this interesting suggestion, and are grateful for their understanding of time constraints during the revision period. Indeed, it is not feasible**

for us to perform this experiment as we do not have aging mouse cohorts at this time. However, we can draw attention to the many studies using our reporter systems that have observed changes in the number of tissue mitolysosomes in response to various interventions. Examples include increased mitophagy following pharmacogenetic inhibition of LRRK2 (PMID: 34340748), altered macroautophagy following nutrient restriction (PMID: 29337137), therapeutic rapamycin administration (PMID: 35030325) and increased physiological mitophagy following Roxadustat/hypoxia treatment (PMID: 33536245) or ROCK inhibition (PMID: 31900402). It is difficult to say whether any of these could improve ‘brain function’ in the context of aging, but they independently demonstrate that therapeutic interventions can alter the number of mCherry-only structures. We anticipate the present study will provide a useful blueprint for future precision medicine efforts. We have added the following line to the revised discussion:

“It will be intriguing to examine how longitudinal autophagy dynamics and lysosomal homeostasis are modified by emergent geroprotective and neuroprotective interventions.”

“Referee #3:

“This resource manuscript by Rappe and colleagues employs two well-characterized reporter mice, the mito-QC and the auto-QC to investigate in vivo the effects of postnatal maturation and aging on brain mitophagy and general autophagy, respectively. To this end, they focus on three main brain areas, the prefrontal cortex, dentate gyrus of the hippocampus and cerebellum. Subsequently, they also investigate three distinct neuronal types, namely Purkinje neurons of the cerebellum, and TH-positive neurons of the substantia nigra and of the ventral tegmental area (VTA). Finally, they also investigate microglia of the cerebellum and of the dentate gyrus, as well as cells of the choroid plexus.

The main conclusion of these analyses is that a) the regulation of mitophagy across lifespan in the brain does not follow a general pattern but is instead cell-type specific; b) Contrary to previous reports, mitophagy does not decline with aging across all brain cells and may even increase with aging in some cell types.

Major concerns:

Overall, I find this to be a very interesting approach. In addition to providing a spatiotemporal map of mitophagy in vivo, it also showcases the potential of these reporter mice for studying the regulation of mitophagy in brain cells under other physiological or pathological conditions. However, there are some weaknesses mainly related to the breadth of the approach and to how the analyses have been performed, that need to be addressed for this work to be suitable for publication in EMBOJ.”

- **We thank the the reviewer for their positive remarks and for recognising the importance of our findings uncovering cell-type specific autophagy regulation in the aging mammalian brain. Our responses to both reviewers’ comments have greatly improved the breadth of this study, as detailed in our point-by-point response.**

“1. A major concern relates to the quantification method. The authors measure mitophagy and autophagy by reporting the density of red-only puncta, meaning of mitochondria or

autophagosomes that have fused with acidic lysosomes. I find this analysis to provide incomplete information, as it doesn't tell us whether an increase or decrease in this number reflects corresponding changes in the total number of mitochondria or autophagosomes that appear yellow. Therefore, one cannot deduce how homeostatic the differences in mitophagic or autophagic flux and activity are. One would expect that a cell with significantly more mitochondria will correspondingly have more mitolysosomes, but this may not always be as trivial an equation, especially in the aged condition. I believe that it's essential to present the mito- and auto-lysosomes as percentages of all mitochondria and autophagosomes, respectively, for all the different areas and cell types investigated. This will allow for additional or alternative interpretations of the data."

- **We appreciate the reviewer's comments. One of the intriguing early findings from the original characterisation of the *mito*-QC mouse model is that cellular mitophagy levels *in vivo* do not correlate with mitochondrial mass (PMID: 27458135). This highlights the unique complexity of physiological mitochondrial turnover in tissues, where even neighbouring cells can exhibit heterogeneous mitophagy dynamics. In response to this complexity, many researchers, including ourselves, have adopted ratiometric quantification methods for *in vivo* analyses using reporter systems. For example, a mitophagy index can be calculated by dividing the number of acidified mitolysosomes (mCherry-only puncta) by mitochondrial (GFP-only) area, or by dividing the number of mitolysosomes to areas defined by established molecular markers or anatomical features (per μm^2). This approach captures the dynamics of endpoint mitophagy without oversimplification and has become a standard parameter in the field.**
- **Regarding the suggestion to depict the data as percentages, we have plotted this for the reviewer's consideration (please see below). However, as our primary conclusions remain unchanged, we prefer to align our manuscript with established field conventions and report the raw data. We believe that percentage-based analysis is more appropriate for refined scenarios, such as pairwise comparisons of young vs. old animals or in cultured cell paradigms.**

- The proportional relationships between mitophagy and mitochondrial content are not straightforward, and we do not believe there is one “correct” method, although there is consensus that intensity measurements are not optimal. Regardless, our method and approach offers significant advantages over classical techniques that require

tissue disruption, like immunoblotting or citrate synthase assays, which lack spatial resolution and provide only a gross assessment of mitochondrial content. As the field evolves, we anticipate that reporters like *mito-QC* and *auto-QC* will be integrated with additional metrics (e.g. readouts of biogenesis, dynamics and metabolic state) to offer a more comprehensive snapshot of homeostatic degradation and mitochondrial supply – which is not currently feasible. We acknowledge the reviewer's salient point, which adds to the rigor and nuance of our manuscript. We have incorporated these considerations into the discussion to address the reviewer's concerns comprehensively, and we have also worked in some of these alternate calculations into the new Figure EV2 – although our conclusions remain changed.

“2. If I understood it well, when investigating brain areas, 2-D images were analyzed and the results are normalized for area (μm^2). The total tile area imaged for the PFC, DG and CB is not clearly indicated in the materials and methods, making it difficult to judge how representative the resulting data are of the entirety of each structure. However, when analyzing immunolabelled cell types (such as TH-positive neurons, or IBA-1-positive microglia), a 3-D z-stack is imaged and analyzed. Therefore, the results in Figure 4 and 5 should be normalized to volume and not to area.”

- **We appreciate the reviewer's detailed feedback. To clarify, all images for quantification were acquired as single optical sections, where random sampling was employed within defined anatomical regions or for immunolabelled cell subsets. Consequently, all tissue imaging data analysed, including those in Figures 4 and 5, are normalized to area (μm^2), and not volume. Indeed, we have used confocal z-stacks for representative images in some figures, and have now annotated “maximum projection” on photomicrographs where this is the case. We apologise for this lack of clarity and now include more detailed descriptions in the Materials and Methods section.**

“3. While I appreciate that it is challenging for this type of approach to be exhaustive and address all brain cells, the breadth of the study seems fragmented and should be expanded. There can be a justification for focusing on most brain areas or cell subtypes. There are several very interesting brain areas, such as the CA1 and CA3 regions of the hippocampus, the amygdala etc., that have not been considered, making this manuscript not as encompassing as expected for a resource. I urge the authors to at least provide data for all the hippocampal areas to complement their analyses in the dentate gyrus. Similarly, the fact that they have not investigated mitophagy and autophagy in astrocytes and oligodendrocytes, despite the fact that there are good antibodies for immunohistochemistry to mark these cells, is not satisfactory given the mounting interest in autophagy in these pivotal glial cell types. These analyses should be performed and included.”

- **We thank the reviewer for their excellent critique, which we feel has greatly improved the manuscript. The revised study includes substantially more analyses, including:**

- *****NEW ADDITION***** - Profiling of the hippocampal formation in healthy aging (new Figure 4, below, and additional data in EV6-7). This was a fantastic suggestion - our new data reveals that CA1 is a critically affected area during healthy aging. These findings also reveal an interesting interplay between selective and non-selective autophagy in this brain region.

- *****NEW ADDITION***** Discovery and characterisation of autophagy pathways in inhibitory interneurons (integrated into new Figure 4)
- We are additionally excited to include important new data on hippocampal inhibitory interneurons, which to our knowledge, have not been formally investigated using reporter mice before. We find these cells have robust autophagic activity that changes through time. Given their increasing prominence in the modulation of neural circuits, we think the inclusion of this data will enhance the appeal of our manuscript to a broader audience. Unravelling the contribution of autophagy to interneuron function and integrity will prove an exciting future avenue of research.

○ *****NEW ADDITION***** Characterisation of astroglia in distinct regional niches (new Figure 6)

- This was also an excellent suggestion, and we are pleased to include longitudinal macroautophagy and mitophagy data from two distinct astroglial niches in our revised manuscript.
- We note that there are differences in the trajectories of autophagy pathways between astrocytes in these regionally distinct niches. For example, in cerebellar GFAP-positive astrocytes, mitophagy increases significantly throughout life, with only a very modest increase observed in mitophagy during old age (please see new Figure 6A-D below).

- In contrast, in the hippocampal dentate gyrus, mitophagy and macroautophagy are both stable throughout life, with only a very modest increase observed in mitophagy during old age (please see new Figure 6-H below).

- Another intriguing observation here is that compared to neurons and microglia, GFAP+ astroglia accumulate differentially acidified lysosomes at a reduced magnitude – suggesting differences in endolysosomal integrity between different neural populations.
- These collective findings are now presented as primary data in new Figure 6 and associated Figure EV10 of the revised manuscript.

On oligodendrocytes

- We agree that understanding physiological autophagy in aging oligodendrocytes (OGs) is a fascinating question. Despite extensive efforts with different antibodies and protocols, achieving reliable imaging of OG morphology alongside robust reporter signals has proven challenging. We found that while Olig2 antibodies provided clear nuclear labeling, they lacked the specificity to accurately segment OGs, which is required for rigorous analysis of autophagy/mitophagy *in vivo*. Consequently, we switched to CNPase antibodies, which effectively highlighted OG cell morphology using heat-mediated antigen retrieval (HMAR). However, HMAR resulted in the loss of the genetically encoded reporter signal. Despite testing multiple antigen retrieval methods, including chemical and enzymatic approaches, we were unable to maintain both OG staining quality and reporter integrity (see detailed table below). HMAR was effective for CNPase labeling, but also caused superficial tissue damage and requires a different sectioning strategy than our standard cryosections. Due to these technical limitations and the lack of an appropriate cohort of aging mice, we were unable to include OG data in this manuscript. We have also addressed this in the revised limitations section. We intend to integrate and build upon these findings in a future focused study on mitophagy/autophagy in mammalian OGs.

Staining approach								Staining outcome		
Condition	AR method	AR condition	PA	PA dilution	PA condition	SA dilution	SA condition	Reporter signal	Staining fidelity	Tissue integrity
1			AB9610	1:500	o/n RT	1:500	45min RT	✓	×	✓
2			AB9610	1:1000	o/n RT	1:500	45min RT	✓	×	✓
3			MAB326R	1:250	o/n RT	1:500	45min RT	✓	×	✓
4			MAB326R	1:500	o/n RT	1:500	45min RT	✓	×	✓
5			MAB326R	1:250	o/n 4 °C	1:500	45min RT	✓	×	✓
6			AB9610	1:500	o/n 4 °C	1:500	45min RT	✓	×	✓
7			AB9610	1:250	o/n 4 °C	1:500	45min RT	✓	×	✓
8	CB pH 6	20 min 95 °C	MAB326R	1:2000	36h 4 °C	1:1000	o/n 4 °C	×	✓	×
9	CB pH 6	20 min 95 °C	AMAB91072	1:2000	36h 4 °C	1:1000	o/n 4 °C	×	✓	×
10	CB pH 6	2 min 95 °C	AMAB91072	1:2000	o/n 4 °C	1:400	2h RT	×	✓	×
11	PK pH 7.5	15min 37 °C	MAB326R	1:2000	o/n 4 °C	1:500	45min RT	✓	×	×
12	PK pH 8	15min 37 °C	MAB326R	1:2000	o/n 4 °C	1:500	45min RT	✓	✓	×
13	PK pH 7.5	15min 37 °C	MAB326R	1:2000	36h 4 °C	1:500	45min RT	✓	✓	×
14	PK pH 8	15min 37 °C	MAB326R	1:2000	36h 4 °C	1:500	45min RT	✓	✓	×
15	SDS	5 min RT	MAB326R	1:2000	o/n 4 °C	1:500	45min RT	✓	✓	×
16	SDS	5 min RT	MAB326R	1:2000	36h 4 °C	1:500	45min RT	✓	✓	×

Abbreviations:
 AR Antigen retrieval
 PA Primary antibody
 SA Secondary antibody
 CB Citrate buffer
 PK Proteinase K
 SDS Sodium Dodecyl Sulfate
 RT Room temperature
 o/n Over night

“4. The findings on microglia are very interesting. They suggest that both in the DG and the cerebellum, mitophagy follows a similar pattern, increasing progressively with aging, while general autophagy is more or less stable across ages. I wonder whether this reflects the state of microglia as opposed to their homogeneity. IBA-1, the marker used to label these cells, is known to be enriched in active microglia, which may be more abundant or activated to a greater extent in the aged brain. To exclude this possibility, the authors should use an additional microglial markers, such as P2Y12R for example.”

- We appreciate the reviewer’s positive remarks. We optimized staining for P2Y12R antibodies, although the pattern was not as robust as our anti-IBA1 protocol in terms of intensity, uniformity, and background. While IBA1 uniformly labels the entire microglial cytoplasm with minimal background, P2Y12R staining shows distinct localization within the fine arbors and perinuclear regions of microglia. Unfortunately, our initial plan to co-label IBA1 and P2Y12R and quantify microglial subpopulations was hindered by species cross-reactivity between the antibodies, making multiplexing unfeasible in this study. Regarding the interplay between mitophagy, macroautophagy, and microglial activation, we agree it is a fascinating topic. However, establishing the dynamic range of microglial activation in a controlled paradigm is crucial for interpreting insights from healthy aging contexts. While a comprehensive analysis falls outside the scope of our current manuscript, we recognize the importance of studying microglial activation states and mitochondrial regulation in depth. This is an area that warrants future investigation, and we are committed to pursuing it formally.

“5. With regards to Figure 3, there's a lot of evidence that LAMP1 is a promiscuous marker that does not exclusively mark lysosomes (see for example PMID: 29940787). In neuronal dendrites, about 70% of LAMP1-positive structure contain lysosomal enzymes (see for example PMID: 37590146). Antibodies against active lysosomal enzymes and other lysosomal markers, such as

LAMP2, should be used to support any claim to the fusion of the yellow mitochondria with non-acidified lysosomes.”

- To address this great point, we have now reprobated sections with antibodies to four additional lysosomal epitopes, which include LAMP2, Cathepsin-B, Cathepsin-D and TMEM55B (see below, new Figure EV6). Mitolysosomes stained positively for all markers, and mCherry-only mitolysosomes have also been validated by the group of Åsa Birgisdottir using correlative light and transmission electron microscopy/CLEM – PMID: 37405374. We also provide functional evidence to reviewer 1 that the formation of red-only structures in both reporter systems is heavily dependent on lysosomal activity, through cell culture based experiments with lysosomotropic inhibitors (see new Figure EV1).

- **Although LAMP1 marks acidic endolysosomal compartments (late endosomes and lysosomes), its function remained elusive until a recent study demonstrated a critical role for both LAMP1 and LAMP2 play in ensuring lysosomal acidification via a TMEM175-mediated mechanism (PMID: 37390818). This highlights the complexity of lysosomal biology in tissues and we consider the use of LAMP1 to be appropriate in the current context.**

“Minor concerns:

1. In their discussion, the authors should compare their approach not only to the mito-keima reporter mouse, but also to what has been recently learned about mitophagy from other types of approaches, such as proteomic analyses of purified brain autophagosomes and of autophagosome enriched brain fractions.

- **We thank the reviewer again for their astute suggestion. We strongly agree that combining optical reporter systems such as ours, with such emergent proteomics strategies, represents a powerful path towards an integrative understanding of physiological autophagy pathways. Indeed, Nikolettou, Dengjel and colleagues recently published a study investigating autophagic cargo subtypes from brain tissues (PMID: 37279748) and a similar proteomics-based approach was taken by the Holzbaur group (PMID: 35051374). As requested, we now reference this approach in our revised discussion.**

2. It seems that some graphs are repeated in the main figures. Unless I am mistaken, the top graphs of Figure 3d and e are the same as panels Figure 2f and l, respectively. A better arrangement should be applied to avoid such repetitions.

- **We apologise for any confusion. We have revisited these graphs and they are actually distinct from each other. We have made a better effort to annotate all metrics in the revised manuscript to better differentiate each dataset. The only instance is Figure EV8gh, EV9k-t, for data completeness in the extended version graphs – and where this has occurred, it is clearly annotated in the figure/legend.**

3. High magnification representative images should also be provided for the green channel in Figure 2 and where else applicable.”

- **We appreciate this suggestion. To comply with the reviewers request, we now include all channels for Figure 2 in the the new Figure EV2, as the main figure is already densely populated.**

Dear Dr McWilliams,

Thank you for submitting your revised manuscript (EMBOJ-2023-115700R) to The EMBO Journal. As mentioned, your amended study was sent back to the two referees for their re-evaluation, and we have received comments from both of them, which I enclose below. As you will see, the experts stated that the work has been substantially improved by the revisions and they are now in favour of publication, pending minor revision.

Thus, we are pleased to inform you that your manuscript has been accepted in principle for publication in The EMBO Journal.

Please consider the remaining points by the referees carefully and adjust the text where appropriate, introducing additional discussion and caveats.

Also, we now need you to take care of a number of minor issues related to formatting and data presentation as detailed below, which should be addressed at re-submission.

Please contact me at any time if you have additional questions related to below points.

As you might have seen on our web page, every paper at the EMBO Journal now includes a 'Synopsis', displayed on the html and freely accessible to all readers. The synopsis includes a 'model' figure as well as 2-5 one-short-sentence bullet points that summarize the article. I would appreciate if you could provide this figure and the bullet points.

Thank you for giving us the chance to consider your manuscript for The EMBO Journal.
I look forward to your final revision.

Again, please contact me at any time if you need any help or have further questions.

Best regards,

Daniel Klimmeck

>> Please limit the number of keywords for your study to maximally five.

>> Limit the abstract length to maximally 175 words.

>> Author Contributions: Please remove the author contributions information from the manuscript text. Note that CRediT has replaced the traditional author contributions section as of now because it offers a systematic machine-readable author contributions format that allows for more effective research assessment. and use the free text boxes beneath each contributing author's name to add specific details on the author's contribution.

More information is available in our guide to authors.
<https://www.embopress.org/page/journal/14602075/authorguide>

>> Please change the title of the 'Competing Interests Declaration' to 'Disclosure and Competing Interests Statement'.

>> Section order should be corrected as follows: title page with complete author information, abstract, keywords, introduction, results, discussion, methods, data availability section, acknowledgements, disclosure and competing interests statement, references, main figure legends, tables, expanded figure legends.'

>> Appendix-supplementary material: There are currently ten supplemental figures. Up to five can be made EV figures, uploaded as individual figure files, and their legends should be in the manuscript, under the heading Expanded View Figure Legends. The remaining supplemental figures should be compiled in an appendix: a PDF File containing a table of contents on its first page followed by the figures and their legends ideally placed underneath each corresponding figure. The correct nomenclature is "Appendix Figure S1" etc. .

>> Callouts: callouts for Figure panels Fig.6E, G need to be added to the manuscript.

>> Funding: information on funding is incomplete in our online system. 'Academy of Finla TGM 310814' is missing in in our online system and in our online system contains several funders that are not mentioned in the manuscript. Please ensure that the funding information matches and please add project numbers where available.

>> Data availability section: please move the Data availability section to the end of the Methods part.

>> Source data: provide a completed source data checklist; provide image data in our manuscript system or deposit externally, adding a link to the Data availability section.

>> Complete the Author Checklist, including the 'Human research participants' and 'Core Facilities' parts.

>> Indicate redisplay of cellular data from Figure 2a-h in the figure legend of Fig. EV2.

>> Consider additional changes and comments from our production team as indicated below:

1. Please note that the box plots need to be defined in terms of minima, maxima, centre, bounds of box and whiskers, and percentile in the legends of figures 2b, d, f, h, j, l; 3d-e, g; 4b, d, f, h; 5c, e, g, i; 6b, d, f, h; 7b, d, f, h; 8c, e; EV 1c-g; EV 2b, d, f, h, j, l; EV 3a-ah; EV 4a-ah; EV 5a-b, d-e, g-h; EV 8a-ah; EV 9a-ah; EV 10a-ah.
2. Please note that information related to n is missing in the legends of figures EV 5b.
3. Although 'n' is provided, please describe the nature of entity for 'n' in the legends of figures 4b, d, f, h; 5c, e, i; 8e; EV 1c-g; EV 3m, o.

- Figure legends:

1. Please note that the exact p values are not provided in the legends of figures 2f; 4b, d, f, h; 7a, f; EV 1c-g; EV 2d, f, h, j; EV 3a, c, e, g, i, k, q, s, u, w, y, aa, ac, ae; EV 4c, e, k, m, o; EV 5a, c-e; EV 8a, c, e, g, i, k, q, s, u, w, y, aa, ac, ae; EV 10b, f, h, j-r, t, ab-af.
2. Please indicate the statistical test used for data analysis in the legends of figures EV 5a-b.
3. Please note that in figures 2f, h, j; 3d, g; 4b, d, f, h; EV 3c, e, k, q, u; EV 4e; EV 8c, e, g, u, ae; there is a mismatch between the annotated p values in the figure legend and the annotated p values in the figure file that should be corrected.
3. Please note that the scale bar needs to be defined for figures EV 1a-b.
4. Please note that scale bar and its definition are missing for figures 2a, c, g, i, k.

Referee #1:

The authors provide an outstanding amount of new data in the revised version of the manuscript, with the most significant being the new data on DALs and demonstrating the longitudinal changes on lysosomal acidification in the aged brain. As there was no availability of aged mice, no experiments could be done to address one of the main concerns in the previous round: whether some of the accumulation of mitoQC red structures in aged brains were a consequence of a subtle defect in lysosomal acidification, which can decrease mitophagic flux by impairing the complete digestion of mitophagolysosome content without quenching mitoQC green fluorescence.

While I appreciate very much the experiments showing that baflomycin treatment eliminates the mitoQC red structures in cell lines, the defect in lysosomal acidification induced by this antibiotic cannot be compared to the milder defect in lysosomal acidification induced by ageing. In addition, the other examples discussed by the authors (Atg5, mTOR in flies) consist of

manipulations primarily blocking autophagosome formation and initiation, not clearance. Thus, the question on whether increased mitophagic flux is what the increase in red mitophagosome structures is reporting in the aged brain is reporting is still unclear. On the other hand, there is no question that a higher number of mitophagolysosomal structures are increased with age in certain areas of the brain. And this information is very relevant.

Minor concerns:

My recommendation would be to claim that changes in mitophagolysosome content exist with aging, but that whether this change represents increased formation of mitophagolysosomes or decreased elimination of mitophagolysosomes remains to be determined. The new DAL data however supports that a likely scenario is that the increase in red mitophagolysosomes observed with ageing might a consequence of decreased clearance with age, caused by a mild defect in lysosomal acidification in aged brains. In other words, this longitudinal study uniquely measuring autophagy and mitophagy concurrently support that lysosomal dysfunction is one of the earlier events failing in some neurons in ageing, and that this might initiate the accumulation of mitophagolysosomes (red or yellow, depending on the degree in decreased acidification).

This is a highly relevant point, as it is suggesting that anti-ageing approaches increasing mitophagolysosome formation without restoring lysosomal acidification could be more harmful than beneficial. In this regard, Figure EV6 should show quantifications of DALs.

Additional concerns:

There is a sentence in the results that needs to be revised: "When mitochondria are destroyed during mitophagy, the low pH environment of the acidic endolysosome quenches GFP fluorescence, while the mCherry signal remains stable". The mitoQC protein, being in the outer membrane, will be one of the first mitochondrial component to be completely degraded by lysosomal proteases. Thus, it is unlikely that mitoQC will survive and shine red light when all mitochondria are completely degraded. Authors should be more accurate and should just state that the decrease in pH caused by lysosomal fusion to the autophagosome is what quenches the GFP, which occurs and has to occur before complete destruction.

For the rest of the paper, I want to congratulate the authors for the hard work they did.

Referee #3:

The revised manuscript by Rappe and colleagues is significantly improved compared to its original version. Most of my concerns have been addressed by the added experiments and analyses. This is a tour de force resource that will be valuable to the fields of brain autophagy and brain aging. However, I still have some important comments related to semantics, which need clarification or experimental work to avoid misunderstandings within the community, or mislead readers.

1. Line 115: "To uncouple basal mitophagy events from nonselective autophagy in vivo, we also analyzed brain tissues from auto-QC autophagy reporter mice across the same aging trajectory". This sentence is misleading, as the auto-QC reporter mouse doesn't report only nonselective autophagy, but also all forms of selective autophagy (including mitophagy) that are mediated by LC3-positive autophagosomes. This "non-selective autophagy" should be replaced by "general autophagy, pan-autophagy, other forms of autophagy" or equivalent throughout the text (abstract, and lines 81, 115, 185, 228, 385, 635 etc)

2. Lines 257-261: DAL (differentially acidified lysosomes) is introduced in the revised manuscript as a term to describe structures that retain both GFP and mCherry LC3-fluorescence and are also positive for LAMP1. I find that calling these structures lysosomes is unsubstantiated, in view of mounting evidence that LAMP1 is a promiscuous marker in organelles of the autophagy-lysosomal pathway. For example, in this recent work that I already mentioned, <https://doi.org/10.1083/jcb.201711083>, immuno-EM was performed on spinal cord sections with an antibody against LAMP1: signals were readily detected around the surface of various endosome- and lysosome-like organelles including mature lysosomes, autolysosomes, immature autophagosomes with double membranes containing nondegraded translucent contents, multivesicular bodies (MVBs) containing multiple internal vesicles, and multilamellar bodies (MLBs) composed of concentric membrane layers (presented in Fig. 3). Therefore, it's plausible that what is referred to here as DALs are a subpopulation of autophagic vesicles that expresses LAMP1 or that have fused with LAMP1+ endosomes. The fact that such structures (whatever they may be) accumulate with aging is interesting in itself and invites further investigation. However, in the absence of CLEM microscopy to describe the ultrastructural morphology of these puncta as single-membraned autolysosomes that are not acidified, the whole claim is weak and should be rephrased to avoid misleading readers. To retain this argument, the authors need to present the ultrastructure of the LAMP1+/GFP+/mCherry+ puncta.

More generally, if these do turn out to be lysosomes, the use of different terms, for example differentially or poorly acidified lysosomes is confusing without a molecular profile that distinguishes them from each other (or if they are the same we should call them with one name).

Minor comment:

Line 320: The word "Discovery" sounds pompous. There's no biological reason to suspect that interneurons should be devoid of

autophagic pathways, therefore, I find that "characterization" is sufficient and more appropriate.

Author's response

We thank the Editor and Reviewers for their constructive contributions, which have strengthened our manuscript. We are grateful for the insightful and collegial exchange during the peer review process. We will integrate these suggestions into the final manuscript as detailed below.

**On behalf of the authors,
Tom McWilliams**

Referee #1:

“The authors provide an outstanding amount of new data in the revised version of the manuscript, with the most significant being the new data on DALs and demonstrating the longitudinal changes on lysosomal acidification in the aged brain. As there was no availability of aged mice, no experiments could be done to address one of the main concerns in the previous round: whether some of the accumulation of mitoQC red structures in aged brains were a consequence of a subtle defect in lysosomal acidification, which can decrease mitophagic flux by impairing the complete digestion of mitophagolysosome content without quenching mitoQC green fluorescence.

While I appreciate very much the experiments showing that bafilomycin treatment eliminates the mitoQC red structures in cell lines, the defect in lysosomal acidification induced by this antibiotic cannot be compared to the milder defect in lysosomal acidification induced by ageing. In addition, the other examples discussed by the authors (Atg5, mTOR in flies) consist of manipulations primarily blocking autophagosome formation and initiation, not clearance. Thus, the question on whether increased mitophagic flux is what the increase in red mitophagosome structures is reporting in the aged brain is reporting is still unclear. On the other hand, there is no question that a higher number of mitophagolysosomal structures are increased with age in certain areas of the brain. And this information is very relevant.”

Minor concerns:

My recommendation would be to claim that changes in mitophagolysosome content exist with aging, but that whether this change represents increased formation of mitophagolysosomes or decreased elimination of mitophagolysosomes remains to be determined. The new DAL data however supports that a likely scenario is that the increase in red mitophagolysosomes observed with ageing might a consequence of decreased clearance with age, caused by a mild defect in lysosomal acidification in aged brains. In other words, this longitudinal study uniquely measuring autophagy and mitophagy concurrently support that lysosomal dysfunction is one of the earlier events failing in some neurons in ageing, and that this might initiate the accumulation of mitophagolysosomes (red or yellow, depending on the degree in decreased acidification).”

We are grateful to the Reviewer for recognising the breadth and depth of our study and appreciate their helpful feedback. We agree and acknowledge the complexity

of comparing acidification phenotypes between acute in vitro and chronic in vivo paradigms. Our results show that aging clearly alters the dynamics and trajectories of mitochondrial turnover and lysosomal homeostasis in a cell-type and region-specific manner. We also agree that there is much to learn and remain open to alternative interpretations. We modified the discussion to reflect this perspective:

"Importantly, we cannot preclude that progressive lysosomal dysfunction may in some cases impair autophagic flux, which may somehow contribute to the accumulation of red-only puncta." – Lines 572-574.

"This is a highly relevant point, as it is suggesting that anti-ageing approaches increasing mitophagolysosome formation without restoring lysosomal acidification could be more harmful than beneficial. In this regard, Figure EV6 should show quantifications of DALs."

- **We thank the Reviewer for highlighting this salient issue. Indeed, while mitophagy remains an important target, emerging therapeutics should be assessed for their impact on endolysosomal homeostasis and substrate digestion, in addition to increasing turnover. This point is emphasised in the revised discussion:**

"Our findings highlight the importance of integrating autophagy-enhancing therapeutic strategies with assessments of endolysosomal homeostasis to prevent cellular dysfunction and achieve effective neuroprotection." – Lines 585-587.

"Additional concerns:

There is a sentence in the results that needs to be revised: "When mitochondria are destroyed during mitophagy, the low pH environment of the acidic endolysosome quenches GFP fluorescence, while the mCherry signal remains stable". The mitoQC protein, being in the outer membrane, will be one of the first mitochondrial component to be completely degraded by lysosomal proteases. Thus, it is unlikely that mitoQC will survive and shine red light when all mitochondria are completely degraded. Authors should be more accurate and should just state that the decrease in pH caused by lysosomal fusion to the autophagosome is what quenches the GFP, which occurs and has to occur before complete destruction."

- **We appreciate the Reviewer's astute observation and have modified the sentence to focus on the specifics as follows:**

"During mitophagy, the fusion of acidic endolysosomes with autophagosomes quenches GFP fluorescence, whereas the mCherry signal remains unaffected by the pH shift." – Lines 111-113.

"For the rest of the paper, I want to congratulate the authors for the hard work they did."

- **We thank the Reviewer for their supportive remarks, recognition of the work, and for their valuable contributions to improving the manuscript.**

Referee #3:

“The revised manuscript by Rappe and colleagues is significantly improved compared to its original version. Most of my concerns have been addressed by the added experiments and analyses. This is a tour de force resource that will be valuable to the fields of brain autophagy and brain aging. However, I still have some important comments related to semantics, which need clarification or experimental work to avoid misunderstandings within the community, or mislead readers.”

- **We are grateful to the Reviewer for their positive remarks and constructive feedback, which we feel has greatly strengthened our manuscript.**

“1. Line 115: “To uncouple basal mitophagy events from nonselective autophagy in vivo, we also analyzed brain tissues from auto-QC autophagy reporter mice across the same aging trajectory”. This sentence is misleading, as the auto-QC reporter mouse doesn't report only nonselective autophagy, but also all forms of selective autophagy (including mitophagy) that are mediated by LC3-positive autophagosomes. This “non-selective autophagy” should be replaced by “general autophagy, pan-autophagy, other forms of autophagy” or equivalent throughout the text (abstract, and lines 81, 115, 185, 228, 385, 635 etc)”

- **We have now updated the text with either “macroautophagy” or “general autophagy”, as appropriate.**

“2. Lines 257-261: DAL (differentially acidified lysosomes) is introduced in the revised manuscript as a term to describe structures that retain both GFP and mCherry LC3-fluorescence and are also positive for LAMP1. I find that calling these structures lysosomes is unsubstantiated, in view of mounting evidence that LAMP1 is a promiscuous marker in organelles of the autophagy-lysosomal pathway. For example, in this recent work that I already mentioned, <https://doi.org/10.1083/jcb.201711083>, immuno-EM was performed on spinal cord sections with an antibody against LAMP1: signals were readily detected around the surface of various endosome- and lysosome-like organelles including mature lysosomes, autolysosomes, immature autophagosomes with double membranes containing nondegraded translucent contents, multivesicular bodies (MVBs) containing multiple internal vesicles, and multilamellar bodies (MLBs) composed of concentric membrane layers (presented in Fig. 3). Therefore, it's plausible that what is referred to here as DALs are a subpopulation of autophagic vesicles that expresses LAMP1 or that have fused with LAMP1+ endosomes. The fact that such structures (whatever they may be) accumulate with aging is interesting in itself and invites further investigation. However, in the absence of CLEM microscopy to describe the ultrastructural morphology of these puncta as single-membraned autolysosomes that are not acidified, the whole claim is weak and should be rephrased to avoid misleading readers. To retain this argument, the authors need to present the ultrastructure of the LAMP1+/GFP+/mCherry+ puncta. More generally, if these do turn out to be lysosomes, the use of different terms, for example differentially or poorly acidified lysosomes is confusing without a molecular profile that distinguishes them from each other (or if they are the same we should call them with one name).”

- **We value the Reviewer's insightful feedback. We selected "differentially acidified" over "de-acidified" or "poorly acidified" lysosomes (PALs - coined by the Nixon lab) to capture the possibility that endolysosomal impairments may occur either before or after mitophagosome/autophagosome fusion. We find this simplified nomenclature reflects our observations amidst ongoing efforts in the field. We acknowledge the reported promiscuity of LAMP1 as a marker in the autophagy-lysosomal pathway, as highlighted by the reviewer (PMID: 29695488), where the authors also use "differential" in the context of endolysosomal maturation. We respectfully refer to our previous response, noting the argument against LAMP1 as a lysosomal marker is not definitive, given its recently reported role in acidification (PMID: 37390818). We agree on the importance of determining the precise molecular identity of these structures and are committed to pursuing this in future work. We have updated the discussion to address the Reviewer's concern, citing the referenced study and emphasizing the need for further research.**

“Further understanding the molecular profile and mechanistic origin of the differentially acidified LAMP1-positive puncta will be particularly important, as LAMP1 can demarcate a variety of endolysosomal structures (Cheng et al. 2018).” – Lines 576-579.

“Minor comment:

Line 320: The word "Discovery" sounds pompous. There's no biological reason to suspect that interneurons should be devoid of autophagic pathways, therefore, I find that "characterization" is sufficient and more appropriate.”

- **We appreciate the Reviewer's feedback and have amended the text accordingly.**

Dear Dr McWilliams,

Thank you for submitting the revised version of your manuscript. I have now evaluated your amended manuscript and concluded that the remaining minor concerns have been sufficiently addressed.

I am thus pleased to inform you that your manuscript has been accepted for publication in the EMBO Journal.

Related, I kindly ask for your consent on keeping the referee figures included in this file.

On a different note, I would like to alert you that EMBO Press offers a format for a video-synopsis of work published with us, which essentially is a short, author-generated film explaining the core findings in hand drawings, and, as we believe, can be very useful to increase visibility of the work. Please see the following link for representative examples and their integration into the article web page:

<https://www.embopress.org/doi/full/10.15252/emj.2019103932>

Finally, we have noted that the submitted version of your article is also posted on the preprint platform bioRxiv. We would appreciate if you could alert bioRxiv on the acceptance of this manuscript at The EMBO Journal in order to allow for an update of the entry status. Thank you in advance!

Best regards,

Daniel Klimmeck

Daniel Klimmeck, PhD
Senior Editor
The EMBO Journal
EMBO
Postfach 1022-40
Meyerohofstrasse 1

D-69117 Heidelberg
contact@embojournal.org
Submit at: <http://emboj.msubmit.net>